MIT-CTP/5732

# Gauging modulated symmetries:
# Kramers-Wannier dualities and non-invertible reflections

Salvatore D. Pace,[1] Guilherme Delfino,[2] Ho Tat Lam,[1] and Ömer M. Aksoy[1]

[1] *Department of Physics, Massachusetts Institute of Technology, Cambridge, MA 02139, USA*

[2] *Physics Department, Boston University, Boston, MA, 02215, USA*

July 2, 2024

## Abstract

Modulated symmetries are internal symmetries that act in a non-uniform, spatially modulated way and are generalizations of, for example, dipole symmetries. In this paper, we systematically study the gauging of finite Abelian modulated symmetries in $1+1$ dimensions. Working with local Hamiltonians of spin chains, we explore the dual symmetries after gauging and their potential new spatial modulations. We establish sufficient conditions for the existence of an isomorphism between the modulated symmetries and their dual, naturally implemented by lattice reflections. For instance, in systems of prime qudits, translation invariance guarantees this isomorphism. For non-prime qudits, we show using techniques from ring theory that this isomorphism can also exist, although it is not guaranteed by lattice translation symmetry alone. From this isomorphism, we identify new Kramers-Wannier dualities and construct related non-invertible reflection symmetry operators using sequential quantum circuits. Notably, this non-invertible reflection symmetry exists even when the system lacks ordinary reflection symmetry. Throughout the paper, we illustrate these results using various simple toy models.

# 1 Introduction

Gauging is a formal procedure used throughout theoretical physics, where starting from a theory $\mathfrak{T}$ with global symmetry $\mathcal{S}$, gauging $\mathcal{S}$ produces a new theory $\mathfrak{T}/\mathcal{S}$ with global symmetry $\mathcal{S}^\vee$ [1]. The so-called dual symmetry $\mathcal{S}^\vee$ is determined by $\mathcal{S}$ and details of the gauging procedure, i.e., discrete torsion [2]. Since $\mathfrak{T}/\mathcal{S}$ can be constructed from gauging $\mathcal{S}$, it can be presented as an $\mathcal{S}$ gauge theory in which $\mathcal{S}^\vee$ is the magnetic symmetry. Even when $\mathcal{S}$ is an ordinary symmetry, $\mathcal{S}^\vee$ can be various types of generalized symmetries. Therefore, given the relationship between $\mathcal{S}$ and $\mathcal{S}^\vee$, gauging provides a systematic way to construct theories $\mathfrak{T}/\mathcal{S}$ with possible generalized symmetries. See [3–11] for reviews on generalized symmetries.

When $\mathcal{S}$ is a finite symmetry, $\mathcal{S}^\vee$ is always nontrivial and finite, and there exists a gauging procedure for $\mathcal{S}^\vee$ in which $\mathfrak{T}/\mathcal{S}$ returns to $\mathfrak{T}$ (i.e., $\mathfrak{T} = (\mathfrak{T}/\mathcal{S})/\mathcal{S}^\vee$). Therefore, the theory $\mathfrak{T}$ and $\mathfrak{T}/\mathcal{S}$ contains the same physical information. In particular, the phases and phase transitions of $\mathfrak{T}$ are in a one-to-one correspondence to those of $\mathfrak{T}/\mathcal{S}$ and can be inferred from the latter [12, 13]. This relation reflects a more fundamental correspondence between finite symmetries $\mathcal{S}$ and classes of gapped boundaries of related topological orders $\mathcal{Z}(\mathcal{S})$ in one-higher dimension, known as SymTFTs [12, 14–26].

An important class of finite symmetries are those that satisfy $\mathcal{S}^\vee = \mathcal{S}$. For such $\mathcal{S}$, the corresponding isomorphism between local symmetric operators in $\mathfrak{T}$ and $\mathfrak{T}/\mathcal{S}$ is called a Kramers-Wannier (KW) duality [27–31].[1] For example, when $\mathcal{S}$ is an invertible 0-form symmetry in $1 + 1$D, $\mathcal{S}^\vee = \mathcal{S}$ whenever $\mathcal{S}$ is described by a finite Abelian group. In $d + 1$D, gauging a $p$-form symmetry $\mathcal{S}$, leads to a $(d - p - 1)$-form symmetry $\mathcal{S}^\vee$ [32]. Therefore, a KW duality can exist only if for each $q$-form symmetry, its corresponding dual $(d - q - 1)$-form symmetry also appears in $\mathcal{S}$ [33, 34]. KW dualities can also arise from gauging subsystem symmetries [35, 36], dipole symmetries [36, 37] and non-invertible symmetries [38–40].

Since local Hamiltonians are sums of local symmetric operators, a KW duality maps an $\mathcal{S}$-symmetric Hamiltonian $H$ to an $\mathcal{S}$-symmetric Hamiltonian $H^\vee$. Because $H$ and $H^\vee$ are related by gauging, there is a canonical isomorphism between their Hilbert spaces, and we denote by $\mathsf{D}_{\mathrm{KW}}$ the operator implementing the KW duality,

$$\mathsf{D}_{\mathrm{KW}} H = H^\vee \mathsf{D}_{\mathrm{KW}}. \tag{1.1}$$

The duality operator $\mathsf{D}_{\mathrm{KW}}$ is always non-invertible, annihilating states charged under the $\mathcal{S}$ symmetry. In the contexts of Hamiltonian lattice models, various KW duality operators $\mathsf{D}_{\mathrm{KW}}$ have been constructed (see, for instance, Refs. 35, 37, 41–52).

When $H = H^\vee$, $\mathsf{D}_{\mathrm{KW}}$ commutes with the Hamiltonian $H$ and the KW duality becomes a non-invertible symmetry of $H$, enlarging the $\mathcal{S}$ symmetry already present. As for ordinary symmetries, the KW symmetries can be used to characterize spontaneous symmetry breaking [44, 49, 52–55] and

---

[1]Strictly speaking, a KW duality is not an actual duality because it is an isomorphism between only local symmetric operators of two theories. A genuine duality, such as electromagnetic duality, provides an isomorphism between *all* operators of two theories, symmetric or charged, local or non-local.

symmetry protected topological phases [47, 49]. They can also have 't Hooft anomalies that constrain the dynamics of the systems [33, 44, 52–54, 56–60].

## 1.1   Modulated symmetries

While the dual symmetries $\mathcal{S}^\vee$ arising from gauging invertible finite 0-form symmetries [15, 32, 61–65] and higher-form/higher-group symmetries [66–68] has received much attention, there has yet to be an exploration of $\mathcal{S}^\vee$ obtained by gauging finite modulated symmetries. Modulated symmetries are internal symmetries that act in a non-uniform, spatially modulated way. They have been explored in various settings and shown to give rise to slow thermalization and Hilbert space fragmentation [69–76], UV/IR mixing [77–79], and fractons [80–85]. Furthermore, they have been used in characterizing symmetry-enriched topologically ordered phases [79, 86–91], spontaneous symmetry broken phases [92–98], and symmetry protected topological phases [99–109].

For invertible 0-form symmetries, the above colloquial definition of modulated symmetries is formalized as follows. Suppose the internal symmetries are described by a group $G_{\text{int}}$, and the spatial symmetries by $G_{\text{space}}$. When the internal symmetries are not modulated, the total symmetry group is $G_{\text{sym}} = G_{\text{int}} \times G_{\text{space}}$. However, when internal symmetries are spatially modulated, their symmetry transformation is position-dependent, so $G_{\text{space}}$ has a nontrivial action on $G_{\text{int}}$. Letting this action be described by the group homomorphism

$$\varphi \colon G_{\text{space}} \to \text{Aut}(G_{\text{int}}), \tag{1.2}$$

where $\text{Aut}(G_{\text{int}})$ is the automorphism group of $G_{\text{int}}$, the total symmetry group is the semidirect product[2]

$$G_{\text{sym}} = G_{\text{int}} \rtimes_\varphi G_{\text{space}}. \tag{1.3}$$

Each group element $g \in G_{\text{int}}$ is represented by a unitary operator $U_{f_g}^{(g)}$, where $\{f_g\}$ are functions describing the operator's spatial modulation. In this paper, we assume space is a lattice $\Lambda$, which makes $\{f_g(\boldsymbol{a})\}$ lattice functions with $\boldsymbol{a} \in \Lambda$. Assuming the internal symmetry operators are onsite, under a lattice transformation $s \in G_{\text{space}}$ represented by $T^{(s)}$, they transform as[3]

$$T^{(s)} U_{f_g}^{(g)} T^{(s)\dagger} = U_{f_g \circ s}^{(g)} \equiv U_{f_{\varphi_s(g)}}^{(\varphi_s(g))}, \tag{1.4}$$

which shows how the modulated functions $\{f_g\}$ encode the group homomorphism $\varphi$. When using unitary operators $U_{f_g}^{(g)}$ to describe a modulated symmetry (1.3), it is important to ensure $U_{f_g}^{(g)}$ are closed the under lattice symmetries. In particular, if a Hamiltonian commutes with $U_{f_g}^{(g)}$ and $T^{(s)}$, then it must also commute with $U_{f_{\varphi_s(g)}}^{(\varphi_s(g))}$. Furthermore, Eq. (1.4) implies that $U_{f_g}^{(g)} = U_{f_g \circ s}^{(g)}$ if $T^{(s)} = 1$, which

---

[2]The semidirect product of $H$ and $N$ is a group $G = N \rtimes_\varphi H$ defined by the group homomorphism $\varphi \colon H \to \text{Aut}(N)$ that describes the action of $H$ on $N$. The group elements of $G$ are $(n, h) \in N \times H$ and they obey the group multiplication $(n_1, h_1) \cdot (n_2, h_2) = (n_1 \cdot \varphi_{h_1}(n_2), h_1 \cdot h_2)$. It is straightforward to verify using the group multiplication rule that $N$ is a normal subgroup of $G$ and that $h \, n \, h^{-1} = \varphi_h(n)$.

[3]Throughout this paper, we assume spatial symmetries act on operators as a passive transformation.

gives rise to constraints, for example, from lattice translations with periodic boundary conditions. In this paper, we work in one-dimensional space and require $G_{\text{space}}$ to include at least lattice translations.

For modulated subsystem and higher-form symmetries (e.g., see Refs. 86, 110–113), the above definition needs slight modification. For these generalized symmetries, the symmetry operators $S_a(\Sigma)$ are labeled by the symmetry element $a$ and the closed subspace $\Sigma$ on which they act. For higher-form symmetries, $S_a(\Sigma)$ and $S_a(\Sigma')$ are equivalent if $\Sigma$ and $\Sigma'$ are in the same (cellular) homology classes $[\Sigma'] = [\Sigma]$, while for subsystem symmetries, $S_a(\Sigma')$ and $S_a(\Sigma)$ need not be equivalent. When acting the operator $T^{(s)}$ representing the proper transformation $s \in G_{\text{space}}$ on $S_a(\Sigma)$, the most general transformation is

$$T^{(s)} S_a(\Sigma) T^{(s)\dagger} = S_{a_s}(\Sigma_s). \tag{1.5}$$

If $a = a_s$, then $S_a(\Sigma)$ is a non-modulated operator; otherwise, it is modulated.

A canonical example of a modulated invertible 0-form symmetry is a $U(1)$ dipole symmetry. On a $d$-dimensional infinite lattice $\Lambda$, a $U(1)$ dipole symmetry is a $U(1)^{\times 1+d}$ internal symmetry[4] with conserved symmetry charge operators

$$N = \sum_{\boldsymbol{a} \in \Lambda} n_{\boldsymbol{a}}, \qquad \boldsymbol{P}_j = \sum_{\boldsymbol{a} \in \Lambda} \boldsymbol{a}_j \, n_{\boldsymbol{a}}, \tag{1.6}$$

where $n_{\boldsymbol{a}}$ is the local boson number operator at site $\boldsymbol{a} \in \Lambda$ and $\boldsymbol{a}_j$ corresponds to the $j$-th component of lattice vector $\boldsymbol{a}$ for $j = 1, \ldots, d$. Therefore, systems with $U(1)$ dipole symmetry conserve both total particle number and dipole moment under Hamiltonian time evolution. The internal $U(1)^{\times 1+d}$ symmetry is generated by the unitary operators

$$U_1^{(\alpha)} = \exp[\mathrm{i}\,\alpha\,N], \qquad U_{\boldsymbol{a}_j}^{(\beta)} = \exp[\mathrm{i}\,\beta\,\boldsymbol{P}_j], \tag{1.7}$$

where $U_{\boldsymbol{a}_j}^{(\beta)}$ are modulated symmetry operators. Under a lattice translation by the lattice vector $\boldsymbol{x}$, these modulated symmetry operators satisfy

$$T_{\boldsymbol{x}} U_{\boldsymbol{a}_j}^{(\beta)} T_{\boldsymbol{x}}^{\dagger} = U_{\boldsymbol{a}_j + \boldsymbol{x}_j}^{(\beta)} \equiv (U_1^{(\beta)})^{\boldsymbol{x}_j} U_{\boldsymbol{a}_j}^{(\beta)}. \tag{1.8}$$

Therefore, the total symmetry group is

$$G_{\text{sym}} = U(1)^{\times 1+d} \rtimes_{\varphi} G_{\text{space}}, \tag{1.9}$$

where $\varphi \colon G_{\text{space}} \to \text{Aut}(U(1)^{\times 1+d})$ captures (1.8) and other transformations deduced from (1.6).

## 1.2 Summary

Consider translation invariant systems of $\mathbb{Z}_N$ qudits residing on sites $j$ of a one-dimensional spatial chain and acted on by the clock and shift operators $\mathcal{Z}_j$ and $\mathcal{X}_j$, respectively. We suppose such systems have finite invertible modulated symmetries generated by the symmetry operators

$$U_q := \prod_j (\mathcal{X}_j)^{f_j^{(q)}}, \tag{1.10}$$

---

[4]For a group $G$, we use the notation $G^{\times n} := \overbrace{G \times G \times \cdots \times G}^{n \text{ copies of } G}$.

with $q = 1, 2, \cdots n$. The lattice functions $f_j^{(q)} \in \mathbb{Z}_N$ encode the homomorphism $\varphi$ and are linearly independent over $\mathbb{Z}_N$. Therefore, the internal symmetry group $G_{\text{int}}$ is finite Abelian and a subgroup of $\mathbb{Z}_N^{\times n}$. Gauging this $G_{\text{int}}$ modulated symmetry amounts to gauging the $G_{\text{int}}$ sub-symmetry of the total $G_{\text{sym}} = G_{\text{int}} \rtimes_{\varphi} G_{\text{space}}$ symmetry which has a nontrivial homomorphism $\varphi \colon G_{\text{space}} \to \text{Aut}(G_{\text{sym}})$.

In this paper, we systematically study the gauging of such finite Abelian modulated symmetries. After gauging the modulated $G_{\text{int}}$ symmetry, the dual symmetry group has the general form

$$G_{\text{sym}}^{\vee} = G_{\text{int}}^{\vee} \rtimes_{\varphi^{\vee}} G_{\text{space}}. \tag{1.11}$$

The spatial modulations of the dual symmetry are described by the group homomorphism

$$\varphi^{\vee} \colon G_{\text{space}} \to \text{Aut}(G_{\text{int}}^{\vee}). \tag{1.12}$$

In all of the examples we consider in this paper, and something we expect to be generally true, the dual and original internal symmetry groups are isomorphic to each other: $G_{\text{int}}^{\vee} \simeq G_{\text{int}}$. When $\varphi^{\vee} = \varphi$ and $G_{\text{sym}}^{\vee} \simeq G_{\text{sym}}$, the modulated symmetry is invariant under gauging. However, more generally, the dual symmetry will have a new type of spatial modulation causing $\varphi^{\vee} \neq \varphi$ and $G_{\text{sym}}^{\vee} \not\simeq G_{\text{sym}}$.

We start our study in Section 2 by first considering the case when $N = p$ is a prime integer. In this case, each $U_q$ is a $\mathbb{Z}_p$ symmetry operator, so the internal symmetry group is $G_{\text{int}} = \mathbb{Z}_p^{\times n}$. After warming up with simple examples, we investigate general aspects of gauging these modulated $\mathbb{Z}_p^{\times n}$ symmetries. We first prove that the algebra of local symmetric operators—the so-called bond algebra—is generated by

$$\mathfrak{B} = \left\langle \mathcal{X}_j, \quad \prod_{\ell} \mathcal{Z}_{\ell}^{\Delta_{j,\ell}} \right\rangle, \qquad \sum_{\ell=j}^{j+n} \Delta_{j,\ell} f_{\ell}^{(q)} = 0 \bmod p, \tag{1.13}$$

where $\Delta_{j,\ell}$ is a unique (up to multiplicative factors) $\mathbb{Z}_p$ valued matrix with finite support $n + 1$. We show that functions $f_j^{(q)}$ are highly constrained when $N = p$ is prime to be only a sum of exponential times polynomial functions

$$f_j^{(q)} = \sum_{\alpha} \left( c_{\alpha,1}^{(q)} + c_{\alpha,2}^{(q)} j + \cdots + c_{\alpha,n_{\alpha}}^{(q)} j^{n_{\alpha}-1} \right) x_{\alpha}^j, \tag{1.14}$$

where $c_{\alpha,i}^{(q)}$ and $x_{\alpha}$ are elements in the algebraic extension of $\mathbb{Z}_p$ determined by $\Delta_{j,\ell}$. Furthermore, $f_j^{(q)}$ must be periodic with certain finite periodicity.

Using this, we gauge the modulated $\mathbb{Z}_p^{\times n}$ with the translation invariant Gauss's law

$$G_j = \mathcal{X}_j \prod_{\ell} X_{\ell,\ell+1}^{\Delta_{j,\ell}^{\mathsf{T}}} = 1, \tag{1.15}$$

where $X_{j,j+1}$ acts on newly introduced $\mathbb{Z}_p$ qudits residing on links $\langle j, j + 1 \rangle$ of the lattice. This Gauss's law gives rise to the gauging map

$$\prod_{\ell} \mathcal{Z}_{\ell}^{\Delta_{j,\ell}} \mapsto Z_{j,j+1}^{\dagger}, \qquad \mathcal{X}_j^{\dagger} \mapsto \prod_{\ell} X_{\ell,\ell+1}^{\Delta_{j,\ell}^{\mathsf{T}}}. \tag{1.16}$$

The image of $\mathfrak{B}$ under (1.16) yields the dual bond algebra $\mathfrak{B}^\vee$, and we show that the dual modulated $\mathbb{Z}_p^{\times n}$ symmetry (i.e., the commutant of $\mathfrak{B}^\vee$) is generated by

$$U_q^\vee := \prod_j (Z_{j,j+1})^{f_{-j}^{(q)}} \equiv M \left( \prod_j (Z_{j,j+1})^{f_j^{(q)}} \right) M^{-1}, \tag{1.17}$$

where $M \colon j \to -j$ is the site-centered reflection operator. Therefore, there is a canonical isomorphism $U_q^\vee \simeq M\, U_q\, M^{-1}$ between the modulated $\mathbb{Z}_p^{\times n}$ symmetry and its dual modulated $\mathbb{Z}_p^{\times n}$ symmetry, which implies the existence of an isomorphism

$$\mathfrak{B} \simeq M\, \mathfrak{B}^\vee M^{-1}. \tag{1.18}$$

When the system is reflection symmetric, $M\,\mathfrak{B}^\vee M^{-1} \simeq \mathfrak{B}^\vee$ and the isomorphism (1.18) implies $\mathfrak{B} \simeq \mathfrak{B}^\vee$. In other words, when a translation-invariant Hamiltonian of $\mathbb{Z}_p$ qudits has reflection symmetry, $\varphi^\vee = \varphi$ and $G_{\text{sym}}^\vee \simeq G_{\text{sym}}^\vee$, so the modulated $\mathbb{Z}_p^{\times n}$ symmetry is self dual under gauging.

We then explore the gauging of the modulated symmetry (1.10) with general non-prime $N$ in Section 3. While the gauging procedure used for prime qudits does not apply for general $N$, using techniques from ring theory—the concept of regular matrices over commutative rings (see Appendix A)—we identify a sufficient condition for which the modulated symmetry can be gauged using the Gauss's law (1.15) with $\mathbb{Z}_N$ qudits. In this case, the bond algebra again takes the form (1.13), and there exists the canonical isomorphism (1.18) between $\mathfrak{B}$ and $\mathfrak{B}^\vee$ naturally implemented by the reflection operator $M$.

For the modulated symmetries for which the Gauss's law (1.15) cannot be used, our exploration is primarily done through examples. In particular, we explore various types of polynomial symmetries, which we gauge using a sequential gauging procedure. This is done by gauging $G_{\text{int}}$ one sub-symmetry at a time, prioritizing at each step on gauging subgroups closed under the translation $T$ action. For the modulated symmetries (1.10), this causes one type of "gauge field" to act as the "matter field" of another. To illustrate the idea, let us overview the gauging of the $\mathbb{Z}_N$ quadrupole symmetry discussed in Section 3.2.3. This is a modulated $G_{\text{int}} = \mathbb{Z}_N \times \mathbb{Z}_N \times \mathbb{Z}_{N/\gcd(2,N)}$ symmetry that is generated by the operators

$$U = \prod_j \mathcal{X}_j, \qquad D = \prod_j \mathcal{X}^j, \qquad Q = \prod_j \mathcal{X}^{j^2 - j}, \tag{1.19}$$

which satisfy

$$T\,U\,T^{-1} = U, \qquad T\,D\,T^{-1} = U\,D, \qquad T\,Q\,T^{-1} = D^2\,Q. \tag{1.20}$$

This symmetry can be sequentially gauged in three steps. We first gauge the $\mathbb{Z}_N$ subgroup generated by $U$ since it is closed under $T$. Upon doing so, $U \to 1$ and the $\mathbb{Z}_N$ subgroup generated by $D$ becomes invariant under translation. Therefore, the next subgroup to be gauged is this $\mathbb{Z}_N$ subgroup generated by $D$. With the trivialization $D \to 1$, the remaining $\mathbb{Z}_{N/\gcd(2,N)}$ subgroup generated by $Q$ is no longer spatially modulated and is the last subgroup to be gauged. Among other examples, in Section 3.2.4 we

use the sequential gauging procedure to gauge a $\mathbb{Z}_N$ order-$m$ multipole symmetry, whose symmetry operators are

$$\prod_j (\mathcal{X}_j)^{\sum_{k=0}^{m} c_k \, j^k},\tag{1.21}$$

with $c_k \in \mathbb{Z}$. When $m = 1$ (resp. $m = 2$), this is a $\mathbb{Z}_N$ dipole (resp. quadrupole) symmetry. We prove that this symmetry is self-dual under gauging if and only if $m!$ is coprime to $N$, in which case the sequential gauging processes is unitarily equivalent to gauging using the Gauss's law (1.15). Additionally, we present a modified version of the order-$m$ multipole symmetry that is self-dual under gauging for all $N$ and can always be gauged using (1.15).

As we demonstrate throughout Sections 2 and 3, the isomorphism $\mathfrak{B} \simeq M\,\mathfrak{B}^\vee M^{-1}$ exists for a large class of finite Abelian modulated symmetries and values of $N$. In Section 4, we discuss the implications of this isomorphism in greater detail, relating it to new Kramers-Wannier dualities and a non-invertible reflection operator $\mathsf{D}_\mathrm{M}$. In particular, we consider a generalization of the transverse field Ising model to $\mathbb{Z}_N$ qudits models with modulated symmetries (1.10), where $G_\mathrm{int} = \mathbb{Z}_N^{\times n}$. This generalized Ising model can realize all modulated symmetries discussed in Section 2 when $N = p$ is prime and a large class of modulated symmetries for non-prime $N$. We prove that the bond algebra of its modulated symmetries is always of the form (1.13), and so its modulated symmetries can always be gauged using the Gauss's law (1.15). This generalized Ising model has a self-dual point (i.e., a $J = h$ point) at which its Hamiltonian commutes with $\mathsf{D}_\mathrm{M}$ regardless if it commutes with $M$. This non-invertible reflection operator generates the transformations

$$\mathsf{D}_\mathrm{M} \prod_\ell \mathcal{Z}_\ell^{\Delta_{j,\ell}} = \mathcal{X}_{-j}\,\mathsf{D}_\mathrm{M}, \qquad \mathsf{D}_\mathrm{M}\,\mathcal{X}_j = \prod_\ell \mathcal{Z}_\ell^{-\Delta_{-j,\ell}}\,\mathsf{D}_\mathrm{M},\tag{1.22}$$

and satisfies the fusion algebra

$$\mathsf{D}_\mathrm{M}\,\mathsf{D}_\mathrm{M} = \mathsf{C} \prod_{q=1}^{n} \left(1 + U_q + \cdots + U_q^{N-1}\right), \qquad \mathsf{D}_\mathrm{M}^\dagger = \mathsf{D}_\mathrm{M}\,\mathsf{C},$$
$$U_q\,\mathsf{D}_\mathrm{M} = \mathsf{D}_\mathrm{M}\,U_q = \mathsf{D}_\mathrm{M},\tag{1.23}$$

where $\mathsf{C}$ is charge conjugation $(\mathcal{X}, \mathcal{Z}) \mapsto (\mathcal{X}^\dagger, \mathcal{Z}^\dagger)$. From its fusion algebra, $\mathsf{D}_\mathrm{M}$ is non-invertible since $\mathsf{D}_\mathrm{M}\,\mathsf{D}_\mathrm{M}$ is proportional to a projector. When the generalized Ising model has ordinary reflection symmetries, the modulated symmetry is self-dual under gauging, and the model's Hamiltonian commutes with the canonically defined Kramers-Wannier symmetry operator

$$\mathsf{D}_\mathrm{KW} := T^{-\left\lfloor \frac{n+1}{2} \right\rfloor} M\,\mathsf{D}_\mathrm{M},\tag{1.24}$$

where $\lfloor \cdot \rfloor$ is the floor function. We find that depending on details of $\Delta_{j,\ell}$, $\mathsf{D}_\mathrm{KW}\mathsf{D}_\mathrm{KW}$ sometimes implements a lattice translation $T$ and/or charge conjugation $\mathsf{C}$. Using sequential quantum circuits, we construct explicit expressions for these non-invertible symmetry operators $\mathsf{D}_\mathrm{M}$ and $\mathsf{D}_\mathrm{KW}$ for $\mathbb{Z}_N$ dipole symmetries and $\mathbb{Z}_p$ exponential symmetries.

We end with Section 5, presenting some final remarks and discussing open questions raised by our work.

# 2 Finite Abelian modulated symmetries from prime qudits

This section explores a simple class of modulated symmetries in systems of $\mathbb{Z}_p$ qudits residing on sites $j$ of a one-dimensional infinite spatial chain. We restrict ourselves to qudits for which $p$ is a prime number. The Hilbert space of such models has the tensor product decomposition

$$\mathcal{H} = \bigotimes_j \mathbb{C}[\mathbb{Z}_p], \qquad \mathbb{C}[\mathbb{Z}_p] := \mathbb{C}^p. \tag{2.1}$$

Furthermore, the $\mathbb{Z}_p$ qudit at site $j$ is acted on by the unitary clock and shift operators $\mathcal{Z}_j$ and $\mathcal{X}_j$ obeying

$$\mathcal{Z}_j \, \mathcal{X}_i = \omega_p^{\delta_{ij}} \, \mathcal{X}_i \, \mathcal{Z}_j, \qquad (\mathcal{Z}_j)^p = (\mathcal{X}_j)^p = 1, \tag{2.2}$$

where $\omega_p := e^{2\pi i/p}$ is the $p$-th root of unity. In the $\mathcal{Z}$ eigenbasis, the clock and shift operators act on states $|\cdots, n_{j-1}, n_j, n_{j+1}, \cdots\rangle$, where $n_j \in \{0, 1, \cdots p-1\}$ for all $j$, as

$$\mathcal{Z}_j \, |\cdots, n_{j-1}, n_j, n_{j+1}, \cdots\rangle = \omega_p^{n_j} \, |\cdots, n_{j-1}, n_j, n_{j+1}, \cdots\rangle, \tag{2.3}$$

$$\mathcal{X}_j \, |\cdots, n_{j-1}, n_j, n_{j+1}, \cdots\rangle = |\cdots, n_{j-1}, (n_j + 1 \bmod p), n_{j+1}, \cdots\rangle. \tag{2.4}$$

Models constructed from $\mathbb{Z}_p$ qudits can have finite Abelian modulated symmetries. Here, we consider translation invariant models with finite symmetries generated by the unitary operators

$$U_q = \prod_j (\mathcal{X}_j)^{f_j^{(q)}}, \tag{2.5}$$

where $q = 1, 2, \cdots, n$ labels the finite set of lattice functions $S = \{f^{(1)}, f^{(2)}, \cdots, f^{(n)}\}$ for which each $f_j^{(q)} \in \mathbb{Z}_p$.[5] Since the operators $U_q$ are representations of group generators, we assume the lattice functions $f_j^{(q)}$ are linearly independent over the field $\mathbb{Z}_p$ (i.e., $\sum_q C_q \, f_j^{(q)} \neq 0 \bmod p$ with $C_q \in \mathbb{Z}_p$ unless $C_q = 0$). Therefore, since $p$ is a prime number, the operators $U_q$ generate an internal symmetry described by the group $G_{\text{int}} = \mathbb{Z}_p^{\times n}$ whose elements we denote by $(g^{(1)}, g^{(2)}, \cdots, g^{(n)}) \in G_{\text{int}}$. When $n > 1$, this symmetry always acts in a spatially modulated way, as described by the functions $f_j^{(q)}$. Consequently, the lattice translation symmetry group $G_{\text{space}} = \mathbb{Z}$, generated by $T \colon j \mapsto j+1$, has a nontrivial action on $G_{\text{int}}$ described by

$$T \, U_q \, T^{-1} = \prod_j (\mathcal{X}_j)^{f_{j+1}^{(q)}} \equiv \prod_{a=1}^n (U_a)^{N_a^{(q)}}. \tag{2.6}$$

Denoting by $g^{(q)} = s$ the generator of $G_{\text{int}} = \mathbb{Z}_p^{\times n}$ represented by $U_q$, this defines a group homomorphism $\varphi \colon \mathbb{Z} \to \text{Aut}(\mathbb{Z}_p^{\times n})$ for which (2.6) becomes

$$\varphi_1 \big( \, (1, \cdots, 1, s, 1, \cdots, 1) \, \big) = \left( s^{N_1^{(q)}}, s^{N_2^{(q)}}, \cdots, s^{N_n^{(q)}} \right), \tag{2.7}$$

and the total symmetry group is the semidirect product

$$G_{\text{sym}} = \mathbb{Z}_p^{\times n} \rtimes_\varphi \mathbb{Z}. \tag{2.8}$$

---

[5] In practice, $f_j^{(q)}$ are typically refereed to by functions $\widetilde{f}_j^{(q)} \in \mathbb{Z}$ satisfying $f_j^{(q)} = \widetilde{f}_j^{(q)} \bmod p$.

When the model is invariant under site-centered reflections $M : j \mapsto -j$ as well, the total spatial symmetry group becomes the infinite dihedral group: $G_{\text{space}} = \mathbb{Z} \rtimes \mathbb{Z}_2 \simeq D_\infty$, where the $\mathbb{Z}_2$ action on $\mathbb{Z}$ is $M\,T\,M^{-1} = T^{-1}$. The total symmetry group is then

$$G_{\text{sym}} = \mathbb{Z}_p^{\times n} \rtimes_\varphi D_\infty, \tag{2.9}$$

where $\varphi$ includes the action of translations and site-centered reflections on the internal symmetry group $G_{\text{int}} = \mathbb{Z}_p^{\times n}$.

The rest of this section is dedicated to gauging the $\mathbb{Z}_p^{\times n}$ subgroup of $G_{\text{sym}}$—gauging the modulated symmetry—and finding the symmetry of the gauged model—the dual symmetry $G_{\text{sym}}^\vee$ of $G_{\text{sym}}$. Before gauging the general $G_{\text{int}}$ symmetry generated by (2.5), we consider some simple examples in Section 2.1 to warm up. We then return to the general symmetry (2.5) in Section 2.2.

## 2.1 Some simple examples

### 2.1.1 Exponential symmetries

Let us first consider a model with a modulated $\mathbb{Z}_p$ symmetry generated by

$$U = \prod_j \mathcal{X}_j^{a^j}, \tag{2.10}$$

where $a$ can be any element of $\{1, 2, \cdots, p - 1\}$[6] and $a^{-1}$ is identified with $a^{p-2}$ because by Fermat's little theorem $a^{-1} = a^{p-2} \bmod p$. In what follows, we assume $p > 2$ and $a > 1$ so $U$ generates a modulated symmetry. A simple translation invariant Hamiltonian commuting with this $\mathbb{Z}_p$ symmetry operator is

$$H = -J \sum_j \mathcal{Z}_j^a \mathcal{Z}_{j+1}^\dagger - h \sum_j \mathcal{X}_j + \text{H.c..} \tag{2.11}$$

Under translations by one lattice site, the symmetry operator $U$ transforms as

$$T\,U\,T^{-1} = U^a, \qquad T^{-1}\,U\,T = U^{a^{p-2}}. \tag{2.12}$$

Therefore, the total symmetry of $H$ is described by the group

$$G_{\text{sym}} = \mathbb{Z}_p \rtimes_\varphi \mathbb{Z}, \tag{2.13}$$

where the action of $\mathbb{Z}$ on $\mathbb{Z}_p$ is deduced from (2.12) and described by the group homomorphism $\varphi$. Denoting by $g$ the generator of $\mathbb{Z}_p$, $\varphi$ obeys

$$\varphi_1(g) = g^a, \qquad \varphi_{-1}(g) = g^{a^{p-2}}. \tag{2.14}$$

To gauge the $\mathbb{Z}_p$ modulated symmetry, we introduce new $\mathbb{Z}_p$ qudits onto links $\langle j, j+1 \rangle$ of the lattice that are acted on by the clock and shift operators $Z_{j,j+1}$ and $X_{j,j+1}$. With these additional degrees of freedom, the original Hilbert space (2.1) enlarges to

$$\mathcal{H}^{\text{ext}} = \bigotimes_j \mathbb{C}[\mathbb{Z}_p \times \mathbb{Z}_p], \qquad \mathbb{C}[\mathbb{Z}_p \times \mathbb{Z}_p] := \mathbb{C}^{2p}. \tag{2.15}$$

---

[6]For non-prime $\mathbb{Z}_N$ qudits, $a$ must be coprime to $N$ for the symmetry operator (2.10) to be well defined when $j < 0$.

The Gauss operator $G_j$, that implements the gauging by relating $\mathcal{X}_j$ to the new $\mathbb{Z}_p$ qudits, takes the general form

$$G_j = \mathcal{X}_j \prod_\ell X_{\ell,\ell+1}^{\Delta_{\ell,j}}, \tag{2.16}$$

and is defined such that it satisfies

$$\prod_j \left(G_j\right)^{a^j} = U. \tag{2.17}$$

This definition ensures that the only states that exist in the physical Hilbert space,

$$\mathcal{H}^{\text{phy}} := \mathcal{H}^{\text{ext}}\Big|_{G_j=1}, \tag{2.18}$$

which is the subspace of $\mathcal{H}^{\text{ext}}$ obeying Gauss's law $G_j = 1$, are those in the $U = 1$ symmetric sector. Eq. (2.17) provides a defining constraint on $\Delta_{\ell,j}$ in (2.16). Namely, $\Delta_{\ell,j}$ must satisfy

$$\sum_j \Delta_{\ell,j}\, a^j = 0 \bmod p. \tag{2.19}$$

The simplest translation invariant $\Delta_{\ell,j}$ satisfying this is

$$\Delta_{\ell,j} = a\delta_{\ell,j} - \delta_{\ell,j-1}, \tag{2.20}$$

from which the Gauss operator becomes

$$G_j = X_{j-1,j}^{-1}\, \mathcal{X}_j \left(X_{j,j+1}\right)^a. \tag{2.21}$$

Notice that when $a = 1$, Eq. (2.10) generates a non-modulated symmetry, and this Gauss operator reduces to the ordinary Gauss operator of a $\mathbb{Z}_p$ symmetry.

Gauss's law imposes a redundancy on operators acting on $\mathcal{H}^{\text{phy}}$. Indeed, the unitary operator $\prod_j G_j^{\lambda_j}$ generates the gauge redundancy

$$\mathcal{Z}_j \sim \omega_p^{-\lambda_j} \mathcal{Z}_j, \qquad Z_{j,j+1} \sim \omega_p^{-a\lambda_j + \lambda_{j+1}} Z_{j,j+1}. \tag{2.22}$$

The gauged model's Hamiltonian is found by minimally coupling $Z_{j,j+1}$ to the original model's Hamiltonian (2.11). Doing so, we find the translation invariant Hamiltonian

$$H^\vee = -J \sum_j \mathcal{Z}_j^a Z_{j,j+1}^\dagger \mathcal{Z}_{j+1}^\dagger - h \sum_j \mathcal{X}_j + \text{H.c.}. \tag{2.23}$$

The symmetry operators of $H^\vee$ are nontrivial gauge-invariant operators that commute with $H^\vee$. Certainly, any operator constructed from $Z_{j,j+1}$ commutes with $H^\vee$, but not all such operators are gauge-invariant. Those that are gauge-invariant are generated by

$$U^\vee = \prod_j Z_{j,j+1}^{a^{-j}}. \tag{2.24}$$

Therefore, in agreement with Ref. 97, we find that the dual symmetry of an exponential symmetry is also an exponential symmetry, but with $a$ replaced by $a^{-1} \equiv a^{p-2}$.

Since $U^\vee$ is a $\mathbb{Z}_p$ symmetry operator, like the original model, the internal symmetry of the gauged model is $\mathbb{Z}_p$. However, because their modulated functions differ, the total symmetry group of $H^\vee$ is different from $H$. Indeed, the dual symmetry group is

$$G^\vee_{\text{sym}} = \mathbb{Z}_p \rtimes_{\varphi^\vee} \mathbb{Z}, \tag{2.25}$$

where the group homomorphism $\varphi^\vee$ describes the relation

$$T \, U^\vee \, T^{-1} = (U^\vee)^{a^{p-2}}, \qquad T^{-1} \, U^\vee \, T = (U^\vee)^a. \tag{2.26}$$

When $a \neq a^{p-2} \bmod p$, which is obeyed by all $a \neq \pm 1 \bmod p$, this action of $\mathbb{Z}$ on $\mathbb{Z}_p$ is different from Eq. (2.12), and $\varphi^\vee$ is different from $\varphi$ so the dual symmetry group $G^\vee_{\text{sym}} \not\simeq G_{\text{sym}}$.

Before concluding this example, let us present the gauged model (2.23) in a more convenient basis where the physical Hilbert space $\mathcal{H}^{\text{phy}}$ admits a tensor product decomposition. This does not modify the dual symmetry $G^\vee_{\text{sym}}$ and is equivalent to gauge fixing using the unitary gauge. This basis transformation is implemented using the unitary operator $W$ that satisfies

$$\begin{aligned} W \, \mathcal{X}_j \, W^\dagger = X_{j-1,j} \, \mathcal{X}_j \, X^{-a}_{j,j+1}, & \qquad W \, \mathcal{Z}_j \, W^\dagger = \mathcal{Z}_j, \\ W \, X_{j,j+1} \, W^\dagger = X_{j,j+1}, & \qquad W \, Z_{j,j+1} \, W^\dagger = \mathcal{Z}^a_j \, Z_{j,j+1} \, \mathcal{Z}^\dagger_{j+1}, \end{aligned} \tag{2.27}$$

and has the explicit form

$$W = \prod_j \sum_{\alpha=0}^{p} \mathcal{Z}^\alpha_j P^{(\alpha)}_j, \tag{2.28}$$

where $P^{(\alpha)}_j$ is a projector to the $X_{j-1,j} X^{-a}_{j,j+1} = \omega^\alpha_p$ subspace. After performing this unitary transformation, Gauss's law becomes $G_j = \mathcal{X}_j = 1$, which projects out the qudits on sites. The gauged model's Hamiltonian then becomes

$$H^\vee = -J \sum_j Z_{j,j+1} - h \sum_j X^\dagger_{j-1,j} \, X^a_{j,j+1} + \text{H.c.}, \tag{2.29}$$

and the physical Hilbert space is $\mathcal{H}^{\text{phy}} = \bigotimes_j \mathbb{C}[\mathbb{Z}_p]$ and spanned by the eigenstates of $Z_{j,j+1}$. In this basis, $H^\vee$ is related to $H$ by exchanging the $X$ and $Z$ type operators, exchanging the coupling $J \leftrightarrow h$, and then performing a spatial reflection.

### 2.1.2 Sums of exponentials symmetry and trigonometric symmetry

A more general type of modulated symmetry involving exponential functions is the one generated by

$$U_{\{a_i\}} = \prod_j (\mathcal{X}_j)^{\sum_i c_i \, a^j_i}, \tag{2.30}$$

and the translated operators $T^n U_{\{a_i\}} T^{-n}$ with $n \in \mathbb{Z}$, where $c_i$ and $a_i$ are elements in the algebraic extension of $\mathbb{Z}_p$ satisfying $\sum_i c_i (a_i)^j \in \mathbb{Z}_p$ for all $j$. In some special cases, when $a_i$ are phases, $\sum_i c_i (a_i)^j$ can be written as a sum of trigonometric functions. Here we will consider two examples of such symmetries.

**Example 1:** In the first example, we consider a $\mathbb{Z}_p \times \mathbb{Z}_p$ modulated symmetry generated by

$$U_1 = \prod_j (\mathcal{X}_j)^{(a_+^j - a_-^j)/(a_+ - a_-)}, \qquad U_2 = \prod_j (\mathcal{X}_j)^{(a_+^{j+1} - a_-^{j+1})/(a_+ - a_-)}, \qquad (2.31)$$

where $a_\pm = (\pm\sqrt{5} - 3)/2$. Importantly, $(a_+^j - a_-^j)/(a_+ - a_-) \in \mathbb{Z}$ for all $j$, which is straightforward to verify using the Binomial expansion. While seemingly complicated, it is a symmetry of a fairly simple model

$$H = -J \sum_j \mathcal{Z}_{j-1} \mathcal{Z}_j^3 \mathcal{Z}_{j+1} - h \sum_j \mathcal{X}_j + \text{H.c.}, \qquad (2.32)$$

since $a_\pm^2 + 3a_\pm + 1 = 0$.

In additional to the internal $\mathbb{Z}_p \times \mathbb{Z}_p$ symmetry (2.31), the Hamiltonian (2.32) is invariant under lattice translations $T\colon j \mapsto j+1$ and site-centered reflections $M\colon j \mapsto -j$, making its total spatial symmetry group $G_{\text{space}} = \mathbb{Z} \rtimes \mathbb{Z}_2 \simeq D_\infty$. Since the $\mathbb{Z}_p \times \mathbb{Z}_p$ symmetry (2.31) is modulated, the total symmetry group is

$$G_{\text{sym}} = (\mathbb{Z}_p \times \mathbb{Z}_p) \rtimes_\varphi D_\infty, \qquad (2.33)$$

where the group homomorphism $\varphi\colon D_\infty \to \text{Aut}(\mathbb{Z}_p \times \mathbb{Z}_p)$ captures

$$
\begin{aligned}
T\, U_1\, T^{-1} &= U_2, & T\, U_2\, T^{-1} &= U_1^{p-1}\, U_2^{p-3}, \\
M\, U_1\, M^{-1} &= U_1^{p-1}, & M\, U_2\, M^{-1} &= U_1^3\, U_2.
\end{aligned}
\qquad (2.34)
$$

To gauge this $\mathbb{Z}_p \times \mathbb{Z}_p$ modulated symmetry, we introduce $\mathbb{Z}_p$ qudits onto the links $\langle j, j+1 \rangle$ of the infinite chain that are acted on by the clock and shift operators $Z_{j,j+1}$ and $X_{j,j+1}$. The Gauss operator describing the gauging is

$$G_j = \mathcal{X}_j\, X_{j-1,j}\, X_{j,j+1}^3\, X_{j+1,j+2}, \qquad (2.35)$$

and because

$$\prod_j (G_j)^{(a_+^j - a_-^j)/(a_+ - a_-)} = U_1, \qquad \prod_j (G_j)^{(a_+^{j+1} - a_-^{j+1})/(a_+ - a_-)} = U_2, \qquad (2.36)$$

implementing Gauss's law $G_j = 1$ projects the enlarged Hilbert space into the $U_1 = U_2 = 1$ symmetric subspace. Furthermore, due to Gauss's law, the unitary $\prod_j (G_j)^{\lambda_j}$ generates the gauge redundancy

$$\mathcal{Z}_j \sim \omega_p^{-\lambda_j} \mathcal{Z}_j, \qquad Z_{j,j+1} \sim \omega_p^{-(3\lambda_j + \lambda_{j+1} + \lambda_{j-1})} Z_{j,j+1}. \qquad (2.37)$$

The gauged model's Hamiltonian is found by minimally coupling $Z_{j,j+1}$ in (2.32), which yields

$$H^\vee = -J \sum_j Z_{j,j+1}^\dagger\, \mathcal{Z}_{j-1} \mathcal{Z}_j^3 \mathcal{Z}_{j+1} - h \sum_j \mathcal{X}_j + \text{H.c.}. \qquad (2.38)$$

However, performing the unitary transformation

$$\mathcal{X}_j \mapsto \mathcal{X}_j\, X_{j-1,j}^{-1}\, X_{j,j+1}^{-3}\, X_{j+1,j+2}^{-1}, \qquad Z_{j,j+1} \mapsto Z_{j,j+1}\, \mathcal{Z}_{j-1} \mathcal{Z}_j^3 \mathcal{Z}_{j+1}, \qquad (2.39)$$

rotates the physical Hilbert space into a basis where $\mathcal{X}_j = 1$ and

$$H^\vee = -J \sum_j Z_{j,j+1} - h \sum_j X_{j-1,j}\, X^3_{j,j+1}\, X_{j+1,j+2} + \text{H.c.} \tag{2.40}$$

In this basis, $H^\vee$ is the same as $H$ with the $X$ and $Z$ type operators exchanged and $J \leftrightarrow h$. Therefore, the internal symmetry of the gauged model is also $\mathbb{Z}_p \times \mathbb{Z}_p$ and generated by

$$U_1^\vee = \prod_j (Z_{j,j+1})^{(a_+^j - a_-^j)/(a_+ - a_-)}, \qquad U_2^\vee = \prod_j (Z_{j,j+1})^{(a_+^{j+1} - a_-^{j+1})/(a_+ - a_-)}. \tag{2.41}$$

Since these modulated functions are the same as those in (2.31), the total dual symmetry group $G^\vee_{\text{sym}} \simeq G_{\text{sym}}$, and the $\mathbb{Z}_p \times \mathbb{Z}_p$ modulated symmetry (2.31) is self-dual.

**Example 2:** In the second example, we consider a system of $\mathbb{Z}_p = \mathbb{Z}_2$ qudits (i.e., qubits) with a $\mathbb{Z}_2 \times \mathbb{Z}_2$ modulated symmetry generated by

$$U_1 = \prod_j (\mathcal{X}_j)^{\frac{2}{\sqrt{3}}\sin\left(\frac{2\pi j}{3}\right)}, \qquad U_2 = \prod_j (\mathcal{X}_j)^{\frac{2}{\sqrt{3}}\sin\left(\frac{2\pi(j+1)}{3}\right)}, \tag{2.42}$$

whose exponents are given by the trigonometric functions. Note that $\frac{2}{\sqrt{3}}\sin\left(\frac{2\pi j}{3}\right) \in \{-1, 0, 1\}$ for all $j$. A Hamiltonian symmetric under this symmetry (2.42) is given by

$$H = -J \sum_j \mathcal{Z}_{j-1}\, \mathcal{Z}_j\, \mathcal{Z}_{j+1} - h \sum_j \mathcal{X}_j + \text{H.c..} \tag{2.43}$$

In additional to the internal $\mathbb{Z}_2 \times \mathbb{Z}_2$ symmetry (2.42), the Hamiltonian (2.43) has a spatial symmetry $G_{\text{space}} = \mathbb{Z} \rtimes \mathbb{Z}_2 \simeq D_\infty$ generated by lattice translations $T\colon j \mapsto j+1$ and site-centered reflections $M\colon j \mapsto -j$. The total symmetry group is

$$G_{\text{sym}} = (\mathbb{Z}_p \times \mathbb{Z}_p) \rtimes_\varphi D_\infty, \tag{2.44}$$

where the group homomorphism $\varphi\colon D_\infty \to \text{Aut}(\mathbb{Z}_2 \times \mathbb{Z}_2)$ captures the modulation of the internal $\mathbb{Z}_2 \times \mathbb{Z}_2$ symmetry,

$$\begin{aligned}
T U_1 T^{-1} &= U_2, & T U_2 T^{-1} &= U_1 U_2, \\
M U_1 M^{-1} &= U_1, & M U_2 M^{-1} &= U_1 U_2.
\end{aligned} \tag{2.45}$$

This $\mathbb{Z}_2 \times \mathbb{Z}_2$ modulated symmetry can be gauged using $\mathbb{Z}_2$ qubits with Gauss operator

$$G_j = \mathcal{X}_j\, X_{j-1,j}\, X_{j,j+1}\, X_{j+1,j+2}. \tag{2.46}$$

After implementing Gauss's law $G_j = 1$, the symmetry is trivialized $U_1 = U_2 = 1$ due to the relation

$$\prod_j (G_j)^{\frac{2}{\sqrt{3}}\sin\left(\frac{2\pi j}{3}\right)} = U_1, \qquad \prod_j (G_j)^{\frac{2}{\sqrt{3}}\sin\left(\frac{2\pi j}{3}\right)} = U_2. \tag{2.47}$$

The gauged model's Hamiltonian is found by minimally coupling $Z_{j,j+1}$ in (2.43), which yields

$$H^\vee = -J \sum_j Z^\dagger_{j,j+1}\, \mathcal{Z}_{j-1}\, \mathcal{Z}_j\, \mathcal{Z}_{j+1} - h \sum_j \mathcal{X}_j + \text{H.c..} \tag{2.48}$$

The internal symmetry of the gauged model is also $\mathbb{Z}_2 \times \mathbb{Z}_2$ generated by the gauge invariant operators

$$U_1^\vee = \prod_j (Z_{j,j+1})^{\frac{2}{\sqrt{3}}\sin\left(\frac{2\pi j}{3}\right)}, \qquad U_2^\vee = \prod_j (Z_{j,j+1})^{\frac{2}{\sqrt{3}}\sin\left(\frac{2\pi(j+1)}{3}\right)}. \qquad (2.49)$$

Since these modulated functions are the same as those in (2.42), the total dual symmetry group $G_{\text{sym}}^\vee \simeq G_{\text{sym}}$, and the $\mathbb{Z}_2 \times \mathbb{Z}_2$ modulated symmetry (2.42) is self-dual.

### 2.1.3 Polynomial symmetries

We now consider models with a $\mathbb{Z}_p^{\times m+1}$ modulated symmetry generated by the unitary operators

$$U_n = \prod_j \mathcal{X}_j^{j^n}, \qquad (2.50)$$

where $n = 0, 1, \cdots, m-1, m$. As shown in Appendix B, any translation invariant model that commutes with $\prod_j \mathcal{X}_j^{p(j)}$, where $p(j)$ is an $m$-th order polynomial with integer coefficients, has the $\mathbb{Z}_p^{\times m+1}$ symmetry generated by $U_n$. A simple Hamiltonian that commutes with these symmetry operators is

$$H = -J \sum_j \prod_\ell \mathcal{Z}_\ell^{[\partial^{m+1}]_{j,\ell}} - h \sum_j \mathcal{X}_j + \text{H.c.}, \qquad (2.51)$$

where $\partial$ is the lattice derivative matrix whose matrix elements are $\partial_{j,\ell} = \delta_{j+1,\ell} - \delta_{j,\ell}$, and $\partial^{m+1}$ is the $(m+1)$-th power of the matrix $\partial$. Indeed, because $\sum_\ell [\partial^{m+1}]_{j,\ell}\, \ell^n = 0$ for all $n \in \{0, 1, \cdots, m\}$, all operators $U_n$ commute with the Hamiltonian $H$.

The Hamiltonian $H$ is invariant under lattice translations and reflections, making its spatial symmetry group $G_{\text{space}} = D_\infty$. Since the internal $\mathbb{Z}_p^{\times m+1}$ symmetry is a modulated symmetry, the spatial symmetry group has a nontrivial action on it, and the total symmetry is

$$G_{\text{sym}} = \mathbb{Z}_p^{\times m+1} \rtimes_\varphi D_\infty. \qquad (2.52)$$

The group homomorphism $\varphi \colon D_\infty \to \text{Aut}(\mathbb{Z}_p^{\times m+1})$ describes the action of $D_\infty$ on $\mathbb{Z}_p^{\times m+1}$ arising from

$$T^{\pm 1}\, U_n\, T^{\mp 1} = \prod_{k=0}^n U_k^{\frac{n!(\pm 1)^{n-k}}{k!(n-k)!}}, \qquad M\, U_n\, M^{-1} = U_n^{(-1)^n}. \qquad (2.53)$$

We now gauge the $\mathbb{Z}_p^{\times m+1}$ subgroup of $G_{\text{sym}}$. Because of translation's action on $U_m$, choosing Gauss's law that sets $U_m = 1$ enforces

$$\prod_{k=0}^{m-1} U_k^{\frac{m! x^{m-k}}{k!(m-k)!}} = 1, \qquad (2.54)$$

for any $x \in \mathbb{Z}$, whose only solution is $U_n = 1$ for all $n$. Therefore, to trivialize the $\mathbb{Z}_p^{\times m+1}$ modulated symmetry generated by (2.50), it is sufficient to trivialize the $U_m$ symmetry operator. We do this by introducing $\mathbb{Z}_p$ qudits onto links $\langle j, j+1 \rangle$ of the lattice, which are acted on by the clock and shift operators $Z_{j,j+1}$ and $X_{j,j+1}$, and consider the Gauss operator

$$G_j = \mathcal{X}_j \prod_\ell X_{\ell,\ell+1}^{[\partial^{m+1}]_{\ell,j}}. \qquad (2.55)$$

Since $\partial$ is a lattice derivative, this Gauss operator satisfies

$$\prod_j (G_j)^{j^n} = U_n. \tag{2.56}$$

Therefore, implementing Gauss's law $G_j = 1$ trivializes the entire $\mathbb{Z}_p^{\times m+1}$ symmetry group, and the unitary operator $\prod_j (G_j)^{\lambda_j}$ generates the gauge redundancy

$$\mathcal{Z}_j \sim \omega_p^{-\lambda_j} \mathcal{Z}_j, \qquad Z_{j,j+1} \sim \prod_i \omega_p^{-[\partial^{m+1}]_{j,i}\lambda_i} Z_{j,j+1}. \tag{2.57}$$

In light of this redundancy, we minimally couple the new qudits to $H$ and find the gauged model's Hamiltonian

$$H^\vee = -J \sum_j Z_{j,j+1}^\dagger \prod_\ell \mathcal{Z}_\ell^{[\partial^{m+1}]_{j,\ell}} - h \sum_j \mathcal{X}_j + \text{H.c.}. \tag{2.58}$$

Using Gauss's law and gauge fixing to the unitary gauge, we choose a basis in which the physical Hilbert space is spanned by the $\mathbb{Z}_p$ qudits on links and $H^\vee$ can be written as

$$H^\vee = -J \sum_j Z_{j,j+1} - h \sum_j \prod_\ell X_{\ell,\ell+1}^{[\partial^{m+1}]_{j,\ell}} + \text{H.c.}. \tag{2.59}$$

This Hamiltonian's terms are the same as the original ones with the $X$ and $Z$-type operators exchanged and $J \leftrightarrow h$. Therefore, the symmetries of $H^\vee$ are the same as those of $H$ with its internal symmetries generated by

$$U_n^\vee = \prod_j Z_{j,j+1}^{j^n}, \qquad n \in \{0, 1, \cdots, m-1, m\}, \tag{2.60}$$

so the dual symmetry group $G_{\text{sym}}^\vee$ is isomorphic to the original symmetry group:

$$G_{\text{sym}}^\vee \simeq G_{\text{sym}} = \mathbb{Z}_p^{\times m+1} \rtimes_\varphi D_\infty. \tag{2.61}$$

## 2.2 General case

We now return to the general modulated symmetry

$$G_{\text{sym}} = \mathbb{Z}_p^{\times n} \rtimes_\varphi G_{\text{space}} \tag{2.62}$$

whose internal $\mathbb{Z}_p^{\times n}$ symmetry is generated by the symmetry operators (2.5) defined by a set of linear independent lattice functions $S = \{f^{(1)}, f^{(2)}, \cdots, f^{(n)}\}$ that is closed under translations.

### 2.2.1 The bond algebra

To go beyond working with a particular symmetric Hamiltonian, we can construct and study an algebra of symmetric operators $\mathfrak{B}[S]$ known as the bond algebra [114].[7] All symmetric Hamiltonians

---

[7]Using bond algebras to study the gauging of (non-modulated) symmetries was also done in Refs. 15, 19, in which the bond algebra was called the algebra of local symmetric operators.

are constructed from operators of the bond algebra, so any statements regarding gauging inferred by $\mathfrak{B}[S]$ will apply to all symmetric Hamiltonians. Furthermore, the commutant of $\mathfrak{B}[S]$ is the algebra describing the symmetry operators' fusion rules.

All operators constructed from $\mathcal{X}_j$ are symmetric under the symmetries $U_q$ in (2.5). In contrast, $\mathcal{Z}_j$ transforms nontrivially for generic $f^{(q)}$ as

$$U_q \mathcal{Z}_j U_q^\dagger = \omega_p^{f_j^{(q)}} \mathcal{Z}_j. \tag{2.63}$$

Using $\mathcal{Z}_j$, we can construct a class of local symmetric operators $\prod_\ell \mathcal{Z}_l^{D_{j,\ell}}$ acting on only qudits near the site $j$, and where each $D_{j,\ell}$ obeys

$$\sum_\ell D_{j,\ell} f_\ell^{(q)} = 0 \bmod p, \tag{2.64}$$

for all $f^{(q)} \in S$. Since the set of function $S$ is closed under translations, without a loss of generality, we can choose a translation-invariant $D_{j,\ell}$ that satisfies

$$D_{j,\ell} = D_{j+1,\ell+1} \implies D_{j,\ell} = D_{1,1+\ell-j}. \tag{2.65}$$

As a matrix, this is the statement that the first row of $D_{j,\ell}$ determines all other rows.

For generic $S$, there will be numerous matrices $D_{j,\ell}$ satisfying (2.64). However, we can construct a basis of translation-invariant matrices, denoted as $\{\Delta_{j,\ell}^{(a)} = \Delta_{j+1,\ell+1}^{(a)}\}$, such that any matrix $D_{j,\ell}$ can be decomposed into a weighted sum of these basis matrices and their translated counterparts as

$$D_{j,\ell} = \sum_{a,k} C_k^{(a)} \, \Delta_{j,\ell+k}^{(a)}. \tag{2.66}$$

Each basis matrix $\Delta_{j,\ell}^{(a)}$ should obey the same condition (2.64) as $D_{j,\ell}$, which is given by

$$\sum_\ell \Delta_{j,\ell}^{(a)} f_\ell^{(q)} = 0 \bmod p. \tag{2.67}$$

Generally, each $\Delta^{(a)}$ is a sparse matrix with a finite interaction range $r_a$ encoding locality. Without a loss of generality, we assume that $\Delta_{j,\ell}^{(a)} = 0$ if $\ell < j$ or $\ell > j + r_a - 1$, and $\Delta_{j,j}^{(a)}$, $\Delta_{j,j+r_a-1}^{(a)} \neq 0 \bmod p$.

In general, one might expect that there are multiple $\Delta_{j,\ell}^{(a)}$ in the basis. However, as we now show, for $\mathbb{Z}_p$ qudits with translation invariance, there is always a single $\Delta_{j,\ell}^{(a)}$ from which all $D_{j,\ell}$ can be constructed. To this end, using the finiteness of the interaction range $r_a$ and setting $j = 1$, we simplify the annihilation condition (2.67) to

$$\sum_{\ell=1}^{r_a} \Delta_{1,\ell}^{(a)} f_\ell^{(q)} = 0 \bmod p, \tag{2.68}$$

which can then be expressed as a matrix equation

$$\begin{pmatrix} f_1^{(1)} & f_2^{(1)} & \cdots & f_{r_a}^{(1)} \\ f_1^{(2)} & f_2^{(2)} & \cdots & f_{r_a}^{(2)} \\ \vdots & & & \vdots \\ f_1^{(n)} & f_2^{(n)} & \cdots & f_{r_a}^{(n)} \end{pmatrix} \begin{pmatrix} \Delta_{1,1}^{(a)} \\ \Delta_{1,2}^{(a)} \\ \vdots \\ \Delta_{1,r_a}^{(a)} \end{pmatrix} = \begin{pmatrix} 0 \\ 0 \\ \vdots \\ 0 \end{pmatrix} \quad \bmod p. \tag{2.69}$$

Let $\mathcal{F}_{j,\ell}^{(a)} = f_\ell^{(j)}$ be the $n \times r_a$ matrix appearing in the equation. Then, the basis matrices $\{\Delta_{j,\ell}^{(a)}\}$ with interaction range less than or equal to $r_a$ span the kernel of $\mathcal{F}^{(a)}$. The number of such matrices is given by the dimension of the kernel of $\mathcal{F}^{(a)}$, denoted by nullity$(\mathcal{F}^{(a)})$. Since the matrix elements of $\mathcal{F}^{(a)}$ are elements of a field (i.e., $\mathbb{Z}_p$ with $p$ a prime number), we can use the rank-nullity theorem [115, Theorem 3.2] to relate nullity$(\mathcal{F}^{(a)})$ with rank$(\mathcal{F}^{(a)})$, the dimension of the subspace generated by the rows of $\mathcal{F}^{(a)}$. For $r_a \geq n$, the rank-nullity theorem gives us

$$\text{rank}(\mathcal{F}^{(a)}) + \text{nullity}(\mathcal{F}^{(a)}) = r_a. \tag{2.70}$$

Since each $f^{(q)}$ is assumed to be linearly independent, we have

$$\text{rank}(\mathcal{F}^{(a)}) = n \implies \text{nullity}(\mathcal{F}^{(a)}) = r_a - n. \tag{2.71}$$

From Eq. (2.71) the first nontrivial $r_a \geq n$ solution $\Delta^{(a)}$ exists for interaction range $r_a = n + 1$, which is unique and will simply be denoted by $\Delta$. From Eq. (2.69), its matrix elements are (up to a multiplicative constant)

$$\Delta_{j,\ell} = \begin{cases} 1 & \text{if } \ell = j, \\ -\sum_{k=1}^{n} (\widetilde{\mathcal{F}}^{-1})_{\ell-j,k} \, f_1^{(k)} & \text{if } \ell = j+1, j+1, \cdots, j+n, \\ 0 & \text{if else,} \end{cases} \tag{2.72}$$

where $\widetilde{\mathcal{F}}$ is the $n \times n$ matrix with $\widetilde{\mathcal{F}}_{j,\ell} = f_{\ell+1}^{(j)}$, which is invertible by the assumption that $f^{(q)}$ are linearly independent. For $r_a > n + 1$, there are $r_a - n$ linearly independent solutions to (2.68). However, these linearly independent solutions can all be constructed using the range $r_a = n$ solution $\Delta_{j,\ell}$ by

$$\Delta_{1,j}^{(k_a)} = (\underbrace{0, \cdots, 0}_{k_a \text{ zeros}}, \quad \Delta_{1,1}, \Delta_{1,2}, \cdots, \Delta_{1,1+n}, \quad \underbrace{0, \cdots, 0}_{r_a - k_a - n - 1 \text{ zeros}})^\mathsf{T}, \tag{2.73}$$

where $k_a = 0, 1, \cdots, r_a - n - 1$. For example, the $r_a = n + 2$ solutions to (2.68) form a two-dimension vector space, spanned by the basis vectors $\Delta_{1,j}^{(0)} = (\Delta_{1,j}, 0)^\mathsf{T}$ and $\Delta_{1,j}^{(2)} = (0, \Delta_{1,j-1})^\mathsf{T}$.

From Eq. (2.71), there are no nontrivial interaction range $r_a = n$ solutions. In fact, by translation invariance, this implies that there are no $r_a < n$ solutions. Indeed, the existence of $r_a < n$ solutions would imply the existence of a range $r_a = n$ solution using the construction (2.73). And since by Eq. (2.71) no $r_a = n$ solutions exists, there must also be no nontrivial $r_a < n$ solutions.

The above proves that all symmetric operators $\prod_\ell \mathcal{Z}_\ell^{D_{j,\ell}}$ can be constructed from products of the operators $T^x \prod_\ell \mathcal{Z}_\ell^{\Delta_{j,\ell}} T^{-x}$, $x \in \mathbb{Z}$, that act only on the $n + 1$ sites $j - x, j - x + 1, \cdots, j - x + n$. Therefore, the bond algebra $\mathfrak{B}(S)$ of symmetric operators is generated by

$$\mathfrak{B}[S] := \left\langle \mathcal{X}_j, \quad \prod_\ell \mathcal{Z}_\ell^{\Delta_{j,\ell}} \right\rangle. \tag{2.74}$$

Finding the commutant of $\mathfrak{B}[S]$, and hence the symmetry operators, is equivalent to finding all operators (2.5) such that

$$f^{(q)} \in \ker(\Delta). \tag{2.75}$$

Interestingly, due to translations, this significantly restricts the allowed lattice functions $S$. Indeed, the annihilation condition (2.67) for the generating matrix $\Delta_{j,\ell}$ can be written as

$$\Delta_{1,1}\, f_j^{(q)} + \Delta_{1,2}\, f_{j+1}^{(q)} + \cdots + \Delta_{1,1+n}\, f_{j+n}^{(q)} = 0 \bmod p. \tag{2.76}$$

Since we assume $\Delta_{1,1}, \Delta_{1,1+n} \neq 0 \bmod p$ and that $p$ is a prime number, $\Delta_{1,1}$ and $\Delta_{1,1+n}$ both have multiplicative inverses in $\mathbb{Z}_p$, respectively denoted by $\Delta_{1,1}^{-1}$ and $\Delta_{1,1+n}^{-1}$. Therefore, after specifying $n$ initial values, say the value of $f_j^{(q)}$ at sites $j = 1, \ldots, n$, the lattice functions $f_j^{(q)}$ can be solved recursively using the following two recurrence relations,

$$\begin{aligned}
f_{j+n+1}^{(q)} &= -\Delta_{1,1+n}^{-1}\left(\Delta_{1,1}\, f_{j+1}^{(q)} + \Delta_{1,2}\, f_{j+2}^{(q)} + \cdots + \Delta_{1,n}\, f_{j+n}^{(q)}\right)\ \bmod p, \\
f_j^{(q)} &= -\Delta_{1,1}^{-1}\left(\Delta_{1,2}\, f_{j+1}^{(q)} + \Delta_{1,3}\, f_{j+2}^{(q)} + \cdots + \Delta_{1,1+n}\, f_{j+n}^{(q)}\right)\ \bmod p.
\end{aligned} \tag{2.77}$$

The linearly independent solutions $f_j^{(q)}$ can then be parametrized by their initial conditions, i.e. the value of $f_j^{(q)}$ at sites $j = 1, \ldots, n$, which spans a $n$-dimensional vector space. Note that under translations, such lattice functions are periodic with period at most $p^n$, which is the total number of possible initial conditions to the recurrence relation (2.77).

After specifying the initial conditions, the general solutions to (2.76) are determined by the roots of the characteristic equation [116]

$$\Delta_{1,1} + \Delta_{1,2}\, x + \cdots + \Delta_{1,1+n}\, x^n = 0 \bmod p, \tag{2.78}$$

over the field $\mathbb{Z}_p$, from which

$$f_j^{(q)} = \sum_\alpha \left(c_{\alpha,1}^{(q)} + c_{\alpha,2}^{(q)}\, j + \cdots + c_{\alpha,n_\alpha}^{(q)}\, j^{n_\alpha - 1}\right) x_\alpha^j, \tag{2.79}$$

where $x_\alpha$ is the $\alpha$-th root of (2.78), $n_\alpha$ is its multiplicity, and the coefficients $c_{\alpha,i}^{(q)}$ are determined by the initial conditions $\{f_1^{(q)}, \ldots, f_n^{(q)}\}$. Note that both $x_\alpha$ and $c_{\alpha,i}^{(q)}$ are generally not elements in $\mathbb{Z}_p$ but rather elements in the algebraic extension of $\mathbb{Z}_p$. Therefore, the most general modulated symmetries of a translation invariant Hamiltonian of $\mathbb{Z}_p$ qudits are given by lattice functions of the type (2.79), which are sums of exponential times polynomial functions.

### 2.2.2 Gauging

Having constructed the bond algebra (2.74), we now gauge the $\mathbb{Z}_p^{\times n}$ subgroup generated by (2.5) of the total symmetry group

$$G_{\text{sym}} = \mathbb{Z}_p^{\times n} \rtimes_\varphi G_{\text{space}}. \tag{2.80}$$

Since we always assume lattice translation symmetry, $G_{\text{space}}$ is either $\mathbb{Z} \rtimes \mathbb{Z}_2 \simeq D_\infty$ or $\mathbb{Z}$ depending on if the model has reflection symmetry or not, respectively.

Because $p$ is a prime number and each $\mathbb{Z}_p$ symmetry generator $U_q$ of $\mathbb{Z}_p^{\times n}$ is constructed from only the $\mathcal{X}_j$ operator, the entire symmetry group can be gauged using only one $\mathbb{Z}_p$ qudit. Indeed, we

introduce a $\mathbb{Z}_p$ qudit on each link $\langle j, j+1 \rangle$ that is acted on by the clock and shift operators $Z_{j,j+1}$ and $X_{j,j+1}$, respectively. Doing so enlarges the Hilbert space $\mathcal{H} = \bigotimes_j \mathbb{C}[\mathbb{Z}_p]$ to

$$\mathcal{H}^{\text{ext}} = \bigotimes_j \mathbb{C}[\mathbb{Z}_p \times \mathbb{Z}_p]. \tag{2.81}$$

We now gauge $\mathbb{Z}_p^{\times n}$ by constructing a Gauss operator $G_j$ such that implementing Gauss's law $G_j = 1$ projects states in $\mathcal{H}^{\text{ext}}$ to the symmetric sector of the original Hilbert space. Since the Gauss operator $G_j$ is a product of the $\mathcal{X}_j$ operator and operators acting on the new qudits (i.e., $Z_{\ell,\ell+1}$ and $X_{\ell,\ell+1}$), we require it to obey

$$\prod_{j \in \Lambda} (G_j)^{f_j^{(q)}} = U_q, \tag{2.82}$$

for all $\{f^{(q)}\}$. Indeed, upon enforcing Gauss's law $G_j = 1$, the symmetry operators satisfy $U_q = 1$ for all remaining states in the physical Hilbert space

$$\mathcal{H}^{\text{phy}} := \mathcal{H}^{\text{ext}}\Big|_{G_j=1}. \tag{2.83}$$

A Gauss operator that satisfies (2.82) for a general modulated symmetry (2.5) is[8]

$$G_j := \mathcal{X}_j \prod_\ell X_{\ell,\ell+1}^{\Delta_{j,\ell}^{\mathsf{T}}}, \tag{2.84}$$

with $\Delta_{j,\ell}^{\mathsf{T}} \equiv \Delta_{\ell,j}$. Due to the Gauss's law, the unitary operator $G[\lambda] := \prod_{j \in \Lambda} G_j^{\lambda_j}$, where $\lambda_j \in \mathbb{Z}_p$, generates the gauge redundancy

$$\mathcal{Z}_j \sim \omega_p^{-\lambda_j} \mathcal{Z}_j \qquad Z_{j,j+1} \sim \omega_p^{-\sum_\ell \Delta_{j,\ell} \cdot \lambda_\ell} Z_{j,j+1}. \tag{2.85}$$

Since $\mathcal{Z}_j$ is not gauge-invariant, the bond algebra (2.74) is also no longer gauge-invariant. However, by minimal coupling, we can produce the new gauge-invariant bond algebra

$$\mathfrak{B}^\vee[S] := \left\langle \mathcal{X}_j, \quad Z_{j,j+1}^\dagger \prod_\ell \mathcal{Z}_\ell^{\Delta_{j,\ell}} \right\rangle, \tag{2.86}$$

which includes the operators from which all Hamiltonians of the gauged model are constructed.

It is convenient to gauge fix and choose a basis of $\mathcal{H}_{\text{phys}}$ in which the old and new $\mathbb{Z}_p$ qudits are decoupled. First using Gauss's law to express $\mathcal{X}_j$ in terms of $X_{j,j+1}$ operators, we rewrite the gauged bond algebra as

$$\mathfrak{B}^\vee[S] = \left\langle \prod_\ell X_{\ell,\ell+1}^{-\Delta_{j,\ell}^{\mathsf{T}}}, \quad Z_{j,j+1}^\dagger \prod_\ell \mathcal{Z}_\ell^{\Delta_{j,\ell}} \right\rangle. \tag{2.87}$$

---

[8]We discuss an alternative Gauss operator that is position-dependent and breaks lattice translation symmetry in Appendix C.

We then gauge fix to the unitary gauge using a unitary operator $W$ that rotates the physical Hilbert space to a basis in which[9]

$$W \, \mathcal{X}_j \, W^\dagger = \mathcal{X}_j \prod_\ell X_{\ell,\ell+1}^{-\Delta_{j,\ell}^\mathsf{T}}, \qquad W \, Z_{j,j+1} \, W^\dagger = Z_{j,j+1} \prod_\ell \mathcal{Z}_\ell^{\Delta_{j,\ell}},$$
$$W \, \mathcal{Z}_j \, W^\dagger = \mathcal{Z}_j, \qquad\qquad W \, X_{j,j+1} \, W^\dagger = X_{j,j+1}, \tag{2.88}$$

which causes the gauged bond algebra to become

$$\mathfrak{B}^\vee[S] = \left\langle Z_{j,j+1}, \quad \prod_\ell X_{\ell,\ell+1}^{\Delta_{j,\ell}^\mathsf{T}} \right\rangle. \tag{2.89}$$

In this new basis, Gauss's law becomes $\mathcal{X}_j = 1$. Therefore, implementing Gauss's law is equivalent to projecting out the original $\mathbb{Z}_p$ qudits and treating the new $\mathbb{Z}_p$ qudits as the physical degrees of freedom which make up the tensor product Hilbert space $\mathcal{H}_{\text{phys}} = \bigotimes_j \mathbb{C}[\mathbb{Z}_p]$. Therefore, gauging induces a map between the local symmetric operators of the original and gauged model described by

$$\prod_\ell \mathcal{Z}_\ell^{\Delta_{j,\ell}} \mapsto Z_{j,j+1}^\dagger, \qquad \mathcal{X}_j^\dagger \mapsto \prod_\ell X_{\ell,\ell+1}^{\Delta_{j,\ell}^\mathsf{T}}. \tag{2.90}$$

### 2.2.3 Dual symmetry

Having found the gauged bond algebra $\mathfrak{B}^\vee[S]$, we can now find the dual symmetry group $G_{\text{sym}}^\vee$ from its commutant. In what follows, we will construct the dual symmetry in the basis for which $\mathfrak{B}^\vee[S]$ is given by Eq. (2.89). In this basis, the gauged bond algebra is strikingly similar to the original bond algebra (2.74). Indeed, the commutant of $\mathfrak{B}^\vee[S]$ is generated by modulated operators

$$U_q^\vee = \prod_j Z_{j,j+1}^{f_j^{\vee\,(q)}}, \tag{2.91}$$

where

$$\sum_\ell \Delta_{j,\ell}^\mathsf{T} f_\ell^{\vee\,(q)} = 0 \bmod p. \tag{2.92}$$

So while the modulated functions $f^{(q)}$ of the original internal symmetry span $\ker(\Delta)$, the modulated functions $f^{\vee\,(q)}$ of the dual internal symmetry span $\ker(\Delta^\mathsf{T})$.

Recall that due to translation invariance, which we have assumed throughout, the matrix elements $\Delta_{j,\ell}$ must take the form

$$\Delta_{j,\ell} = \Delta_{1,1+\ell-j}, \tag{2.93}$$

and therefore, its transpose is

$$\Delta_{j,\ell}^\mathsf{T} = \Delta_{1,1+j-\ell} = \Delta_{-j,-\ell}. \tag{2.94}$$

---

[9]The explicit form of the operator $W$ is given by

$$W := \prod_j W_j, \qquad W_j := \sum_{\alpha=0}^{p-1} \mathcal{Z}_j^\alpha P_j^{(\alpha)},$$

where $P_j^{(\alpha)}$ is the projector onto the $\prod_\ell X_{\ell,\ell+1}^{-\Delta_{j,\ell}^\mathsf{T}} = \omega_p^\alpha$ subspace.

Therefore, there is a canonical isomorphism between the original bond algebra $\mathfrak{B}[S]$ and gauged bond algebra $\mathfrak{B}^\vee[S]$ implemented by the site-centered reflection operator $M$ that maps site $j$ to $-j$:

$$\mathfrak{B}^\vee[S] \simeq M\,\mathfrak{B}[S]\,M^{-1}. \tag{2.95}$$

This isomorphism applies to their commutants as well, and therefore $U_q^\vee \simeq MU_q M^{-1}$ which implies that the dual symmetry can be generated by[10]

$$f_j^{\vee\,(q)} = f_{-j}^{(q)}. \tag{2.96}$$

Therefore, the internal part of the dual symmetry is $\mathbb{Z}_p^{\times n}$ and generated by

$$U_q^\vee = \prod_j Z_{j,j+1}^{f_{-j}^{(q)}} = M\left(\prod_j Z_{j,j+1}^{f_j^{(q)}}\right) M^{-1} = M\,U_q\,M^{-1}, \tag{2.97}$$

so the total dual symmetry is described by the group

$$G_{\text{sym}}^\vee = \mathbb{Z}_p^{\times n} \rtimes_{\varphi^\vee} G_{\text{space}}. \tag{2.98}$$

When $G_{\text{space}} = \mathbb{Z}$ and the gauged model is not reflection-symmetric, $\varphi^\vee$ will differ from $\varphi$ and $G_{\text{sym}}^\vee \not\simeq G_{\text{sym}}$. In particular, from Eq. (2.97), the group homomorphisms are related by

$$\varphi_k^\vee(\,\cdot\,) = \varphi_{-k}(\,\cdot\,), \tag{2.99}$$

where $k \in \mathbb{Z}$. However, when $G_{\text{space}} = D_\infty$, because lattice reflections are also a symmetry, the dual symmetry will be isomorphic to the original one: $G_{\text{sym}}^\vee \simeq G_{\text{sym}}$.

# 3 General finite Abelian modulated symmetries

Section 2 explored a specialized class of translation invariant models constructed from $\mathbb{Z}_p$ qudits with $p$ a prime number. The restriction of $p$ to a prime number constrained the internal symmetry to be direct products of $\mathbb{Z}_p$ and introduced simplifications that allowed us to gauge general modulated symmetries and find their dual symmetries. This section explores more general finite modulated symmetries in systems of $\mathbb{Z}_N$ qudits with general non-prime $N$. While $\mathbb{Z}_p$ was a field—each non-zero element has a multiplicative inverse—$\mathbb{Z}_N$ is generally a commutative ring (see Appendix A.1). This introduces numerous number theoretic subtleties, making developing a procedure to gauge general finite Abelian modulated symmetries much more challenging. Because of this, we do not attempt to arrive at the most general understanding. Instead, in addition to discussing tractable general aspects, we focus on examples that emphasize numerous subtleties of gauging general finite Abelian modulated symmetries and subtleties of their dual symmetries.

---

[10]This agrees with the result of Section 2.1.1, where the dual symmetry of $f_j = a^j$ is given by is reflected partner $f_j^\vee = a^{-j}$ (see Eq. (2.24)).

## 3.1 Aspects of a general theory

Let us consider the modulated symmetries generated by

$$U_q = \prod_j \mathcal{X}_j^{f_j^{(q)}}, \tag{3.1}$$

where $\mathcal{X}_j$ acts on the $\mathbb{Z}_N$ qudit at site $j$ and $q = 1, 2, \cdots, n$ label the set of independent $\mathbb{Z}_N$-valued lattice functions $S = \{f^{(1)}, f^{(2)}, \cdots, f^{(n)}\}$. To define what independence means here, we introduce the $n \times r_\mathrm{a}$ matrix

$$\mathcal{F}^{(\mathrm{a})} = \begin{pmatrix} f_1^{(1)} & f_2^{(1)} & \cdots & f_{r_\mathrm{a}}^{(1)} \\ f_1^{(2)} & f_2^{(2)} & \cdots & f_{r_\mathrm{a}}^{(2)} \\ \vdots & & & \vdots \\ f_1^{(n)} & f_2^{(n)} & \cdots & f_{r_\mathrm{a}}^{(n)} \end{pmatrix}. \tag{3.2}$$

When $N = p$ is prime, $\mathcal{F}^{(\mathrm{a})}$ is a matrix over the field $\mathbb{Z}_p$, and we define independence as the condition that $\mathrm{rank}(\mathcal{F}^{(\mathrm{a})}) = n$ for all $r_\mathrm{a} \geq n$. Recall that the rank of a matrix over a field is defined as the dimension of the vector space spanned by the rows of the matrix, so this notion of independence is equivalent to the notion of linear independence. For general $N$, however, $\mathbb{Z}_N$ is not a field but a commutative ring, which causes the notion of vector spaces and the above rank function to no longer apply. For general $N$, the rank function $\mathrm{rank}(\mathcal{F}^{(\mathrm{a})})$ is naturally generalized by the determinantal rank $\rho(\mathcal{F}^{(\mathrm{a})})$, defined as the largest integer $\ell$ such that there exists an $\ell \times \ell$ sub-matrix of $\mathcal{F}^{(\mathrm{a})}$ with non-zero determinant. Therefore, the requirement that the $\mathbb{Z}_N$ functions $\{f^{(1)}, f^{(2)}, \cdots, f^{(n)}\}$ are independent means that $\rho(\mathcal{F}^{(\mathrm{a})}) = n$ for all $r_\mathrm{a} \geq n$.[11]

Since $\mathcal{X}_j$ at different sites commute and $\mathcal{X}_j^N = 1$, the operators $U_q$ generate a finite internal symmetry described by an Abelian group. Recall that an operator $\prod_\ell \mathcal{Z}_\ell^{D_{j,\ell}}$ is symmetric if

$$\sum_\ell D_{j,\ell} f_\ell^{(q)} = 0 \bmod N, \tag{3.3}$$

for all $f^{(q)} \in S$. For a system with $L$ sites, Eq. (3.3) is an ordinary matrix equation when $N$ is prime, with $D$ an $L \times L$ matrix and $|f^{(q)}\rangle$ an $L$-dimensional vector. For general $N$, where the notion of vector spaces no longer apply, (3.3) is most naturally formulated using $R$-modules (See Appendix A.1 for an introduction). In particular, the matrices $D$ are elements in the ring $\mathrm{M}_L(\mathbb{Z}_N)$ of $L \times L$ matrices with entries in $\mathbb{Z}_N$, and $|f^{(q)}\rangle$ are elements in the module $\mathbb{Z}_N^{\times L}$ over $\mathrm{M}_L(\mathbb{Z}_N)$. The condition (3.3) then becomes the requirement that $D \in \mathrm{M}_L(\mathbb{Z}_N)$ is in the annihilator $\mathrm{Ann}_{\mathrm{M}_L(\mathbb{Z}_N)}(S)$ of the subset $S = \{f^{(q)}\} \subset \mathbb{Z}_N^{\times L}$.

We denote by $\{\Delta_{j,\ell}^{(\mathrm{a})}\}$ a basis for all $D_{j,\ell}$ satisfying (3.3), which in terms of $R$-modules are the generators of the ideal $\mathrm{Ann}_{\mathrm{M}_L(\mathbb{Z}_N)}(S)$. As argued in Section 2.2.1, which remains true for all $N$,

---

[11]We emphasize that this notion of independence is not the same as linear independence for modules (in the context of $S$, this is the requirement that $\sum_q a_q f^{(q)} = 0$ implies that $a_q = 0$). To illustrate this distinction, consider $S = \{f_j^{(1)} = 1, f_j^{(2)} = 2j\}$ with $N = 4$. In this case, $f^{(1)}$ and $f^{(2)}$ are linearly dependent since $2f_j^{(2)} = 0 \bmod 4$ but they are independent according to the definition in the main text as $\rho(\mathcal{F}^{(a)}) = 2$ when $r_\mathrm{a} \geq 2$.

locality and translation invariance allows us to recast (3.3) for $\{\Delta^{(\mathrm{a})}\}$ as

$$\mathcal{F}^{(\mathrm{a})} \cdot \Delta^{(\mathrm{a})} = 0 \bmod N, \tag{3.4}$$

where $\mathcal{F}^{(\mathrm{a})}$ is defined in (3.2) and

$$\Delta^{(\mathrm{a})} = \left( \Delta^{(\mathrm{a})}_{1,1}, \ \Delta^{(\mathrm{a})}_{1,2}, \ \cdots, \ \Delta^{(\mathrm{a})}_{1,r_{\mathrm{a}}} \right)^{\mathsf{T}}, \tag{3.5}$$

where $r_{\mathrm{a}}$ is the interaction range of $\Delta^{(\mathrm{a})}$.

When $N$ is prime, we applied the rank-nullity theorem (2.70) to $\mathcal{F}^{(\mathrm{a})}$ in Section 2.2.1 to prove that all matrices $D$ can be constructed from a single $\Delta$ matrix, with minimal interaction range $r = n + 1$. The gauging procedure introduced in Section 2.2.2 assumed and relied on this result. When $N$ is not prime, however, the usual tools from linear algebra cease to apply, so this result may no longer hold, and the matrices $D$ can require multiple $\Delta^{(\mathrm{a})}$ basis matrices. For general $N$, the theory of linear equations over rings is necessary [117], relevant aspects of which are reviewed in Appendix A.2.

Below, we outline a sufficient condition for the existence of a unique $\Delta$ matrix for general $N$. An important concept we will need is the regularity of a matrix. The $n \times r_{\mathrm{a}}$ matrix $\mathcal{F}^{(\mathrm{a})}$ with elements in $\mathbb{Z}_N$ is regular if it has a generalized inverse, which is an $r_{\mathrm{a}} \times n$ matrix $\mathcal{G}^{(\mathrm{a})}$ with elements in $\mathbb{Z}_N$ satisfying

$$\mathcal{F}^{(\mathrm{a})} \cdot \mathcal{G}^{(\mathrm{a})} \cdot \mathcal{F}^{(\mathrm{a})} = \mathcal{F}^{(\mathrm{a})}. \tag{3.6}$$

Notice that when $N$ is prime, the generalized inverse $\mathcal{G}^{(\mathrm{a})}$ always exists, so every $\mathcal{F}^{(\mathrm{a})}$ is regular. When $\mathcal{F}^{(\mathrm{a})}$ is regular, we can construct the matrix

$$\mathcal{M}^{(\mathrm{a})} = I_{r_{\mathrm{a}}} - \mathcal{G}^{(\mathrm{a})} \cdot \mathcal{F}^{(\mathrm{a})}. \tag{3.7}$$

Because of Eq. (3.6), each column vector of $\mathcal{M}^{(\mathrm{a})}$ is in the kernel of $\mathcal{F}^{(\mathrm{a})}$. Furthermore, each vector $\Delta^{(\mathrm{a})}_{1,j}$ in the kernel of $\mathcal{F}^{(\mathrm{a})}$ can be decomposed into a linear combination of the column vectors of $\mathcal{M}^{(\mathrm{a})}$ with coefficient given by the components of $\Delta^{(a)}_{1,j}$, i.e.,

$$\Delta^{(\mathrm{a})}_{1,j} = \sum_i \Delta^{(\mathrm{a})}_{1,i} \, \mathcal{M}^{(\mathrm{a})}_{j,i}. \tag{3.8}$$

Thus, the kernel of $\mathcal{F}^{(\mathrm{a})}$ is the submodule spanned by the column vectors of $\mathcal{M}^{(\mathrm{a})}$, and the dimension of the kernel is given by the determinantal rank of $\mathcal{M}^{(\mathrm{a})}$

$$\mathrm{nullity}(\mathcal{F}^{(\mathrm{a})}) = \rho(\mathcal{M}^{(\mathrm{a})}). \tag{3.9}$$

If the determinantal rank of $\mathcal{M}^{(\mathrm{a})}$ satisfies

$$\rho(\mathcal{M}^{(\mathrm{a})}) = r_{\mathrm{a}} - n, \tag{3.10}$$

we have $\mathrm{nullity}(\mathcal{F}^{(\mathrm{a})}) = r_{\mathrm{a}} - n$ as in Eq. (2.71). Then, following a similar argument below Eq. (2.71) for prime qudits, we conclude there is a unique $\Delta$ with minimal interaction range $n + 1$ and all the generators of the kernel of $\mathcal{F}^{(a)}$ with $r_{\mathrm{a}} > n + 1$ can be constructed from the $r_{\mathrm{a}} = n + 1$ generator.

Given a general $\mathcal{F}^{(a)}$, one can systematically check whether it is regular using a decomposition theorem and Rao-regularity [117, 118], as reviewed in Appendix A.2. However, in practice, it is typically straightforward to directly check for a given $\mathcal{F}^{(a)}$ if there exists a $\mathcal{G}^{(a)}$ satisfying (3.6). For regular matrices $\mathcal{F}^{(a)}$ in which Eq. (3.10) holds, from our previous arguments there exists a unique $\Delta$, with interaction range $n+1$, generating $\mathrm{Ann}_{M_L(\mathbb{Z}_N)}(S)$. Therefore, in this case, the modulated symmetry (3.1) can be gauged using the Gauss operator (2.84) from Section 2.2 using $\mathbb{Z}_N$ qudits. Otherwise, there can exist multiple inequivalent generators $\Delta^{(a)}$ of $\mathrm{Ann}_{M_L(\mathbb{Z}_N)}(S)$, which can cause the gauging procedure from Section 2.2 to fail.

In the rest of this section, instead of further pursuing a general formalism for gauging these modulated symmetries, we explore gauging through examples with regular and non-regular $\mathcal{F}^{(a)}$ matrices. In these examples, there exists a convenient basis for $S = \{f^{(1)}, f^{(2)}, \cdots, f^{(n)}\}$ in which each

$$f_j^{(q)} = m_q \, \widetilde{f}_j^{(q)}, \tag{3.11}$$

where $m_q$ is the largest element of $\{1, 2, \cdots, N-1\}$ such that $\widetilde{f}_j^{(q)} \in \mathbb{Z}$ for all $j$. In this basis, the operators $U_q$ become the minimal generators

$$U_q = \prod_j (\mathcal{X}_j^{m_q})^{\widetilde{f}_j^{(q)}}, \tag{3.12}$$

and each are $\mathbb{Z}_{N/\gcd(m_q,N)}$ symmetry operators. Therefore, the internal symmetry group $G_{\mathrm{int}}$ is the direct products of $\mathbb{Z}_{N/\gcd(m_q,N)}$ and the total symmetry group is

$$G_{\mathrm{sym}} = \left( \mathbb{Z}_{N/\gcd(m_1,N)} \times \mathbb{Z}_{N/\gcd(m_2,N)} \times \cdots \times \mathbb{Z}_{N/\gcd(m_n,N)} \right) \rtimes_\varphi G_{\mathrm{space}}, \tag{3.13}$$

where the group homomorphism $\varphi\colon G_{\mathrm{space}} \to \mathrm{Aut}(G_{\mathrm{int}})$ describes the action of $G_{\mathrm{space}}$ on $U_q$ as determined by the lattice functions $S$.

For general $f_j^{(q)}$ and $N$, the techniques developed in Section 2.2 will no longer apply, and a new formalism is required. Indeed, while using a Gauss operator like the one in Section 2.2 can trivialize all symmetry operators $U_q$, it may "over-gauge" the symmetry by trivializing additional onsite unitary operators that are not symmetry operators. A promising strategy to avoid over-gauging is to sequentially gauge $G_{\mathrm{int}}$ one $\mathbb{Z}_{N/\gcd(m_q,N)}$ subgroup at a time. In particular, one can first gauge subgroups that are closed under the translation group action. After doing so, subgroups that have not been gauged and previously not closed under translations can become closed, which are the subgroups that are gauged next.

Because $G_{\mathrm{int}}$ is a finite Abelian group, we generally expect the dual internal symmetry $G_{\mathrm{int}}^\vee$ to be the Pontryagin dual group $\mathrm{Hom}(G_{\mathrm{int}}, U(1)) \simeq G_{\mathrm{int}}$. Indeed, this is because we expect the dual internal symmetry operators are always able to end on the original symmetry's charged operators and remain gauge invariant (i.e., the symmetry charges become gauge charges after gauging, and the dual symmetry operators are the Wilson lines). Therefore, since the symmetry charges are described by the irreducible representations of $G_{\mathrm{int}}$, they fuse according to the group $\mathrm{Hom}(G_{\mathrm{int}}, U(1)) \simeq G_{\mathrm{int}}$, and the dual internal symmetry operators must as well by gauge invariance. However, the dual group's action $\varphi^\vee$ will generally not be the same as $\varphi$, so the total dual symmetry group

$$G_{\mathrm{sym}}^\vee = \left( \mathbb{Z}_{N/\gcd(m_1,N)} \times \mathbb{Z}_{N/\gcd(m_2,N)} \times \cdots \times \mathbb{Z}_{N/\gcd(m_n,N)} \right) \rtimes_{\varphi^\vee} G_{\mathrm{space}}, \tag{3.14}$$

is generally not isomorphic to $G_{\text{sym}}$.

## 3.2 Examples

We now explore examples of gauging general finite Abelian modulated symmetries using the aforementioned sequential gauging strategy.

### 3.2.1 A $\mathbb{Z}_2 \times \mathbb{Z}_4$ modulated symmetry

In this example, consider a system of $\mathbb{Z}_4$ qudits on an infinite chain whose Hamiltonian commutes with the $G_{\text{int}} = \mathbb{Z}_4 \times \mathbb{Z}_2$ modulated symmetry generated by

$$U_{\mathbb{Z}_4} = \prod_j \mathcal{X}_j, \qquad U_{\mathbb{Z}_2} = \prod_j (\mathcal{X}_j)^{2j}. \tag{3.15}$$

Using these modulated functions, the $\mathcal{F}^{(\text{a})}$ matrices take the form

$$\mathcal{F}^{(\text{a})} = \begin{pmatrix} 1 & 1 & \dots & 1 \\ 2 & 4 & \dots & 2\,r_{\text{a}} \end{pmatrix} \bmod 4. \tag{3.16}$$

If the symmetry (3.15) has only one $\Delta$, it would have range $r = 2 + 1$ and span the kernel $\ker(\mathcal{F}^{(3)})$. However, the kernel of

$$\mathcal{F}^{(3)} = \begin{pmatrix} 1 & 1 & 1 \\ 2 & 4 & 6 \end{pmatrix} \bmod 4 \tag{3.17}$$

is two-dimensional, spanned by

$$\Delta_1^{(3)} = \begin{pmatrix} 2 & -2 & 0 \end{pmatrix}^{\mathsf{T}} \bmod 4, \qquad \Delta_2^{(3)} = \begin{pmatrix} 1 & 0 & -1 \end{pmatrix}^{\mathsf{T}} \bmod 4. \tag{3.18}$$

Therefore, the matrices $\mathcal{F}^{(\text{a})}$ are non-regular, and the gauging procedure from Section 2.2 cannot be used.

The bond algebra of this symmetry is

$$\mathfrak{B} := \left\langle \mathcal{X}_j, \quad (\mathcal{Z}_j \mathcal{Z}_{j+1}^\dagger)^2, \quad \mathcal{Z}_j \mathcal{Z}_{j+2}^\dagger \right\rangle. \tag{3.19}$$

If the Hamiltonian is translation invariant, the total symmetry group is

$$G_{\text{sym}} = (\mathbb{Z}_4 \times \mathbb{Z}_2) \rtimes_\varphi \mathbb{Z}, \tag{3.20}$$

where the group homomorphism $\varphi \colon \mathbb{Z} \to \mathrm{Aut}(\mathbb{Z}_4 \times \mathbb{Z}_2)$ captures the translation symmetry action

$$T\,U_{\mathbb{Z}_4}\,T^{-1} = U_{\mathbb{Z}_4}, \quad T\,U_{\mathbb{Z}_2}\,T^{-1} = U_{\mathbb{Z}_4}^2\,U_{\mathbb{Z}_2}. \tag{3.21}$$

As we now show, this simple example demonstrates how when $N$ is not prime, finite Abelian modulated symmetries are not always self-dual.

We gauge this modulated symmetry sequentially, first gauging the $\mathbb{Z}_4$ subgroup generated by $U_{\mathbb{Z}_4}$, which makes $U_{\mathbb{Z}_2}$ non-modulated, and then gauging the $\mathbb{Z}_2$ symmetry generated by $U_{\mathbb{Z}_2}$. In what follows, we explicitly do each step of the gauging.

To gauge the $\mathbb{Z}_4$ subgroup, we introduce $\mathbb{Z}_4$ qudits onto the links of the lattice that are acted on by $X_{j,j+1}$ and $Z_{j,j+1}$. Since $U_{\mathbb{Z}_4}$ is not modulated, the Gauss operator relating them to the original $\mathbb{Z}_4$ qudit operators is

$$G_j^{(1)} = \mathcal{X}_j \, X_{j-1,j} X_{j,j+1}^\dagger. \tag{3.22}$$

The Gauss's law $G_j^{(1)} = 1$ introduces the gauge redundancy

$$\mathcal{Z}_j^\dagger \sim \mathrm{i}^{\lambda_j^{(1)}} \, \mathcal{Z}_j^\dagger, \qquad\qquad Z_{j,j+1} \sim \mathrm{i}^{\lambda_j^{(1)} - \lambda_{j+1}^{(1)}} \, Z_{j,j+1}, \tag{3.23}$$

which by minimal coupling causes the bond algebra to become

$$\mathfrak{B}_{\text{intermediate}} := \left\langle \mathcal{X}_j, \quad (\mathcal{Z}_j \, Z_{j,j+1} \, \mathcal{Z}_{j+1}^\dagger)^2, \quad \mathcal{Z}_j \, Z_{j,j+1} \, Z_{j+1,j+2} \, \mathcal{Z}_{j+2}^\dagger \right\rangle. \tag{3.24}$$

The nontrivial operators commuting with $\mathfrak{B}_{\text{intermediate}}$ are generated by

$$U_{\mathbb{Z}_4}^\vee = \prod_j Z_{j,j+1}, \qquad\qquad U_{\mathbb{Z}_2} = \prod_j X_{j,j+1}^2. \tag{3.25}$$

They form a $G_{\text{int}}^{\text{intermediate}} = \mathbb{Z}_4 \times \mathbb{Z}_2$ symmetry. Since neither of these operators is modulated, the total symmetry group of a translation invariant model is $G_{\text{sym}}^{\text{intermediate}} = (\mathbb{Z}_4 \times \mathbb{Z}_2) \times \mathbb{Z}$. Importantly, on a length $L$ ring, they satisfy

$$U_{\mathbb{Z}_4}^\vee \, U_{\mathbb{Z}_2} = (-1)^L \, U_{\mathbb{Z}_2} \, U_{\mathbb{Z}_4}^\vee, \tag{3.26}$$

and therefore $G_{\text{sym}}^{\text{intermediate}}$ is realized projectively in systems with an odd number of lattice sites. This is a manifestation of a Lieb-Schultz-Mattis (LSM) anomaly involving the $\mathbb{Z}_4 \times \mathbb{Z}_2$ internal symmetry and translations. A similar relation between LSM anomalies and modulated symmetries was found in Ref. 119.

We next gauge the $\mathbb{Z}_2$ subgroup of the original symmetry using $\mathbb{Z}_2$ qudits that are on the links of the lattice and acted on by $\sigma_{j,j+1}^x$ and $\sigma_{j,j+1}^z$. Since the $\mathbb{Z}_2$ subgroup is generated by $U_{\mathbb{Z}_2}$ the Gauss operator in this step of the gauging is defined as

$$G_j^{(2)} = X_{j,j+1}^2 \, \sigma_{j-1,j}^x \sigma_{j,j+1}^x. \tag{3.27}$$

There are now two Gauss's laws, $G_j^{(1)} = G_j^{(2)} = 1$, which enlarges the gauge redundancy (3.23) to

$$\mathcal{Z}_j^\dagger \sim \mathrm{i}^{\lambda_j^{(1)}} \mathcal{Z}_j^\dagger, \qquad Z_{j,j+1} \sim \mathrm{i}^{\lambda_j^{(1)} - \lambda_{j+1}^{(1)} + 2\lambda_j^{(2)}} Z_{j,j+1}, \qquad \sigma_{j,j+1}^z \sim (-1)^{\lambda_j^{(2)} + \lambda_{j+1}^{(2)}} \sigma_{j,j+1}^z, \tag{3.28}$$

and upon minimal coupling, the intermediate bond algebra becomes

$$\mathfrak{B}^\vee := \left\langle \mathcal{X}_j, \quad (\mathcal{Z}_j Z_{j,j+1} \mathcal{Z}_{j+1}^\dagger)^2, \quad \mathcal{Z}_j Z_{j,j+1} \sigma_{j,j+1}^z Z_{j+1,j+2} \mathcal{Z}_{j+2}^\dagger \right\rangle. \tag{3.29}$$

The nontrivial operators commuting with both $\mathfrak{B}^\vee$ and the Gauss operator $G_j^{(2)}$ are generated by

$$U_{\mathbb{Z}_4}^\vee = \prod_j Z_{j,j+1}(\sigma_{j,j+1}^z)^j, \qquad\qquad U_{\mathbb{Z}_2}^\vee = \prod_j \sigma_{j,j+1}^z, \tag{3.30}$$

and form a $\mathbb{Z}_4 \times \mathbb{Z}_2$ modulated symmetry.

The dual internal symmetry group is $G_{\text{int}}^\vee = \mathbb{Z}_4 \times \mathbb{Z}_2$ and therefore $G_{\text{int}}^\vee \simeq G_{\text{int}}$. The dual symmetry group $G_{\text{sym}}^\vee = (\mathbb{Z}_4 \times \mathbb{Z}_2) \rtimes_{\varphi^\vee} \mathbb{Z} \not\simeq G_{\text{sym}}$ because translations act on the dual symmetry operators as

$$T \, U_{\mathbb{Z}_2}^\vee \, T^{-1} = U_{\mathbb{Z}_2}^\vee, \quad T \, U_{\mathbb{Z}_4}^\vee \, T^{-1} = U_{\mathbb{Z}_4}^\vee \, U_{\mathbb{Z}_2}^\vee. \tag{3.31}$$

Because this differs from (3.21), the dual group homomorphism $\varphi^\vee \neq \varphi$ and $G_{\text{sym}}^\vee \not\simeq G_{\text{sym}}$.

### 3.2.2 $\mathbb{Z}_N$ dipole symmetry

Having covered an example of a finite Abelian modulated symmetry that is not self-dual in systems of $\mathbb{Z}_N$ qudits with $N$ not prime, let's consider an example that is self-dual for all $N$. We consider a $\mathbb{Z}_N$ dipole symmetry, which is a $G_{\text{int}} = \mathbb{Z}_N \times \mathbb{Z}_N$ symmetry generated by

$$U = \prod_j \mathcal{X}_j, \qquad D = \prod_j (\mathcal{X}_j)^j, \tag{3.32}$$

where $\mathcal{X}_j$ is the shift operator acting on the $\mathbb{Z}_N$ qudit at site $j$ of an infinite chain. The $\mathcal{F}^{(\text{a})}$ matrix for the $\mathbb{Z}_N$ dipole symmetry generators (3.32) is the $2 \times r_{\text{a}}$ matrix

$$\mathcal{F}^{(\text{a})} = \begin{pmatrix} 1 & 1 & 1 & \cdots & 1 \\ 1 & 2 & 3 & \cdots & r_{\text{a}} \end{pmatrix} \mod N. \tag{3.33}$$

It is a regular matrix for all $r_{\text{a}} \geq n = 2$ because the $r_{\text{a}} \times 2$ matrix

$$\mathcal{G}^{(\text{a})} = \begin{pmatrix} 2 & -1 \\ -1 & 1 \\ 0 & 0 \\ \vdots & \vdots \\ 0 & 0 \end{pmatrix} \mod N \tag{3.34}$$

is always a generalized inverse. Explicitly computing the $\mathcal{M}^{(\text{a})}$ matrix, as in Eq. (3.7),

$$\mathcal{M}^{(\text{a})} = \begin{pmatrix} 0 & 0 & 1 & 2 & \ldots & r_{\text{a}} - 2 \\ 0 & 0 & -2 & -3 & \ldots & -(r_{\text{a}} - 1) \\ 0 & 0 & 1 & 0 & \ldots & 0 \\ 0 & 0 & 0 & 1 & \ldots & 0 \\ \vdots & \vdots & \vdots & \vdots & \ddots & \vdots \\ 0 & 0 & 0 & 0 & \ldots & 1 \end{pmatrix}, \tag{3.35}$$

we see that besides the two first columns (that are zero), all the other columns are independent of each other, which implies that the determinantal rank $\rho(\mathcal{M}^{(\text{a})}) = r_{\text{a}} - 2$ obeys Eq. (3.10). Therefore, the $\mathbb{Z}_N$ dipole symmetry always has a unique minimal $\Delta$ of range $n + 1 = 3$ that generates all the

symmetric $\mathcal{Z}$-operators in the bond algebra. Indeed, the kernel of $\mathcal{F}^{(a)}$ with $r_a \geq n+1 = 3$ is spanned by the $r_a - 2$ vectors

$$\Delta_{1,j}^{(k_a)} = (\underbrace{0, \cdots, 0}_{k_a \text{ zeros}}, \ 1, \ -2, \ 1, \ \underbrace{0, \cdots, 0}_{r_a - k_a - 3 \text{ zeros}} )^\mathsf{T} \bmod N, \tag{3.36}$$

where $k_a = 0, 1, \cdots, r_a - 3$, and the $\mathbb{Z}_N$ dipole symmetry can be gauged using the techniques from Section 2.2 for all $N$. However, in the rest of this section, we show how to gauge it using the sequential gauging strategy and prove that doing so is unitarily equivalent to the gauging from Section 2.2.

From Eq. (3.36), the bond algebra of the $\mathbb{Z}_N$ dipole symmetry is

$$\mathfrak{B} := \left\langle \mathcal{X}_j, \quad \mathcal{Z}_{j-1} \mathcal{Z}_j^{-2} \mathcal{Z}_{j+1} \right\rangle. \tag{3.37}$$

If the Hamiltonian is translation invariant, the total symmetry group is

$$G_{\text{sym}} = (\mathbb{Z}_N \times \mathbb{Z}_N) \rtimes_\varphi \mathbb{Z}, \tag{3.38}$$

where the group homomorphism $\varphi \colon \mathbb{Z} \to \text{Aut}(\mathbb{Z}_N \times \mathbb{Z}_N)$ captures the translation symmetry action

$$T U T^{-1} = U, \quad T D T^{-1} = U D. \tag{3.39}$$

We sequentially gauge the symmetry using the Gauss operators

$$G_j^{(1)} = \mathcal{X}_j X_{j-1,j}^{(1)} (X_{j,j+1}^{(1)})^\dagger, \qquad G_j^{(2)} = X_{j,j+1}^{(1)} X_{j-1,j}^{(2)} (X_{j,j+1}^{(2)})^\dagger. \tag{3.40}$$

The Gauss's law $G_j^{(1)} = 1$ trivializes $U$ and renders $D$ non-modulated. Subsequently, the Gauss's law $G_j^{(2)} = 1$ further trivializes $D$. These Gauss's laws introduce the redundancy

$$\mathcal{Z}_j^\dagger \sim \omega_N^{\lambda_j^{(1)}} \mathcal{Z}_j^\dagger, \qquad Z_{j,j+1}^{(1)} \sim \omega_N^{\lambda_j^{(1)} - \lambda_{j+1}^{(1)} - \lambda_j^{(2)}} Z_{j,j+1}^{(1)}, \qquad Z_{j,j+1}^{(2)} \sim \omega_N^{\lambda_j^{(2)} - \lambda_{j+1}^{(2)}} Z_{j,j+1}^{(2)}, \tag{3.41}$$

where $\omega_N := \exp[2\pi \mathrm{i}/N]$, and the gauged model's bond algebra is

$$\mathfrak{B}^\vee := \left\langle \mathcal{X}_j, \quad Z_{j-1,j}^{(2)} Z_{j-1,j}^{(1)} (Z_{j,j+1}^{(1)})^\dagger \mathcal{Z}_{j-1} \mathcal{Z}_j^{-2} \mathcal{Z}_{j+1} \right\rangle. \tag{3.42}$$

Its commutant is generated by the gauge-invariant operators

$$U^\vee = \prod_j Z_{j,j+1}^{(1)} (Z_{j,j+1}^{(2)})^j, \qquad D^\vee = \prod_j Z_{j,j+1}^{(2)}. \tag{3.43}$$

These generate a $\mathbb{Z}_N \times \mathbb{Z}_N$ symmetry that obeys

$$T U^\vee T^{-1} = D^\vee U^\vee, \qquad T D^\vee T^{-1} = D^\vee, \tag{3.44}$$

and, therefore, generate a $\mathbb{Z}_N$ dipole symmetry. Hence, the dual symmetry group is isomorphic to the original one, and $\mathbb{Z}_N$ dipole symmetry is self-dual for all $N$.

Recall from Section 2.1.3 that for $\mathbb{Z}_p$ qudits with $p$ prime, $\mathbb{Z}_p$ dipole symmetry can be gauged using a single Gauss operator (2.55) with $m = 1$. The Gauss operators (3.40) can be related to (2.55) using the unitary transformation

$$Z_{j-1,j}^{(2)} \mapsto Z_{j-1,j}^{(2)} (Z_{j-1,j}^{(1)})^\dagger Z_{j,j+1}^{(1)}, \qquad X_{j-1,j}^{(1)} \mapsto X_{j-1,j}^{(1)} X_{j-1,j}^{(2)} (X_{j-2,j-1}^{(2)})^\dagger. \tag{3.45}$$

In this new basis, the Gauss operators (3.40) become

$$G_j^{(1)} = \mathcal{X}_j \, (X_{j-2,j-1}^{(2)})^\dagger \, (X_{j-1,j}^{(2)})^2 \, (X_{j,j+1}^{(2)})^\dagger, \tag{3.46}$$

$$G_j^{(2)} = X_{j,j+1}^{(1)}. \tag{3.47}$$

The Gauss operator $G_j^{(1)}$ is now the same as (2.55) with $m = 1$. Furthermore, from the Gauss's law $G_j^{(2)} = 1$, we can ignore the $\mathbb{Z}_N$ qudits acted on by $X^{(1)}$ and $Z^{(1)}$. Therefore, a $\mathbb{Z}_N$ dipole symmetry, for general $N$, can be gauged using only one type of $\mathbb{Z}_N$ qudits with the Gauss operator (3.46).

### 3.2.3 $\mathbb{Z}_N$ quadrupole symmetry

Having gauged a $\mathbb{Z}_N$ dipole symmetry in the previous example, it is interesting to wonder about a general $\mathbb{Z}_N$ multipole symmetry. However, before doing so in the next example, let us consider a $\mathbb{Z}_N$ quadrupole symmetry in a system of $\mathbb{Z}_N$ qudits. This is a $G_{\mathrm{int}} = \mathbb{Z}_N \times \mathbb{Z}_N \times \mathbb{Z}_{N/\gcd(2,N)}$ symmetry generated by the unitary operators

$$U = \prod_j \mathcal{X}_j, \qquad D = \prod_j \mathcal{X}_j^j, \qquad Q = \prod_j \mathcal{X}_j^{j^2-j}. \tag{3.48}$$

$U$ and $D$ are $\mathbb{Z}_N$ symmetry operators, while $Q$ is a $\mathbb{Z}_{N/\gcd(2,N)}$ symmetry operator because $j^2 - j$ is always an even integer.

If the $\mathcal{F}^{(\mathrm{a})}$ matrices are regular, then a single $\Delta$ with range $r = 3 + 1$ would generate the bond algebra and the symmetry can be gauged using the procedure from Section 2.2. However, the kernel of

$$\mathcal{F}^{(4)} = \begin{pmatrix} 1 & 1 & 1 & 1 \\ 1 & 2 & 3 & 4 \\ 0 & 2 & 6 & 12 \end{pmatrix} \mod N \tag{3.49}$$

is generally two-dimensional, spanned by

$$\Delta_1^{(4)} = \begin{pmatrix} -1 & 3 & -3 & 1 \end{pmatrix} \mod N, \qquad \Delta_2^{(4)} = \begin{pmatrix} \frac{N}{\gcd(2,N)} & 0 & \frac{N}{\gcd(2,N)} & 0 \end{pmatrix} \mod N. \tag{3.50}$$

Therefore, when $N$ is even and $\Delta_2^{(4)}$ nontrivial, all $\mathcal{F}^{(\mathrm{a})}$ are non-regular. For odd $N$, however, $\Delta_2^{(4)}$ is trivial and

$$\mathcal{F}^{(\mathrm{a})} = \begin{pmatrix} 1 & 1 & 1 & 1 & \cdots & 1 \\ 1 & 2 & 3 & 4 & \cdots & r_{\mathrm{a}} \\ 0 & 2 & 6 & 12 & \cdots & r_{\mathrm{a}}(r_{\mathrm{a}} - 1) \end{pmatrix} \mod N \tag{3.51}$$

is regular for all $r_{\mathrm{a}} \geq n = 3$ since

$$\mathcal{G}^{(\mathrm{a})} = \begin{pmatrix} 3 & -2 & 2^{-1} \\ -3 & 3 & -1 \\ 1 & -1 & 2^{-1} \\ 0 & 0 & 0 \\ \vdots & \vdots & \vdots \\ 0 & 0 & 0 \end{pmatrix} \mod N \tag{3.52}$$

is a generalized inverse. Notice that $\mathcal{G}^{(\mathrm{a})}$ is well-defined only for odd $N$, when $2^{-1}$, the $\mathbb{Z}_N$ inverse of 2, exists. In what follows, we gauge the symmetry for general $N$ using sequential gauging and show that when $N$ is odd, gauging this way is unitarily equivalent to the procedure defined in Section 2.2. This is possible because Eq. (3.10) holds, $\rho(\mathcal{M}^{(\mathrm{a})}) = r_\mathrm{a} - 3$, which can be seen from the explicit from

$$\mathcal{M}^{(\mathrm{a})} = \begin{pmatrix} 0 & 0 & 0 & -1 & -3 & \ldots & -2^{-1}(r_\mathrm{a} - 3)(r_\mathrm{a} - 2) \\ 0 & 0 & 0 & 3 & 8 & \ldots & (r_\mathrm{a} - 3)(r_\mathrm{a} - 1) \\ 0 & 0 & 0 & -3 & -6 & \ldots & -2^{-1}(r_\mathrm{a} - 2)(r_\mathrm{a} - 1) \\ 0 & 0 & 0 & 1 & 0 & \ldots & 0 \\ 0 & 0 & 0 & 0 & 1 & \ldots & 0 \\ \vdots & \vdots & \vdots & \vdots & \vdots & \ddots & \vdots \\ 0 & 0 & 0 & 0 & 0 & \ldots & 1 \end{pmatrix}. \tag{3.53}$$

Besides from the first three, all other columns of $\mathcal{M}^{(\mathrm{a})}$ are independent from each other, which implies that $\rho(\mathcal{M}^{(\mathrm{a})}) = r_\mathrm{a} - 3$.

The bond algebra of the $\mathbb{Z}_N$ quadruple symmetry is

$$\mathfrak{B} := \left\langle \mathcal{X}_j, \quad (\mathcal{Z}_j \mathcal{Z}_{j+2})^{N/\gcd(2,N)}, \quad \mathcal{Z}_{j-1}^{-1} \mathcal{Z}_j^3 \mathcal{Z}_{j+1}^{-3} \mathcal{Z}_{j+2} \right\rangle. \tag{3.54}$$

If the Hamiltonian is translation invariant, the total symmetry group is

$$G_\mathrm{sym} = (\mathbb{Z}_N \times \mathbb{Z}_N \times \mathbb{Z}_{N/\gcd(2,N)}) \rtimes_\varphi \mathbb{Z}, \tag{3.55}$$

where the group homomorphism $\varphi \colon \mathbb{Z} \to \mathrm{Aut}(\mathbb{Z}_N \times \mathbb{Z}_N \times \mathbb{Z}_{N/\gcd(2,N)})$ captures the translation symmetry action

$$T U T^{-1} = U, \qquad T D T^{-1} = U D, \qquad T Q T^{-1} = D^2 Q. \tag{3.56}$$

We sequentially gauge this $\mathbb{Z}_N$ quadrupole symmetry using two species of $\mathbb{Z}_N$ qudits on links, acted on by $X^{(1)}_{j,j+1}$ and $X^{(2)}_{j,j+1}$, and a single species of $\mathbb{Z}_{N/\gcd(2,N)}$ qudit on links acted on by $X^{(3)}_{j,j+1}$. The Gauss operators relating these qudits with the original $\mathbb{Z}_N$ qudits are

$$\begin{aligned} G^{(1)}_j &= \mathcal{X}_j \, (X^{(1)}_{j-1,j})^\dagger X^{(1)}_{j,j+1}, \\ G^{(2)}_j &= X^{(1)}_{j-1,j} \, (X^{(2)}_{j-1,j})^\dagger X^{(2)}_{j,j+1}, \\ G^{(3)}_j &= (X^{(2)}_{j-1,j})^{\gcd(2,N)} \, (X^{(3)}_{j-1,j})^\dagger X^{(3)}_{j,j+1}. \end{aligned} \tag{3.57}$$

The Gauss operator $G^{(3)}_j$ has $X^{(2)}_{j-1,j}$ raised to $\gcd(2,N)$ because after setting $G^{(1)}_j = G^{(2)}_j = 1$, $Q = \prod_j (X^{(2)}_{j,j+1})^2$. The Gauss's law $G^{(1)}_j = 1$ trivializes $U$ and makes $D$ a non-modulated symmetry, then $G^{(2)}_j = 1$ trivializes $D$ and makes $Q$ a non-modulated symmetry, and lastly $G^{(3)}_j = 1$ trivializes $Q$. These Gauss's laws induce the gauge redundancy

$$\begin{aligned} \mathcal{Z}_j &\sim (\omega_N)^{-\lambda_j^{(1)}} \mathcal{Z}_j, & Z^{(1)}_{j,j+1} &\sim (\omega_N)^{\lambda_{j+1}^{(1)} - \lambda_j^{(1)} - \lambda_{j+1}^{(2)}} Z^{(1)}_{j,j+1}, \\ Z^{(2)}_{j,j+1} &\sim (\omega_N)^{\lambda_{j+1}^{(2)} - \lambda_j^{(2)} - \gcd(2,N)\lambda_{j+1}^{(3)}} Z^{(2)}_{j,j+1}, & Z^{(3)}_{j,j+1} &\sim (\omega_N)^{\gcd(2,N)[\lambda_{j+1}^{(3)} - \lambda_j^{(3)}]} Z^{(3)}_{j,j+1}. \end{aligned} \tag{3.58}$$

Upon minimal coupling, the gauged bond algebra becomes

$$\mathfrak{B}^\vee := \Big\langle \mathcal{X}_j, \quad (\mathcal{Z}_j \, (Z_{j,j+1}^{(1)})^\dagger \, Z_{j+1,j+2}^{(1)} \, Z_{j+1,j+2}^{(2)} \, \mathcal{Z}_{j+2})^{N/\gcd(2,N)},$$
$$Z_{j,j+1}^{(3)} \, (Z_{j-1,j}^{(2)})^\dagger \, Z_{j,j+1}^{(2)} \, Z_{j-2,j-1}^{(1)} \, (Z_{j-1,j}^{(1)})^{-2} \, Z_{j,j+1}^{(1)} \, \mathcal{Z}_{j-2}^{-1} \, \mathcal{Z}_{j-1}^{3} \, \mathcal{Z}_j^{-3} \, \mathcal{Z}_{j+1} \Big\rangle. \tag{3.59}$$

The gauge invariant operators commuting with $\mathfrak{B}^\vee$ describe a $\mathbb{Z}_N \times \mathbb{Z}_N \times \mathbb{Z}_{N/\gcd(2,N)}$ symmetry, and are generated by

$$U^\vee = \prod_j (Z_{j,j+1}^{(3)})^{j(j-1)/2} \, (Z_{j,j+1}^{(2)})^{-j} \, Z_{j,j+1}^{(1)}, \qquad D^\vee = \prod_j (Z_{j,j+1}^{(3)})^j (Z_{j,j+1}^{(2)})^{-1},$$
$$Q^\vee = \prod_j Z_{j,j+1}^{(3)}. \tag{3.60}$$

$U^\vee$ and $D^\vee$ are both $\mathbb{Z}_N$ symmetry operators, while $Q^\vee$ is a $\mathbb{Z}_{N/\gcd(2,N)}$ symmetry operator. They transform under lattice translations as

$$T \, U^\vee \, T^{-1} = D^\vee \, U^\vee, \qquad T \, D^\vee \, T^{-1} = Q^\vee \, D^\vee, \qquad T \, Q^\vee \, T^{-1} = Q^\vee, \tag{3.61}$$

which is described by the group homomorphism $\varphi^\vee \colon \mathbb{Z} \to \mathrm{Aut}(\mathbb{Z}_N \times \mathbb{Z}_N \times \mathbb{Z}_{N/\gcd(2,N)})$. Therefore, the dual symmetry group is $G_{\mathrm{sym}}^\vee = (\mathbb{Z}_N \times \mathbb{Z}_N \times \mathbb{Z}_{N/\gcd(2,N)}) \rtimes_{\varphi^\vee} \mathbb{Z}$. When $N$ is odd, the symmetry operators are all $\mathbb{Z}_N$ operators and $\varphi^\vee = \varphi$ with the identification

$$Q^\vee \simeq U, \qquad D^\vee \simeq D, \qquad (U^\vee)^2 \simeq Q. \tag{3.62}$$

Therefore, when $N$ is odd, the dual symmetry group $G_{\mathrm{sym}}^\vee$ is isomorphic to $G_{\mathrm{sym}}$. Indeed, when $N$ is odd, the unitary transformations

$$\mathcal{X}_j \mapsto \mathcal{X}_j \, (X_{j-1,j}^{(2)}) \, (X_{j,j+1}^{(2)})^{-2} \, (X_{j+1,j+2}^{(2)}),$$
$$X_{j,j+1}^{(1)} \mapsto X_{j,j+1}^{(1)} \, X_{j,j+1}^{(2)} \, (X_{j+1,j+2}^{(2)})^\dagger,$$
$$Z_{j,j+1}^{(2)} \mapsto Z_{j,j+1}^{(2)} \, Z_{j-1,j}^{(1)} \, (Z_{j,j+1}^{(1)})^\dagger \, \mathcal{Z}_{j-1}^\dagger \, \mathcal{Z}_j^2 \, \mathcal{Z}_{j+1}^\dagger, \tag{3.63}$$

then

$$X_{j-1,j}^{(2)} \mapsto X_{j-1,j}^{(2)} \, X_{j-1,j}^{(3)} \, (X_{j,j+1}^{(3)})^\dagger,$$
$$Z_{j,j+1}^{(3)} \mapsto Z_{j,j+1}^{(3)} \, Z_{j-1,j}^{(2)} \, (Z_{j,j+1}^{(2)})^\dagger, \tag{3.64}$$

cause the Gauss's laws to enforce $\mathcal{X}_j = X_{j-1,j}^{(1)} = X_{j-1,j}^{(2)} = 1$ and the dual bond algebra to become

$$\mathfrak{B}^\vee = \Big\langle X_{j-1,j}^{(3)} \, (X_{j,j+1}^{(3)})^{-3} \, (X_{j+1,j+2}^{(3)})^3 \, (X_{j+2,j+3}^{(3)})^{-1}, \quad Z_{j,j+1}^{(3)} \Big\rangle, \quad (\text{odd } N) \tag{3.65}$$

which is isomorphic to $\mathfrak{B}$ with odd $N$. However, when $N$ is even, $\varphi^\vee \neq \varphi$ and the dual symmetry group is not isomorphic to the original symmetry. Therefore, a $\mathbb{Z}_N$ quadrupole symmetry is self-dual only for odd $N$.

When $N$ is a prime number $p$, as discussed in Section 2.1.3, the $\mathbb{Z}_p$ quadrupole symmetry can be gauged in one-step using $\mathbb{Z}_p$ qudits on links, acted on by $X_{j,j+1}$, and imposing Gauss's law on the physical Hilbert space

$$G_j = \mathcal{X}_j \, (X_{j-1,j})^{-1} \, (X_{j,j+1})^3 \, (X_{j+1,j+2})^{-3} \, X_{j+2,j+3} = 1 \tag{3.66}$$

It is natural to wonder whether this one-step gauging generalizes to the case when $N$ is not prime if the $\mathbb{Z}_p$ qudits on links are replaced by $\mathbb{Z}_N$ qudits. In this case, however, the Gauss's law (3.66) trivializes not just the symmetry generators of the $\mathbb{Z}_N$ quadrupole symmetry

$$U = \prod_j G_j = 1, \qquad D = \prod_j (G_j)^j = 1, \qquad Q = \prod_j (G_j)^{j^2 - j} = 1, \tag{3.67}$$

but also the operator

$$\prod_j \mathcal{X}_j^{j(j-1)/2} = \prod_j (G_j)^{j(j-1)/2} = 1. \tag{3.68}$$

When $N$ is odd, this operator is the symmetry operator $Q^{(N+1)/2}$, but for even $N$ it is not a symmetry operator. Therefore, when $\gcd(2, N) \neq 1$, we over gauge the symmetry, and this one-step gauging does not correctly gauge the $\mathbb{Z}_N$ quadrupole symmetry.

While the $\mathbb{Z}_N$ quadrupole symmetry (3.48) is self-dual for only odd $N$, we can consider a closely related modulated $\mathbb{Z}_N^{\times 3}$ symmetry generated by

$$U = \prod_j \mathcal{X}_j, \qquad D = \prod_j \mathcal{X}_j^j, \qquad \widetilde{Q} = \prod_j \mathcal{X}_j^{j(j-1)/2}. \tag{3.69}$$

The bond algebra for this symmetry is

$$\mathfrak{B} := \left\langle \mathcal{X}_j, \quad \mathcal{Z}_{j-1}^{-1} \, \mathcal{Z}_j^3 \, \mathcal{Z}_{j+1}^{-3} \, \mathcal{Z}_{j+2} \right\rangle. \tag{3.70}$$

When $N$ is odd, the bond algebra is identical to (3.54), and the symmetry reduces to the $\mathbb{Z}_N$ quadrupole symmetry. However, for even $N$, the bond algebra has one less generator compared to (3.54), and the symmetry differs from the $\mathbb{Z}_N$ quadrupole symmetry by the operator (3.68). Unlike the $\mathbb{Z}_N$ quadrupole symmetry, this modulated $\mathbb{Z}_N^{\times 3}$ symmetry can be gauged using the Gauss's law (3.66) for all $N$, and is consequently self-dual for all $N$ as well.

### 3.2.4 $\mathbb{Z}_N$ multipole symmetry

Having gauged $\mathbb{Z}_N$ dipole and quadrupole symmetries, this example discusses a $\mathbb{Z}_N$ order-$m$ multipole symmetry. The symmetry operators of a $\mathbb{Z}_N$ order-$m$ multipole symmetry take the general form

$$\prod_j (\mathcal{X}_j)^{\sum_{k=0}^m c_k j^k}, \tag{3.71}$$

where $c_k \in \mathbb{Z}$. When $m = 1$ (resp. $m = 2$), it is a $\mathbb{Z}_N$ dipole (resp. quadrupole) symmetry. Importantly, (3.71) is not a $\mathbb{Z}_N^{\times m+1}$ symmetry operator for general $N$. Indeed, a set of minimal generators for the symmetry are

$$U_n = \prod_j (\mathcal{X}_j)^{P_j^{(n)}} \qquad P_j^{(n)} = \prod_{k=0}^{n-1} (j - k), \tag{3.72}$$

where $P_j^{(0)} = 1$ and $n = 0, 1, \cdots, m$. These generators are independent because $U_n$ acts as 1 at site $j = 0, 1, \cdots, n-1$ and as $(\mathcal{X}_j)^{n!}$ at site $j = n$. The generator $U_n$ is a $\mathbb{Z}_{N/\gcd(n!,N)}$ operator because $n!$ is always a divisor of $P_j^{(n)}$. Therefore, the internal symmetry group is

$$G_{\text{int}} = \mathbb{Z}_{N/\gcd(0!,N)} \times \mathbb{Z}_{N/\gcd(1!,N)} \times \mathbb{Z}_{N/\gcd(2!,N)} \times \cdots \times \mathbb{Z}_{N/\gcd(m!,N)}. \tag{3.73}$$

If the Hamiltonian is translation invariant, the total symmetry group is $G_{\text{sym}} = G_{\text{int}} \rtimes_\varphi \mathbb{Z}$, where the group homomorphism $\varphi \colon \mathbb{Z} \to \text{Aut}(G_{\text{int}})$ captures the translation symmetry action

$$T \, U_n \, T^{-1} = \begin{cases} U_n & n = 0, \\ U_n \, (U_{n-1})^n & \text{else.} \end{cases} \tag{3.74}$$

We gauge the symmetry sequentially, first trivializing $U_0$, then $U_1$, then $U_2$, *etc.* A key identity we use while gauging is

$$\partial[P_j^{(n)}] \equiv \sum_k \partial_{j,k} P_k^{(n)} = n \, P_j^{(n-1)}, \tag{3.75}$$

where $\partial$ denotes the lattice derivative. Let us start from where we left off in the quadrupole case, using the notation

$$a_n = \gcd(n!, N), \tag{3.76}$$

Since $a_0 = 1$, $a_1 = 1$, and $a_2 = \gcd(2, N)$, the clock operators $X^{(i)}$ $(i = 1, 2, 3)$ act on $\mathbb{Z}_{N/a_{i-1}}$ qudits and the Gauss operators are

$$G_j^{(1)} = \mathcal{X}_j \, (X_{j-1,j}^{(1)})^\dagger \, X_{j,j+1}^{(1)}, \tag{3.77}$$

$$G_j^{(2)} = (X_{j-1,j}^{(1)})^{a_1/a_0} \, (X_{j-1,j}^{(2)})^\dagger \, X_{j,j+1}^{(2)}, \tag{3.78}$$

$$G_j^{(3)} = (X_{j-1,j}^{(2)})^{a_2/a_1} \, (X_{j-1,j}^{(3)})^\dagger \, X_{j,j+1}^{(3)}. \tag{3.79}$$

Implementing the corresponding Gauss's laws causes the symmetry operators (3.72) to become

$$U_n = \prod \left( X_{j,j+1}^{(3)} \right)^{\partial^3[P_j^{(n)}]/a_2}, \tag{3.80}$$

where $U_n = 1$ for $n = 0, 1, 2$ and $U_3$ is a non-modulated operator.

To perform the next step in the sequential gauging procedure, it is important to take into account that $3!$ is a divisor of all $\partial^3[P_j^{(n)}]$, which follows directly from (3.75). Therefore, the next Gauss operator $G_j^{(4)}$ must be

$$G_j^{(4)} = (X_{j-1,j}^{(3)})^{a_3/a_2} \, (X_{j-1,j}^{(4)})^\dagger \, X_{j,j+1}^{(4)}. \tag{3.81}$$

where we used $a_3/a_2 = \gcd(3!/a_2, N/a_2)$. Furthermore, because $(X_{j,j+1}^{(3)})^{N/a_2} = 1$, $X_{j,j+1}^{(4)}$ must act on $\mathbb{Z}_{N/a_3}$ qudits for the Gauss's law $G_j^{(4)} = 1$ to be consistent. The Gauss's law $G_j^{(4)} = 1$ causes the symmetry operators (3.80) to become

$$U_n = \prod \left( X_{j,j+1}^{(4)} \right)^{\partial^4[P_j^{(n)}]/a_3}, \tag{3.82}$$

and now $U_n$ with $n \leq 3$ is trivialized while $U_4$ becomes a non-modulated operator.

Following the same logic as above, since 4! is a divisor of $\partial^4[P_j^{(n)}]$, the sub-symmetry generated by $U_4$ is gauged using the Gauss operator

$$G_j^{(5)} = (X_{j-1,j}^{(4)})^{a_4/a_3} \, (X_{j-1,j}^{(5)})^\dagger \, X_{j,j+1}^{(5)}, \tag{3.83}$$

where $X^{(5)}$ act on $\mathbb{Z}_{N/a_4}$ qudits. From here, the pattern arising from sequential gauging becomes clear. The entire modulated symmetry is gauged using the Gauss operators

$$G_j^{(1)} = \mathcal{X}_j \, (X_{j-1,j}^{(1)})^\dagger \, X_{j,j+1}^{(1)}, \qquad G_j^{(k)} = (X_{j-1,j}^{(k-1)})^{a_{k-1}/a_{k-2}} \, (X_{j-1,j}^{(k)})^\dagger \, X_{j,j+1}^{(k)}, \tag{3.84}$$

where $k = 2, 3, \cdots, m+1$ and $X^{(k)}$ act on $\mathbb{Z}_{N/a_{k-1}}$ qudits. Notice that each $G_j^{(n)}$ is a $\mathbb{Z}_{N/a_{n-1}}$ operator, reflecting how the symmetry group is $G_{\text{int}} = \mathbb{Z}_{N/a_0} \times \mathbb{Z}_{N/a_1} \times \cdots \times \mathbb{Z}_{N/a_m}$ as claimed.

Gauge invariant operators constructed from only $\{Z^{(k)}\}$ are the gauged model's symmetries. They are generated by

$$U_n^\vee = \prod_j \prod_{k=n}^m (Z_{j,j+1}^{(k+1)})^{\widetilde{P}_j^{(k-n)}}, \qquad \widetilde{P}_j^{(n)} = \frac{(-1)^n}{n!} \, P_j^{(n)}, \tag{3.85}$$

where $n = 0, 1, \cdots m$. For clarity, let us expand the expressions of $U_n^\vee$ for some $n$,

$$U_m^\vee = \prod_j Z_{j,j+1}^{(m+1)}, \qquad U_{m-1}^\vee = \prod_j (Z_{j,j+1}^{(m+1)})^{-j} Z_{j,j+1}^{(m)},$$
$$U_{m-2}^\vee = \prod_j (Z_{j,j+1}^{(m+1)})^{j(j-1)/2} (Z_{j,j+1}^{(m)})^{-j} Z_{j,j+1}^{(m-1)}. \tag{3.86}$$

Since each $U_n^\vee$ is a $\mathbb{Z}_{N/a_n}$ symmetry operator, the dual internal symmetry group $G_{\text{int}}^\vee \simeq G_{\text{int}}$. The total dual symmetry group is $G_{\text{sym}}^\vee = G_{\text{int}} \rtimes_{\varphi^\vee} \mathbb{Z}$, where the translation symmetry action on $G_{\text{int}}^\vee$ is described by $\varphi^\vee$ and arises from

$$T U_n^\vee T^{-1} = \begin{cases} U_n^\vee & n = m, \\ U_n^\vee \, (U_{n+1}^\vee)^\dagger & \text{else.} \end{cases} \tag{3.87}$$

For general $N$, this differs from how translations act before gauging, given by (3.74), so $\varphi^\vee \neq \varphi$ and $G_{\text{sym}}^\vee \not\simeq G_{\text{sym}}$. In particular, while the $\mathbb{Z}_{N/a_m}$ operator $U_m$ before gauging was modulated, the $\mathbb{Z}_{N/a_m}$ operator $U_m^\vee$ after gauging is not a modulated operator. However, when $a_m = 1$, which implies $a_n = 1$ for all $n = 0, 1, \cdots, m$, the symmetry operators are all $\mathbb{Z}_N$ operators and $\varphi^\vee = \varphi$ with the identification

$$(U_{m-n}^\vee)^{(-1)^n \, n!} \simeq U_n. \tag{3.88}$$

Therefore, a $\mathbb{Z}_N$ order-$m$ multipole symmetry is self-dual if and only if $N$ is coprime to $m!$. When $N$ is coprime to $m!$, the Gauss operators become

$$G_j^{(1)} = \mathcal{X}_j \, (X_{j-1,j}^{(1)})^\dagger \, X_{j,j+1}^{(1)}, \qquad G_j^{(k)} = X_{j-1,j}^{(k-1)} \, (X_{j-1,j}^{(k)})^\dagger \, X_{j,j+1}^{(k)}, \tag{3.89}$$

and using a sequential unitary transformation, they become

$$G_j^{(1)} = \mathcal{X}_j \, \prod_\ell (X_{\ell-1,\ell}^{(m+1)})^{[\partial^{m+1}]_{\ell,j}}, \qquad G_j^{(k)} = X_{j-1,j}^{(k-1)}. \tag{3.90}$$

Notice that $G_j^{(1)}$ is equivalent to the Gauss operator from the prime qudit case in Section 2.1.3.

Let us end this example by emphasizing that using the Gauss operator $G_j^{(1)}$ in (3.90) to gauge the multipole symmetry for general $N$ is incorrect. Indeed, while implementing the corresponding Gauss's law trivializes the symmetry operators (3.72), it trivializes additional onsite unitary operators that are not a part of the order-$m$ multipole symmetry. In particular, it trivializes the operators

$$\widetilde{U}_n = \prod_j \mathcal{X}_j^{\widetilde{P}_j^{(n)}}, \tag{3.91}$$

where $n = 2, 3, \cdots, m$ and $\widetilde{P}_j^{(n)}$ is given in Eq. (3.85). When $N$ is coprime to $m!$, these operators are all symmetry operators listed in (3.72). When $N$ is not coprime to $m!$, some of these operators will not be symmetries, and, therefore, this Gauss's law over gauges the order-$m$ multipole symmetry for general $N$. One could instead consider the modulated $\mathbb{Z}_N^{\times m+1}$ symmetry generated by $\widetilde{U}_n$ for $n = 0, 1, 2, 3, \cdots, m$. This symmetry is correctly gauged using the Gauss operator $G_j^{(1)}$ in (3.90) and is self-dual for all $N$.

# 4 Kramers-Wannier dualities and non-invertible reflection symmetries

In Section 2, we showed that for translation invariant systems of prime qubits, there is a canonical isomorphism between the bond algebras before and after gauging finite Abelian modulated symmetries implemented by the reflection operator (recall Eq. (2.95)). When the local degrees of freedom are not prime qudits, we demonstrated through examples in Section 3 that this isomorphism can hold but is not guaranteed by lattice translation symmetry. In this section, we study a non-invertible reflection symmetry arising from this canonical isomorphism and the gauging procedures presented in the previous sections. We do so in the context of a generalized Ising model of $\mathbb{Z}_N$ qudits, which has a modulated $\mathbb{Z}_N^{\times n}$ symmetry that always supports the aforementioned isomorphism between bond algebras. After some general exposition on the model's non-invertible reflection symmetry, we specialize to particular modulated symmetries and construct explicit expressions for non-invertible reflection and KW symmetry operators.

## 4.1 Generalized Ising model for modulated symmetries

Here, we consider a class of models that generalizes the transverse field Ising model to systems with finite Abelian modulated symmetries. These generalized Ising models are defined on a one-dimensional lattice with $\mathbb{Z}_N$ qudits on sites $j$, acted on by $\mathcal{X}_j$ and $\mathcal{Z}_j$. Their Hamiltonians are parametrized by a $\mathbb{Z}_N$-valued matrix $\Delta_{j,\ell}$ specifying the $\mathcal{Z}$-type interactions:

$$H := -\sum_j \left\{ J \prod_\ell \mathcal{Z}_\ell^{\Delta_{j,\ell}} + h\, \mathcal{X}_j + \text{H.c.} \right\}. \tag{4.1}$$

We impose translation invariance on the Hamiltonian with periodic boundary conditions $j \sim j + L$, where $L$ is the number of lattice sites. This implies that $\Delta_{j,\ell}$ must satisfy

$$\Delta_{j,\,\ell} = \Delta_{j+k,\ell+k} = \Delta_{1,1+\ell-j}, \tag{4.2}$$

for any integer $k \sim k + L$. To avoid long-range interactions, we assume $\Delta_{j,\ell}$ has a finite support of $n + 1$ sites by choosing, without a loss of generality,

$$\Delta_{j,\ell} = 0 \ \text{if} \ \ell < j \ \text{or} \ \ell > j + n, \qquad \Delta_{j,j}, \ \Delta_{j,j+n} \neq 0 \bmod N. \tag{4.3}$$

These generalized Ising models can have finite Abelian modulated symmetries

$$U_f = \prod_j (\mathcal{X}_j)^{f_j}, \tag{4.4}$$

parametrized by $\mathbb{Z}_N$-valued functions $f_j$. For $U_f$ to commute with the Hamiltonian (4.1), these lattice functions must solve

$$\sum_\ell \Delta_{j,\ell} f_\ell = \Delta_{1,1}\,f_j + \Delta_{1,2}\,f_{j+1} + \cdots \Delta_{1,1+n}\,f_{j+n} = 0 \bmod N, \tag{4.5}$$

where we use translation invariance (4.2) to expand the expression. However, Eq. (4.5) does not have a nontrivial solution for every $\Delta_{j,\ell}$, and when this is the case, it means the Hamiltonian (4.1) does not have the symmetry (4.4).[12] In the rest of this section, we will restrict ourselves to the subclass of Hamiltonians (4.1) for which both $\Delta_{1,1}$ and $\Delta_{1,1+n}$ are coprime to $N$. As we next show, such generalized Ising models always have nontrivial modulated symmetries (4.4).

When $\Delta_{1,1}$ and $\Delta_{1,1+n}$ are coprime to $N$, they have $\mathbb{Z}_N$ multiplicative inverses, respectively denoted by $\Delta_{1,1}^{-1}$ and $\Delta_{1,1+n}^{-1}$. Using these, we can derive from (4.5) the left and right recurrence relations

$$f_{j-1} = -\Delta_{1,1}^{-1}\,(\Delta_{1,2}\,f_j + \Delta_{1,3}\,f_{j+1} + \cdots \Delta_{1,1+n}\,f_{j+n-1}) \bmod N, \tag{4.8}$$

$$f_{j+n} = -\Delta_{1,1+n}^{-1}\,(\Delta_{1,1}\,f_j + \Delta_{1,2}\,f_{j+1} + \cdots \Delta_{1,n}\,f_{j+n-1}) \bmod N. \tag{4.9}$$

Using these, we can solve for $f_j$ recursively after specifying the initial conditions $f_1, f_2, \cdots, f_n$. From the theory of linear recurrence relations, this reveals that there are $n$ linearly independent generating

---

[12] As an example, consider $N = 4$ and $n = 2$ where $\Delta_{1,1} = 2$, $\Delta_{1,2} = 1$, and $\Delta_{1,3} = 2$. In this case, Eq. (4.5) becomes

$$2f_j + f_{j+1} + 2f_{j+2} = 0 \bmod 4, \tag{4.6}$$

to which there are no nontrivial solutions. Indeed, taking mod 2 on both sides of the equation gives $f_{j+1} = 0 \bmod 2$ for all $j$, and thus $f_j = 2g_j$. Substituting $f_j = 2g_j$ back to the equation, we obtain $f_{j+1} = 2g_{j+1} = 0 \bmod 4$. This means that the Hamiltonian,

$$H = -\sum_j \left\{ J\,\mathcal{Z}_j^2 \mathcal{Z}_{j+1} \mathcal{Z}_{j+2}^2 + h\,\mathcal{X}_j + \text{H.c.} \right\}. \tag{4.7}$$

has no symmetries of the form (4.4) when $N = 4$.

solutions to Eq. (4.5), which can be parametrized by their initial conditions. We denote these generating solutions by $f_j^{(q)}$ for $q = 1, 2, \cdots, n$, and without a loss of generality, choose them to satisfy the initial conditions

$$f_j^{(q)} = \delta_{j,q}, \quad \text{for } j = 1, 2, \cdots, n. \tag{4.10}$$

Therefore, the Hamiltonian then has a $\mathbb{Z}_N^{\times n}$ modulated symmetry generated by

$$U_q = \prod_j \mathcal{X}_j^{f_j^{(q)}}. \tag{4.11}$$

Since there are a finite number of initial conditions (4.10), the generating solutions $f_j^{(q)}$ must repeat themselves after a certain shift of $j$. In other words, these modulated symmetries are periodic with a finite periodicity. Because of the periodic boundary conditions, these modulated symmetries are generically explicitly broken unless their periodicity matches with the number of lattice sites $L$ [78,107]. In order to preserve all of the $\mathbb{Z}_N^{\times n}$ symmetry upon enforcing periodic boundary conditions, $L$ must be chosen such that[13]

$$f_{j+L}^{(q)} = f_j^{(q)} \bmod N, \quad \text{for all } q. \tag{4.12}$$

From here on, we will assume that $L$ is always chosen to accommodate these constraints (4.12).

The modulated symmetries can characterize different phases of the generalized Ising model (4.1) through their spontaneous symmetry breaking patterns. When $J = 0$ and $h > 0$, the model has a non-degenerate gapped ground state that preserves all of the modulated symmetries and describes a disordered phase. When $J > 0$ and $h = 0$, the local operators $\mathcal{Z}_j$ commute with the Hamiltonian and obtain non-vanishing expectation values in the ground state subspace. The ground state is then ordered and spontaneously breaks all the modulated symmetry. Since the corresponding expectation value acquired by $\mathcal{Z}_j$ is generally site-dependent, the spatial symmetries are also generically spontaneously broken.[14] Away from these fixed points, the ground states of Hamiltonian (4.1) may go through multiple intermediate phases and phase transitions between ordered and disordered phases. It would be interesting to study the phase diagram of these generalized Ising models, which we leave for future works.

## 4.2 KW dualities as non-invertible reflections

The assumption that $\Delta_{1,1}$ and $\Delta_{1,1+n}$ are coprime to $N$ in the generalized Ising model (4.1) provides a powerful simplification of the modulated $\mathbb{Z}_N^{\times n}$ symmetry's bond algebra. Indeed, let us consider the

---

[13]If $L$ does not satisfy the constraint (4.12), the $\mathbb{Z}_N^{\times n}$ modulated symmetry will be broken down to its subgroup. In this case, we can still make sense of the full symmetry by resorting to the notion of bundle symmetries introduced in [107].

[14]The precise ground state degeneracy is given by GSD $= |G_{\text{int}}|$, which varies with respect to the number of lattice sites $L$. It is given by $N^n$ when $L$ satisfies the constraints (4.12). Otherwise, it is smaller than $N^n$.

$\mathcal{F}$ matrix defined in (3.2) with $r > n$:

$$\mathcal{F} = \begin{pmatrix} f_1^{(1)} & f_2^{(1)} & \dots & f_r^{(1)} \\ f_1^{(2)} & f_2^{(2)} & \dots & f_r^{(2)} \\ \vdots & \vdots & \ddots & \vdots \\ f_1^{(n)} & f_2^{(n)} & \dots & f_r^{(n)} \end{pmatrix} = \begin{pmatrix} 1 & 0 & \dots & 0 & f_{n+1}^{(1)} & \dots & f_r^{(1)} \\ 0 & 1 & \dots & 0 & f_{n+1}^{(2)} & \dots & f_r^{(2)} \\ \vdots & \vdots & \ddots & \vdots & \vdots & \ddots & \vdots \\ 0 & 0 & \dots & 1 & f_{n+1}^{(n)} & \dots & f_r^{(n)} \end{pmatrix}, \tag{4.13}$$

where we used the initial conditions (4.10). This $\mathcal{F}$ matrix is regular in the sense that it has a generalized inverse $\mathcal{G}$ satisfying $\mathcal{F} \cdot \mathcal{G} \cdot \mathcal{F} = \mathcal{F}$. The generalized inverse $\mathcal{G}$ and the $\mathcal{M}$ matrix (see Eq. (3.7)) are given by

$$\mathcal{G} = \begin{pmatrix} 1 & 0 & \dots & 0 \\ 0 & 1 & \dots & 0 \\ \vdots & \vdots & \ddots & \vdots \\ 0 & 0 & \dots & 1 \\ 0 & 0 & \dots & 0 \\ \vdots & \vdots & \ddots & \vdots \\ 0 & 0 & \dots & 0 \end{pmatrix}, \quad \mathcal{M} = \begin{pmatrix} 0 & 0 & \dots & 0 & -f_{n+1}^{(1)} & \dots & -f_r^{(1)} \\ 0 & 0 & \dots & 0 & -f_{n+1}^{(2)} & \dots & -f_r^{(2)} \\ \vdots & \vdots & \ddots & \vdots & \vdots & \ddots & \vdots \\ 0 & 0 & \dots & 0 & -f_{n+1}^{(n)} & \dots & -f_r^{(n)} \\ 0 & 0 & \dots & 0 & 1 & \dots & 0 \\ \vdots & \vdots & \ddots & \vdots & \vdots & \ddots & \vdots \\ 0 & 0 & \dots & 0 & 0 & \dots & 1 \end{pmatrix}. \tag{4.14}$$

Since the $(r-n) \times (r-n)$ dimensional sub-matrix in the bottom right corner of $\mathcal{M}$ is an identity matrix with determinant 1, the determinantal rank $\rho(\mathcal{M})$ is at least $r-n$. Additionally, the determinantal rank $\rho(\mathcal{M})$ can be at most $r-n$ because any $\ell \times \ell$ dimensional sub-matrix of $\mathcal{M}$ with $\ell > r-n$ necessarily has a column of zeros, resulting in a determinant of 0. Therefore, we have $\rho(\mathcal{M}) = r-n$. Consequently, every product of $\mathcal{Z}$ operators that commute with the modulated symmetries are products of $\prod_\ell \mathcal{Z}_\ell^{\Delta_{j,\ell}}$, as proven in Section 3.1. Therefore, the bond algebra is generated by only the terms that appear in the Hamiltonian (4.1):[15]

$$\mathfrak{B} := \left\langle \mathcal{X}_j, \quad \prod_\ell \mathcal{Z}_\ell^{\Delta_{j,\ell}} \right\rangle. \tag{4.17}$$

Since the $\mathcal{Z}$ symmetric terms in the bond algebra (4.17) of the generalized Ising model (4.1) are generated by a single $\Delta_{j,\ell}$, the modulated symmetry (4.11) can be gauged using the procedure from Section 2.2.2 with $\mathbb{Z}_N$ qudits. Furthermore, it implies there is also a canonical isomorphism between

---

[15]If $\Delta_{1,1}$ and $\Delta_{1,1+n}$ are not coprime to $N$, the operator $\prod_\ell \mathcal{Z}_\ell^{\Delta_{j,\ell}}$ in the Hamiltonian (4.1) may not generate all $\mathcal{Z}$-only symmetric operators. For example, consider $N = 4$ and $n = 2$ where $\Delta_{1,1} = 1$, $\Delta_{1,2} = 1$, and $\Delta_{1,3} = 2$. In this case, Eq. (4.5) becomes

$$f_j + f_{j+1} + 2f_{j+2} = 0 \bmod 4, \tag{4.15}$$

whose only nontrivial solutions are $f_j = \text{constant}$. Indeed, taking mod 2 on both sides of the equation gives $f_j + f_{j+1} = 0 \bmod 2$ for all $j$. Then using that $2f_{j+1} + 2f_{j+2} = 0 \bmod 4$, we can simplify the original equation to $f_j = f_{j+1} \bmod 4$, whose solution is $f_j = \text{constant}$. This means that the symmetries of the Hamiltonian

$$H = -\sum_j \left\{ J \mathcal{Z}_j \mathcal{Z}_{j+1} \mathcal{Z}_{j+2}^2 + h \mathcal{X}_j + \text{H.c.} \right\}, \tag{4.16}$$

are generated by $\prod_j \mathcal{X}_j$, whose $\mathcal{Z}$-only symmetric operators are generated by $\mathcal{Z}_j \mathcal{Z}_{j+1}^\dagger$, not $\mathcal{Z}_j \mathcal{Z}_{j+1} \mathcal{Z}_{j+2}^2$.

the bond algebra $\mathfrak{B}$ and the dual bond algebra $\mathfrak{B}^\vee$ implemented through lattice reflections (recall Eq. (2.95)).

The gauging map from $\mathfrak{B}$ to the dual bond algebra $\mathfrak{B}^\vee$ is

$$\prod_\ell \mathcal{Z}_\ell^{\Delta_{j,\ell}} \mapsto Z_{j,j+1}^\dagger, \qquad \mathcal{X}_j^\dagger \mapsto \prod_\ell X_{\ell,\ell+1}^{\Delta_{j,\ell}^\mathsf{T}}. \tag{4.18}$$

This gauging map implies the existence of a non-invertible operator $\widetilde{\mathsf{D}}_{\mathrm{KW}}$ that implements the KW transformation

$$\widetilde{\mathsf{D}}_{\mathrm{KW}} \prod_\ell \mathcal{Z}_\ell^{\Delta_{j,\ell}} = \mathcal{X}_j \widetilde{\mathsf{D}}_{\mathrm{KW}}, \qquad \widetilde{\mathsf{D}}_{\mathrm{KW}} \mathcal{X}_j = \prod_\ell \mathcal{Z}_\ell^{-\Delta_{j,\ell}^\mathsf{T}} \widetilde{\mathsf{D}}_{\mathrm{KW}}. \tag{4.19}$$

Indeed, the KW transformation is related to (4.18) by first shifting the link degrees of freedom to sites, i.e., $(X_{j,j+1}, Z_{j,j+1}) \mapsto (\mathcal{X}_j, \mathcal{Z}_j)$, and then conjugating by an inverse Hadamard operator $\mathfrak{H}_j^{-1} : (\mathcal{X}_j, \mathcal{Z}_j) \mapsto (\mathcal{Z}_j, \mathcal{X}_j^\dagger)$. The operator $\widetilde{\mathsf{D}}_{\mathrm{KW}}$ is non-invertible because it obeys $\widetilde{\mathsf{D}}_{\mathrm{KW}} U_q = \widetilde{\mathsf{D}}_{\mathrm{KW}}$ for all $q$ and, therefore, has a nontrivial kernel spanned by states with nontrivial symmetry charge. On the other hand, because $U_q^\vee \widetilde{\mathsf{D}}_{\mathrm{KW}} = \widetilde{\mathsf{D}}_{\mathrm{KW}}$, $\widetilde{\mathsf{D}}_{\mathrm{KW}}$ has a nontrivial left kernel spanned by states charged under $U_q^\vee = \prod_j \mathcal{X}_{-j}^{f_j^{(q)}}$.

Since the canonical isomorphism between $\mathfrak{B}$ and $\mathfrak{B}^\vee$ is implemented by the reflection operator $M$, the operator $\widetilde{\mathsf{D}}_{\mathrm{KW}}$ commutes with the Hamiltonian (4.1) for $J = h$ only if $M$ also commutes with it. On the other hand, the non-invertible reflection operator

$$\mathsf{D}_{\mathrm{M}} := M \widetilde{\mathsf{D}}_{\mathrm{KW}} \tag{4.20}$$

always commutes with the Hamiltonian when $J = h$, and relates the $J > h$ and $J < h$ portions of the Hamiltonian's phase diagram. $\mathsf{D}_{\mathrm{M}}$ is a non-invertible reflection operator since it implements the non-invertible KW transformation and then the reflection transformation $j \to -j$.[16] Using the KW transformation (4.19), it satisfies

$$\mathsf{D}_{\mathrm{M}} \prod_\ell \mathcal{Z}_\ell^{\Delta_{j,\ell}} = \mathcal{X}_{-j} \mathsf{D}_{\mathrm{M}}, \qquad \mathsf{D}_{\mathrm{M}} \mathcal{X}_j = \prod_\ell \mathcal{Z}_\ell^{-\Delta_{-j,\ell}} \mathsf{D}_{\mathrm{M}}, \tag{4.21}$$

where we used Eq. (2.94) to simplify $\Delta^\mathsf{T}$. From these transformation rules, we find that the non-invertible reflection operator obeys

$$\mathsf{D}_{\mathrm{M}} \mathsf{D}_{\mathrm{M}} = \mathsf{C} \prod_{q=1}^n \sum_{a=0}^{N-1} U_q^a, \qquad \mathsf{D}_{\mathrm{M}}^\dagger = \mathsf{D}_{\mathrm{M}} \mathsf{C}, \tag{4.22}$$

$$U_q \mathsf{D}_{\mathrm{M}} = \mathsf{D}_{\mathrm{M}} U_q = \mathsf{D}_{\mathrm{M}},$$

where $\mathsf{C} = \mathfrak{H}^2$ implements charge conjugation $(\mathcal{X}, \mathcal{Z}) \mapsto (\mathcal{X}^\dagger, \mathcal{Z}^\dagger)$. Interestingly, the non-invertible reflection operator's fusion rules depend only on the internal symmetry group and not on how the symmetry is modulated.

---

[16]Similar non-invertible reflection symmetry has been found in continuum field theories, such as Maxwell's theory of electromagnetism and non-Abelian supersymmetric gauge theories [120].

When the Hamiltonian (4.1) commutes with $M$, it then also commutes with $\widetilde{\mathsf{D}}_{\mathrm{KW}}$ at $J = h$. Indeed, when $H$ is translation and reflection-invariant, $\Delta_{j,\ell}$ must respectively satisfy Eq. (4.2) and $\Delta_{j,-\ell} = \sigma \, \Delta_{-(j+n),\ell}$, where $\sigma = \pm 1$ encodes whether the first term in (4.1) is mapped to itself ($\sigma = 1$) or its Hermitian conjugate ($\sigma = -1$) under reflections. These constraints on $\Delta_{j,\ell}$ can be combined into

$$\Delta_{j,\ell}^{\mathsf{T}} = \sigma \, \Delta_{j-n,\ell}, \tag{4.23}$$

which we use to write the KW transformation (4.19) as

$$\widetilde{\mathsf{D}}_{\mathrm{KW}} \prod_{\ell} \mathcal{Z}_{\ell}^{\Delta_{j,\ell}} = \mathcal{X}_j \, \widetilde{\mathsf{D}}_{\mathrm{KW}}, \qquad \widetilde{\mathsf{D}}_{\mathrm{KW}} \, \mathcal{X}_{j+n} = \prod_{\ell} \mathcal{Z}_{\ell}^{-\sigma \Delta_{j,\ell}} \, \widetilde{\mathsf{D}}_{\mathrm{KW}}. \tag{4.24}$$

In this form, it is clear that $\widetilde{\mathsf{D}}_{\mathrm{KW}}$ commutes with the Hamiltonian when $J = h$. However, the operator $\widetilde{\mathsf{D}}_{\mathrm{KW}}$ is not the only non-invertible symmetry that commutes with the Hamiltonian at $J = h$ since it can be composed with any other symmetry operator (e.g., $\mathsf{D}_{\mathrm{M}}$ is still a symmetry operator). Using this freedom, we canonically define the KW self-duality symmetry to be

$$\mathsf{D}_{\mathrm{KW}} := T^{-\left\lfloor \frac{n+1}{2} \right\rfloor} \, \widetilde{\mathsf{D}}_{\mathrm{KW}}, \tag{4.25}$$

where $\lfloor \cdot \rfloor$ denotes the floor operation. Importantly, this definition differentiates between the number of independent lattice functions $n$ being even or odd. From its transformation of symmetric local operators, we find the KW symmetry operator $\mathsf{D}_{\mathrm{KW}}$ satisfies

$$\begin{aligned}
\mathsf{D}_{\mathrm{KW}} \, \mathsf{D}_{\mathrm{KW}} &= \mathfrak{H}^{1+\sigma} \, T^{(n \bmod 2)} \prod_{q=1}^{n} \sum_{a=0}^{N-1} U_q^a, \\
U_q \, \mathsf{D}_{\mathrm{KW}} &= \mathsf{D}_{\mathrm{KW}} \, U_q = \mathsf{D}_{\mathrm{KW}}, \\
\mathsf{D}_{\mathrm{KW}}^{\dagger} &= \mathsf{D}_{\mathrm{KW}} \, \mathfrak{H}^{1+\sigma} \, (T^{\dagger})^{(n \bmod 2)}.
\end{aligned} \tag{4.26}$$

This algebra depends on $n$ through whether it is even or odd. Indeed, when $n$ is odd, the operator $\mathsf{D}_{\mathrm{KW}}$ squared delivers a translation by one lattice site. For example, when (4.1) is the usual transverse field Ising model ($n = 1$), this recovers the known fusion rules of the KW duality symmetry of non-modulated finite Abelian symmetries [42–44, 48]. In this case, the corresponding continuum KW symmetry will emanate from the lattice translations [43, 121]. However, when $n$ is even, the $\mathsf{D}_{\mathrm{KW}}$ symmetry operator does not mix with spatial symmetries and is an internal non-invertible symmetry. Such $n$ only arise when the generalized Ising model (4.1) has modulated symmetries. This agrees with the fusion algebras of the $\mathbb{Z}_2$ dipole symmetry KW operator constructed in Refs. 46, 47, 52 and the $\mathbb{Z}_N$ dipole symmetry KW operator from Ref. 37.

The fusion algebra of the $\mathsf{D}_{\mathrm{KW}}$ operator also depends on the modulation of the symmetry through $\sigma$. Indeed, when $\sigma = -1$, $(\mathsf{D}_{\mathrm{KW}})^2$ does not act by charge conjugation $\mathsf{C} = \mathfrak{H}^2$. This is the case for the uniform $\mathbb{Z}_N$ symmetry, where $\prod_{\ell} \mathcal{Z}_{\ell}^{\Delta_{j,\ell}} = \mathcal{Z}_j^{\dagger} \, \mathcal{Z}_{j+1}$ transforms as $M : \mathcal{Z}_j^{\dagger} \, \mathcal{Z}_{j+1} \to (\mathcal{Z}_{-(j+1)}^{\dagger} \, \mathcal{Z}_{-j})^{\dagger}$ and so $\sigma = -1$. However, when the symmetry of the generalized Ising model is modulated, $\sigma$ can be $+1$. In this case, $(\mathsf{D}_{\mathrm{KW}})^2$ acts by charge conjugation $\mathsf{C} = \mathfrak{H}^2$, which is nontrivial for $N > 2$.

In summary, the fusion algebra (4.26) of the lattice $\mathsf{D}_{\mathrm{KW}}$ symmetry operator has two possible fingerprints of the modulated symmetry. Firstly, it can differ from a uniform abelian symmetry by

$\mathsf{D}_{\mathrm{KW}}$ being purely internal while local and $\mathsf{D}_{\mathrm{KW}}\mathsf{D}_{\mathrm{KW}}$ not performing a lattice translation. Secondly, $\mathsf{D}_{\mathrm{KW}}\mathsf{D}_{\mathrm{KW}}$ can also act by charge conjugation $\mathsf{C} = \mathfrak{H}^2$. These two signatures reflect how the $\mathsf{D}_{\mathrm{KW}}$ operator can be constructed by gauging a modulated symmetry.

## 4.3 Constructing the KW operator

So far, our discussion relied on the transformations implemented by non-invertible operators $\widetilde{\mathsf{D}}_{\mathrm{KW}}$, $\mathsf{D}_{\mathrm{M}}$, and $\mathsf{D}_{\mathrm{KW}}$, but what are the explicit forms for these non-invertible operators? For uniform symmetries, i.e., constant function $f_j = 1$, it has been shown in Refs. [42, 43, 45, 48] that the corresponding non-invertible operators implementing the self-duality symmetry on finite chains (with periodic boundary conditions) can be constructed as products of local unitary operators – the so-called sequential circuits – multiplied by appropriate projectors onto symmetric subspaces. Importantly, the Hamiltonian (4.1) for an infinite chain acts on an infinite dimensional Hilbert space. Hence, a sequential circuit implementing the KW duality would consist of products of infinitely many unitaries, which might be potentially ill-defined. In what follows, we make the so-far general discussion more explicit by first constructing a naïve $\widetilde{\mathsf{D}}_{\mathrm{KW}}$ operator on an infinite chain for any modulated symmetry. We will then specialize to a finite ring of length $L$ with three example modulated symmetries.

### 4.3.1 Construction on infinite chain

Recall that in Section 4.1, we proved that in the generalized Ising model (4.1), when $\Delta_{j,j}$ and $\Delta_{j,j+n}$ are coprime to $N$, all symmetric operators written only using $\mathcal{Z}$ can be constructed by taking translated products of $\prod_l \mathcal{Z}_\ell^{\Delta_{j,\ell}}$. Without loss of generality, we can choose the elements of $\Delta_{j,\ell}$ to be

$$\Delta_{j,j} = 1, \quad \Delta_{j,j+1} = g_1, \quad \Delta_{j,j+2} = g_2, \quad \cdots, \quad \Delta_{j,j+n-1} = g_{n-1}, \quad \Delta_{j,j+n} = g_n, \qquad (4.27)$$

where $g_\ell$ with $\ell = 1, \cdots, n$ are $\mathbb{Z}_N$-valued parameters, whose explicit form when $N$ is prime is given by Eq. (2.72). With this choice, the Hamiltonian (4.1) becomes

$$H = -\sum_j \left\{ J \, \mathcal{Z}_j \, \mathcal{Z}_{j+1}^{g_1} \cdots \mathcal{Z}_{j+n-1}^{g_{n-1}} \, \mathcal{Z}_{j+n}^{g_n} + h \, \mathcal{X}_j + \text{H.c.} \right\}. \qquad (4.28)$$

Let us now consider the sequential product of infinitely many unitary operators

$$\widetilde{\mathsf{D}}_{\mathrm{KW}} := \cdots \mathfrak{H}_{j+1} \, \mathrm{CZ}_{j+1} \, \mathfrak{H}_j \, \mathrm{CZ}_j \, \mathfrak{H}_{j-1} \, \mathrm{CZ}_{j-1} \cdots, \qquad (4.29)$$

where $\mathfrak{H}_j$ is the Hadamard operator and $\mathrm{CZ}_j$ is a modified controlled Z operator with $\Delta$ dependence, (see Appendix D for their explicit definitions). We define the Hadamard operator to act as $\mathfrak{H}_j : (\mathcal{X}_j, \mathcal{Z}_j) \mapsto (\mathcal{Z}_j^\dagger, \mathcal{X}_j)$, while the modified controlled Z operators act as

$$\begin{aligned} \mathrm{CZ}_j \, \mathcal{Z}_j \, \mathrm{CZ}_j^\dagger &= \mathcal{Z}_j \\ \mathrm{CZ}_j \, \mathcal{X}_j \, \mathrm{CZ}_j^\dagger &= \mathcal{Z}_{j-n}^{-g_n} \, \mathcal{Z}_{j-(n-1)}^{-g_{n-1}} \cdots \mathcal{Z}_{j-2}^{-g_2} \, \mathcal{Z}_{j-1}^{-g_1} \, \mathcal{X}_j, \\ \mathrm{CZ}_j \, \mathcal{X}_\ell \, \mathrm{CZ}_j^\dagger &= \mathcal{X}_\ell \, \mathcal{Z}_j^{-g_{j-\ell}}, \qquad \ell = j-1, \, j-2, \cdots, \, j-n. \end{aligned} \qquad (4.30)$$

Observing that local operators $\mathcal{X}_j$ and $\mathcal{Z}_j \mathcal{Z}_{j+2}^{g_1} \cdots \mathcal{Z}_{j+n-1}^{g_{n-1}} \mathcal{Z}_{j+n}^{g_n}$ are only affected by a finite number of unitaries, we deduce the transformation rules

$$\widetilde{\mathsf{D}}_{\mathrm{KW}} \, \mathcal{Z}_j \, \mathcal{Z}_{j+1}^{g_1} \cdots \mathcal{Z}_{j+n-1}^{g_{n-1}} \, \mathcal{Z}_{j+n}^{g_n} = \mathcal{X}_j \, \widetilde{\mathsf{D}}_{\mathrm{KW}}, \tag{4.31}$$

$$\widetilde{\mathsf{D}}_{\mathrm{KW}} \, \mathcal{X}_j = \mathcal{Z}_{j-n}^{-g_n} \cdots \mathcal{Z}_{j-2}^{-g_2} \, \mathcal{Z}_{j-1}^{-g_1} \, \mathcal{Z}_j^{-1} \, \widetilde{\mathsf{D}}_{\mathrm{KW}}. \tag{4.32}$$

These are nothing but the KW duality transformations in Eq. (4.19). When Hamiltonian (4.28) is invariant under reflection, Eq. (4.23) applies and enforces

$$g_n = \sigma, \qquad g_{n-k} = \sigma \, g_k, \tag{4.33}$$

where $k = 1, 2, \cdots, n-1$. In this case, the operator $\widetilde{\mathsf{D}}_{\mathrm{KW}}$ then commutes with the Hamiltonian and becomes a non-invertible symmetry. While the $\widetilde{\mathsf{D}}_{\mathrm{KW}}$ defined in Eq. (4.29) is a product of unitary operators, it can still be non-invertible because it is an infinite product of unitary operators.

Starting from $\widetilde{\mathsf{D}}_{\mathrm{KW}}$, we can construct the operators $\mathsf{D}_{\mathrm{M}}$ and $\mathsf{D}_{\mathrm{KW}}$ by composing $\widetilde{\mathsf{D}}_{\mathrm{KW}}$ with spatial symmetries following Eqs. (4.20) and (4.25), respectively. The spatial symmetries can be explicitly represented as products of swap gates [122]

$$\mathsf{S}_{i,\,j} := \frac{1}{N} \sum_{\alpha,\beta=0}^{N-1} (\omega_N)^{\alpha\beta} \, \mathcal{X}_i^\alpha \, \mathcal{Z}_i^\beta \, \mathcal{X}_j^{-\alpha} \, \mathcal{Z}_j^{-\beta}, \tag{4.34a}$$

$$\mathsf{S}_{i,\,j}^2 = \mathbb{1}, \qquad \mathsf{S}_{i,\,j} = \mathsf{S}_{j,\,i}, \qquad \mathsf{S}_{i,\,j} \, \mathsf{S}_{j,\,k} \, \mathsf{S}_{i,\,j} = \mathsf{S}_{i,\,k}, \tag{4.34b}$$

which exchanges all operators written in terms of $\mathcal{Z}$ and $\mathcal{X}$ that are localized at site $j$ with those localized at site $i$. Then, the translation and reflection operators are given by

$$T := \cdots \mathsf{S}_{j+1,\,j} \, \mathsf{S}_{j,\,j-1} \, \mathsf{S}_{j-1,\,j-2} \cdots, \tag{4.35a}$$

$$M := \mathsf{S}_{1,-1} \, \mathsf{S}_{2,-2} \, \mathsf{S}_{3,-3} \cdots, \tag{4.35b}$$

respectively.[17]

### 4.3.2 Non-invertible reflection for exponential symmetry

As a first example, we construct the non-invertible operator $\mathsf{D}_{\mathrm{M}}$ explicitly for the case of an exponential symmetry with periodic boundary conditions. For simplicity, we consider prime $\mathbb{Z}_p$ qudits at sites $j$ of a closed chain with $L$ sites. This symmetry was explored in Section 2.1.1 and is generated by the operator

$$U_a = \prod_j^L \mathcal{X}_j^{a^j}, \tag{4.36}$$

where $a \in \{2, 3, \cdots p-2\}$ is fixed.

---

[17]Note that while the reflection operator $M$ is not written in terms of local operators, it can be using (4.34b): $\mathsf{S}_{j,-j} = \mathsf{S}_{-j,-j+1} \cdots \mathsf{S}_{j-3,j-2} \, \mathsf{S}_{j-1,j-2} \, \mathsf{S}_{j,j-1} \, \mathsf{S}_{j-1,j-2} \cdots \mathsf{S}_{-j+1,-j}$.

For this exponential symmetry, the generalized Ising model Hamiltonian becomes

$$H = -\sum_j \left\{ J \, \mathcal{Z}_j \, \mathcal{Z}_{j+1}^{-a^{-1}} + h \, \mathcal{X}_j + \text{H.c.} \right\}, \tag{4.37}$$

where periodic boundary conditions are implicitly imposed. For $U_a$ to be a symmetry of $H$, periodic boundary conditions require

$$T^L \, U_a \, T^{-L} = U_a^{a^L} \overset{!}{=} U_a. \tag{4.38}$$

This can always be satisfied by taking $L = 0 \bmod (p-1)$, which we assume is true throughout this example.[18] This Hamiltonian is invariant under translations but is not reflection symmetric for the $a$ we restrict to. Relatedly, the $\mathbb{Z}_p$ symmetry operator $U_a$ satisfies

$$M \, U_a \, M^\dagger = \prod_{j=1}^{L} \mathcal{X}_j^{a^{-j}} = U_{a^{-1}}, \tag{4.39}$$

and is, therefore, not closed under reflections.

The non-invertible reflection operator (4.20) is then constructed as follows. We first define the operator

$$\widetilde{\mathsf{D}}_{\text{KW}} := \sqrt{p} \, P_{a^{-1}} \, W \, \mathfrak{H}_L \, \text{CZ}_L \, \mathfrak{H}_{L-1} \, \text{CZ}_{L-1} \cdots \mathfrak{H}_3 \, \text{CZ}_3 \, \mathfrak{H}_2 \, \text{CZ}_2, \tag{4.40}$$

where $\mathfrak{H}_j$ is the usual Hadamard operator while $\text{CZ}_j$ is now a modified controlled Z operator with nontrivial action

$$\text{CZ}_j \, \mathcal{X}_j \, \text{CZ}_j^\dagger = \mathcal{Z}_{j-1}^{a^{-1}} \mathcal{X}_j \qquad \text{CZ}_j \, \mathcal{X}_{j-1} \, \text{CZ}_j^\dagger = \mathcal{X}_{j-1} \, \mathcal{Z}_j^{a^{-1}}. \tag{4.41}$$

The unitary operator $W$ acts only on the first and last sites, $j = 1$ and $j = L$, as

$$W \, \mathcal{X}_1 \, W^\dagger = \mathcal{Z}_1^{-1} \mathcal{X}_1 \, \mathcal{Z}_1^{-1} \, \mathcal{Z}_L^{a^{-1}}, \qquad W \, \mathcal{X}_L \, W^\dagger = \mathcal{Z}_1^{a^{-1}} \, \mathcal{X}_L. \tag{4.42}$$

Furthermore, the operator $P_{a^{-1}}$ is the projector onto the subspace where $U_{a^{-1}} = 1$. The fact that this projects to $U_{a^{-1}} = 1$ is related to how, as discussed in Section 2.1.1, gauging the exponential symmetry $U_a$ delivers a dual symmetry generated by $U_{a^{-1}}$.

The non-invertible reflection symmetry is then defined as the composition

$$\mathsf{D}_{\text{M}} = M \, \widetilde{\mathsf{D}}_{\text{KW}}, \tag{4.43a}$$

where $M$ is the version of operator (4.35b) for a finite lattice. As we show in Appendix D.2, this operator acts on local operators as

$$\mathsf{D}_{\text{M}} \, \mathcal{Z}_j \, \mathcal{Z}_{j+1}^{-a^{-1}} = \mathcal{X}_{-j} \, \mathsf{D}_{\text{M}}, \qquad \mathsf{D}_{\text{M}} \, \mathcal{X}_j = \mathcal{Z}_{-j}^{-1} \, \mathcal{Z}_{-j+1}^{a^{-1}} \, \mathsf{D}_{\text{M}}. \tag{4.43b}$$

It is straightforward to see that this action commutes with the Hamiltonian (4.37) when $J = h$. Therefore, although the Hamiltonian is not invariant under either reflection or the non-invertible KW duality alone, it has a non-invertible reflection symmetry.

---

[18]Depending on the choice of $a$, additional values of $L$ can be allowed that support the exponential symmetry

### 4.3.3 Double exponential symmetry

For this example, we restrict to prime qudits for simplicity and consider the modulated $\mathbb{Z}_p \times \mathbb{Z}_p$ symmetry generated by

$$U_1 = \prod_j \mathcal{X}_j^{a^j}, \qquad U_2 = \prod_j \mathcal{X}_j^{a^{-j}}. \tag{4.44a}$$

For this symmetry, translation and site-centered reflection symmetries act as

$$\begin{aligned}
T\,U_1\,T^{-1} &= U_1^a, & T\,U_2\,T^{-1} &= U_2^{a^{-1}}, \\
M\,U_1\,M^\dagger &= U_2, & M\,U_2\,M^\dagger &= U_1,
\end{aligned} \tag{4.44b}$$

respectively. In contrast with the previous example in Section 4.3.2, the symmetry group is now closed under the reflection transformation. Furthermore, periodic boundary conditions imply the constraints

$$T^L\,U_1\,T^{-L} = U_1^{a^L} \overset{!}{=} U_1, \qquad T^L\,U_2\,T^{-L} = U_2^{a^{-L}} \overset{!}{=} U_2, \tag{4.45}$$

which are satisfied when $L = 0 \bmod p - 1$.

Since there are two independent modulated symmetries, $n = 2$ in Eq. (4.27). The single $\Delta_{i,j}$ that annihilates the two functions $f_j^{(1)} = a^j$ and $f_j^{(2)} = a^{-j}$ has support on $r = n + 1 = 3$ sites and is parametrized as

$$\Delta_{j,\ell} = \delta_{j,\ell} - (a + a^{-1})\delta_{j,\ell-1} + \delta_{j,\ell-2} \equiv (\delta_{j+1,k} - a^{-1}\delta_{jk})(\delta_{k,\ell-1} - a\delta_{k\ell}). \tag{4.46}$$

This corresponds to choosing $g_2 = 1$ and $g_1 = -a - a^{-1}$ in Eq. (4.27). It is even under reflection and hence $\sigma = +1$. With this choice, Hamiltonian (4.28) becomes

$$H = -\sum_j \left\{ J\,\mathcal{Z}_j\,\mathcal{Z}_{j+1}^{-a-a^{-1}}\,\mathcal{Z}_{j+2} + h\,\mathcal{X}_j + \text{H.c.} \right\}. \tag{4.47}$$

The KW self-duality symmetry is then implemented by the non-invertible operator

$$\mathsf{D}_{\mathrm{KW}} = p\,P_1\,P_2\,T^{-1}\,W\,\mathfrak{H}_L\,\mathrm{CZ}_L\,\mathfrak{H}_{L-1}\,\mathrm{CZ}_{L-1}\cdots\mathfrak{H}_3\,\mathrm{CZ}_3, \tag{4.48a}$$

where $\mathrm{CZ}_j$ and $\mathfrak{H}_j$ are appropriate controlled Z and Hadamard operators (see Appendix D for the explicit definitions) that implement the transformations

$$\begin{aligned}
\mathfrak{H}_j\,\mathcal{X}_j\,\mathfrak{H}_j^\dagger &= \mathcal{Z}_j^\dagger, & \mathfrak{H}_j\,\mathcal{Z}_j\,\mathfrak{H}_j^\dagger &= \mathcal{X}_j, \\
\mathrm{CZ}_j\,\mathcal{Z}_j\,\mathrm{CZ}_j^\dagger &= \mathcal{Z}_j & \mathrm{CZ}_j\,\mathcal{X}_j\,\mathrm{CZ}_j^\dagger &= \mathcal{Z}_{j-2}^{-1}\,\mathcal{Z}_{j-1}^{a+a^{-1}}\,\mathcal{X}_j, \\
\mathrm{CZ}_j\,\mathcal{X}_{j-2}\,\mathrm{CZ}_j^\dagger &= \mathcal{X}_{j-2}\,\mathcal{Z}_j^{-1}, & \mathrm{CZ}_j\,\mathcal{X}_{j-1}\,\mathrm{CZ}_j^\dagger &= \mathcal{X}_{j-1}\,\mathcal{Z}_j^{a+a^{-1}},
\end{aligned} \tag{4.48b}$$

while the only nontrivial action by $W$ is given by

$$\begin{aligned}
W\,\mathcal{X}_{L-1}\,W^\dagger &= \mathcal{Z}_1^{-1}\,\mathcal{X}_{L-1} & W\,\mathcal{X}_L\,W^\dagger &= \mathcal{Z}_1^{a+a^{-1}}\,\mathcal{Z}_2^{-1}\,\mathcal{X}_L, \\
W\,\mathcal{X}_2\,W^\dagger &= \mathcal{Z}_1^{a+a^{-1}}\,\mathcal{Z}_2^{-1}\,\mathcal{X}_2\,\mathcal{Z}_2^{-1}\,\mathcal{Z}_L^{-1}, & W\,\mathcal{X}_1\,W^\dagger &= \mathcal{Z}_1^{-1}\,\mathcal{X}_1\,\mathcal{Z}_1^{-1}\,\mathcal{Z}_2^{a+a^{-1}}\,\mathcal{Z}_L^{a+a^{-1}}\,\mathcal{Z}_{L-1}^{-1}. 
\end{aligned} \tag{4.48c}$$

Finally, $P_1$ and $P_2$ are two projectors onto the $U_1 = 1$ and $U_2 = 1$ subspaces, respectively. One verifies that the operator (4.48) implements the KW duality transformation as prescribed in Eq. (4.19). While the operator (4.48) consists of sequentially applying the same operators, $\mathfrak{H}_j$ and $\mathrm{CZ}_j$ in the bulk of the chain, at the boundaries it acts differently via the unitary operator $W$, which is required to impose periodic boundary conditions consistently. Despite this difference and not being manifestly translation invariant, the KW duality operator (4.48) commutes with translation operator $T$.

### 4.3.4 $\mathbb{Z}_N$ dipole symmetry

The final example we consider is the case of a $\mathbb{Z}_N$ dipole symmetry, which is a modulated $\mathbb{Z}_N \times \mathbb{Z}_N$ symmetry generated by the two unitary operators

$$U = \prod_j \mathcal{X}_j, \qquad D = \prod_j \mathcal{X}_j^j. \tag{4.49a}$$

Here, we allow $N$ to be any integer since there is no significant difference in the gauging procedure for $N$ being prime or not, as we showed in Section 3.2.2. This symmetry group is closed both under translation and site-centered reflection symmetries, which act as

$$\begin{aligned} T\,U\,T^{-1} &= U, & T\,D\,T^{-1} &= U\,D, \\ M\,U\,M^\dagger &= U, & M\,D\,M^\dagger &= D^\dagger, \end{aligned} \tag{4.49b}$$

respectively. Periodic boundary conditions imply

$$T^L\,D\,T^{-L} = U^L\,D \overset{!}{=} D, \tag{4.50}$$

which is satisfied when $L = 0 \bmod N$.

The simplest $\Delta_{j,\ell}$ that annihilates both the constant and polynomial degree 1 functions has interaction range $r = n + 1 = 3$ and is given by

$$\Delta_{j,\ell} = \delta_{j,\ell} - 2\delta_{j,\ell-1} + \delta_{j,\ell-2} \equiv \partial_{j,\ell}^2, \tag{4.51}$$

which corresponds to $g_2 = 1$ and $g_1 = -2$ in Eq. (4.27) and $\sigma = +1$. With these, Hamiltonian (4.28) becomes

$$H = -\sum_j \left\{ J\,\mathcal{Z}_j\,\mathcal{Z}_{j+1}^{-2}\,\mathcal{Z}_{j+2} + h\,\mathcal{X}_j + \text{H.c.} \right\}, \tag{4.52}$$

and the KW self-duality symmetry is implemented by the non-invertible operator

$$\mathrm{D}_{\mathrm{KW}} = N\,P_U\,P_D\,T^{-1}\,W\,\mathfrak{H}_L\,\mathrm{CZ}_L\,\mathfrak{H}_{L-1}\,\mathrm{CZ}_{L-1}\cdots\mathfrak{H}_3\,\mathrm{CZ}_3, \tag{4.53a}$$

where $\mathrm{CZ}_j$ and $\mathfrak{H}_j$ implement the transformations (see Appendix D for the explicit definitions of all these operators)

$$\begin{aligned} \mathfrak{H}_j\,\mathcal{X}_j\,\mathfrak{H}_j^\dagger &= \mathcal{Z}_j^\dagger, & \mathfrak{H}_j\,\mathcal{Z}_j\,\mathfrak{H}_j^\dagger &= \mathcal{X}_j, \\ \mathrm{CZ}_j\,\mathcal{Z}_j\,\mathrm{CZ}_j^\dagger &= \mathcal{Z}_j & \mathrm{CZ}_j\,\mathcal{X}_j\,\mathrm{CZ}_j^\dagger &= \mathcal{Z}_{j-2}^{-1}\,\mathcal{Z}_{j-1}^2\,\mathcal{X}_j, \\ \mathrm{CZ}_j\,\mathcal{X}_{j-2}\,\mathrm{CZ}_j^\dagger &= \mathcal{X}_{j-2}\,\mathcal{Z}_j^{-1}, & \mathrm{CZ}_j\,\mathcal{X}_{j-1}\,\mathrm{CZ}_j^\dagger &= \mathcal{X}_{j-1}^2\,\mathcal{Z}_j^{-2}, \end{aligned} \tag{4.53b}$$

and $W$

$$W\,\mathcal{X}_{L-1}\,W^\dagger = \mathcal{Z}_1^{-1}\,\mathcal{X}_{L-1}, \qquad\qquad W\,\mathcal{X}_L\,W^\dagger = \mathcal{Z}_2^{-1}\,\mathcal{Z}_1^2\,\mathcal{X}_L,$$
$$W\,\mathcal{X}_2\,W^\dagger = \mathcal{Z}_1^2\,\mathcal{Z}_2^{-1}\,\mathcal{X}_2\,\mathcal{Z}_2^{-1}\,\mathcal{Z}_L^{-1}, \qquad W\,\mathcal{X}_1\,W^\dagger = \mathcal{Z}_1^{-1}\,\mathcal{X}_1\,\mathcal{Z}_1^{-1}\,\mathcal{Z}_2^2\,\mathcal{Z}_L^2\,\mathcal{Z}_{L-1}^{-1}. \tag{4.53c}$$

Finally, $P_U$ and $P_D$ are two projectors onto the $U=1$ and $D=1$ subspaces, respectively. Just as in the previous section, the operator (4.53), indeed implements the KW duality transformation as prescribed in Eq. (4.26). As announced in Eq. (4.26), $\mathsf{D}_{\mathrm{KW}}$ satisfies the algebra,

$$\mathsf{D}_{\mathrm{KW}}\,\mathsf{D}_{\mathrm{KW}} = \mathsf{C}\,\sum_{a,b=1}^{N} U^a\,D^b, \tag{4.54}$$

which agrees with the algebras obtained in Refs. 46, 47, 52 for $\mathbb{Z}_2$ qubits and Ref. 37 for $\mathbb{Z}_N$ qudits.[19]

# 5  Outlook

In this paper, we explored a systematic formalism for gauging finite Abelian modulated symmetries in $1+1\mathrm{D}$ lattice models. Having worked in the Hamiltonian formalism and with bond algebras, our gauging procedure depended on an appropriate choice of Gauss's law to trivialize the modulated symmetry. By implementing this gauging procedure, we explored the rich landscape of dual modulated symmetries and constructed new Kramers-Wannier dualities and non-invertible symmetries. Our results open the door to a handful of interesting follow-up directions, which we summarize here.

While for simplicity, we considered only finite Abelian modulated symmetries in this paper, modulated symmetry can also be non-Abelian. In an upcoming work, we extend our results to non-Abelian finite modulated symmetries and study the gauging web of models with such symmetries. This gauging web includes models with a new type of Lieb-Schultz-Mattis (LSM) anomaly between non-invertible symmetries and lattice translations and models with non-invertible modulated symmetries.

By studying how modulated symmetries are gauged, our work narrowed in on the kinematic features of modulated symmetries. Nevertheless, there are additional related kinematic aspects that were outside the scope of this work but quite interesting to explore. For instance, we performed an untwisted gauging of modulated symmetries (i.e., gauging with a trivial discrete torsion class), and extending this to twisted gauging is worthwhile. This could be done using SPT entanglers for modulated symmetries [107] or by modifying the Gauss operators as Ref. 50 did for non-modulated symmetries. Furthermore, it would be interesting to understand our results from two related perspectives. First is from the standpoint of the symmetry defects, whose mobility would be affected by the symmetry being modulated. The second is using the framework of topological holography, where symmetry defects and charges are described by topological defects of a gapped theory in one higher dimension. See [123–125] for some progress along this direction for subsystem symmetries. A final kinematic follow-up is generalizing our formalism to higher dimensions, where gauging an invertible

---

[19]See Ref. 36 for the analog of the duality operator $\mathsf{D}_{\mathrm{KW}}$ in continuum theories with dipole symmetries.

Abelian modulated symmetry in $d+1$D would lead to a dual symmetry described by a nontrivial $d$-group formed by lattice symmetry group and dual $(d-1)$-form modulated symmetry.

Using kinematic aspects to guide investigations into dynamical properties of models is fruitful, and our results provide such theoretical guidance. For instance, it would be interesting to use our results to study the dynamical consequences of the non-invertible KW self-dual symmetry for modulated symmetries. This non-invertible symmetry could appear at critical points between modulated symmetry preserving and symmetry breaking phases. Unlike in the non-modulated cases studied, the lattice translations are nontrivial in the IR due to the modulated symmetries. Understanding how it characterizes gapped and gapless phases is an important follow-up that we leave for ongoing future work.

# Acknowledgments

We are thankful to Arkya Chatterjee, Christopher Mudry, and Shu-Heng Shao for helpful related discussion. S.D.P. is supported by the National Science Foundation Graduate Research Fellowship under Grant No. 2141064. G.D. is supported by DOE Grant No. DE-FG02-06ER46316 and is grateful to Perimeter Institute for Theoretical Physics, where part of this work was completed. H.T.L. is supported in part by a Croucher fellowship from the Croucher Foundation, the Packard Foundation and the Center for Theoretical Physics at MIT. Ö.M.A. is supported by Swiss National Science Foundation (SNSF) under Grant No. P500PT-214429 and National Science Foundation (NSF) DMR-2022428.

# A    Relevant aspects of ring theory

In this appendix, we review aspects of ring theory that are relevant to the main text—particularly Section 3.

## A.1    Rings and modules

Let us first review the definition of rings and some fundamental aspects of modules. A ring $R$ is a set equipped with operations

$$+\colon R \times R \to R, \qquad \cdot\colon R \times R \to R, \tag{A.1}$$

referred to as addition and multiplication, respectively. Furthermore, for $R$ to be a ring, it must form an Abelian group under addition $+$ and a monoid under multiplication $\cdot$, and multiplication must be distributive with respect to addition. These requirements are sometimes called the ring axioms, and they explicitly mean that all $a, b, c \in R$ must obey:

1. $+$ is associative: $(a + b) + c = a + (b + c)$,

2. $+$ is commutative: $a + b = b + a$,

3. additive identity: $\exists \, 0 \in R$ such that $a + 0 = a$,

4. additive inverse: $\exists \, -a \in R$ for each $a$ such that $a + (-a) = 0$,

5. $\cdot$ is associative: $(a \cdot b) \cdot c = a \cdot (b \cdot c)$,

6. multiplicative identity: $\exists \, 1 \in R$ such that $1 \cdot a = a \cdot 1 = a$,

7. left distributivity: $a \cdot (b + c) = (a \cdot b) + (a \cdot c)$,

8. right distributivity: $(b + c) \cdot a = (b \cdot a) + (c \cdot a)$.

When the non-zero elements $a \in R$ each have a multiplicative inverse and form an Abelian group under multiplication, $R$ is a field. Therefore, rings are often thought of as generalizations of fields. When $\cdot$ is commutative but each non-zero $a \in R$ does not have a multiplicative inverse, then $R$ is called a commutative ring.

A subring $S$ of a ring $R$ is a subset that itself is a ring under the same binary operations of $R$ restricted to the subset. This implies that the group $(S, +)$ is a subgroup of $(R, +)$, and that the monoid $(S, \cdot)$ is a submonoid of $(R, \cdot)$. A subring is called an ideal if for each $s \in S$ and $r \in R$, $rs$ and $sr$ are in $S$. When only all $rs$ (resp. $sr$) are in $S$, then the subring is called a left (resp. right) ideal.

Three simple examples of rings are:

1. The reals $\mathbb{R}$ is a ring, where addition $+$ and multiplication $\cdot$ operations are the ordinary addition and multiplication of real numbers. Since the multiplication of real numbers is commutative and each non-zero real number has a multiplicative inverse, $\mathbb{R}$ is, in fact, a field.

2. The integers $\mathbb{Z}$ is also an example of a ring, where $+$ and $\cdot$ are the ordinary addition and multiplication of integers. While multiplication of integers is commutative, not every non-zero integer has a multiplicative inverse. Therefore, $\mathbb{Z}$ is not a field but a commutative ring.

3. For general $N \in \mathbb{Z}_{>0}$, $\mathbb{Z}_N \simeq \mathbb{Z}/N\mathbb{Z}$ is a commutative ring, where $+$ and $\cdot$ are the addition and multiplication modulo $N$. When $N$ is a prime number $p$, each $n \neq 0 \bmod p$ in $\mathbb{Z}_p$ has a multiplicative inverse $n^{-1} = n^{p-2} \bmod p$ by Fermat's little theorem. Therefore, $\mathbb{Z}_N$ is a field when $N$ is a prime number.

4. The set of square $n \times n$ matrices with entries in $\mathbb{R}$ forms a ring $\mathrm{M}_n(\mathbb{R})$ whose two binary operations are matrix addition and matrix multiplication. Since matrices do not commute, $\mathrm{M}_n(\mathbb{R})$ is a ring that is noncommutative. More generally, given a ring $R$, there is the matrix ring $\mathrm{M}_n(R)$ of $n \times n$ square matrices with entries in $R$.

Having reviewed the basic aspects of rings, we can now discuss modules. A module is a generalization of a vector space using rings. Recall that a vector space $V$ over a field $F$ is an Abelian group $(V, +)$ with scalar multiplication $*: F \times V \to V$. This scalar multiplication is a binary function that maps any scalar $a \in F$ and vector $\boldsymbol{v} \in V$ to another vector $a * \boldsymbol{v} \in V$. A module is a generalization

of $V$ in which the field of scalars $F$ can be a ring $R$. A module $M$ over the ring $(R, +_R, \cdot)$ is an Abelian group $(M, +)$ with an operation $*\colon R \times M \to M$ that, for all $m_1, m_2 \in M$ and $r_1, r_2 \in R$, obeys

1. $r_1 * (m_1 + m_2) = r_1 * m_1 + r_1 * m_2$,

2. $(r_1 +_R r_2) * m_1 = r_1 * m_1 + r_2 * m_1$,

3. $(r_1 \cdot r_2) * m_1 = r_1 * (r_2 * m_1)$,

4. $1 * m_1 = m_1$,

where $1$ is the multiplicative unit of $R$. When $R$ is a field, the module $M$ becomes a vector space over the field $R$. A subgroup $(N, +) \subseteq (M, +)$ forms a submodule of $M$ if $r * n \in N$ for all $n \in N$. Furthermore, given a subset $S$ of a module $M$ over a ring $R$, the annihilator $\mathrm{Ann}_R(S)$ of $S$ is the ideal

$$\mathrm{Ann}_R(S) = \{r \in R \mid r * s = 0 \text{ for all } s \in S\}. \tag{A.2}$$

Some simple examples of modules are:

1. Every Abelian group $G$ forms a module over the ring $\mathbb{Z}$. Given an integer $n > 0$ and $g \in G$, the scalar multiplication $*\colon \mathbb{Z} \times G \to G$ satisfies $n * g = \overbrace{g + g + \cdots + g}^{n \text{ times}}$, $(-n) * g = -(n * g)$, and $0 * g = e_g$, where $e_g$ is the identity element in $G$.

2. Matrices acting on vectors in an $n$-dimensional vector space $V$ over the field $F$ forms a module over the ring $M_n(F)$. The operation $*\colon M_n(F) \times V \to V$ corresponds to a matrix $M$ acting on a vector $\vec{v}$ and returning the vector $M * \vec{v}$. More generally, modules can describe vectors and matrices whose elements form a ring. For instance, the $\mathbb{Z}_N^{\times n}$ module over the ring $M_n(\mathbb{Z}_N)$ can be understood as $n$ dimensional vectors with elements in $\mathbb{Z}_N$ acted on by $n \times n$ matrices with elements in $\mathbb{Z}_N$ where all addition and multiplication is modulo $N$.

## A.2  Matrices over commutative rings

Given a field, the theory of linear algebra is a mature topic for studying matrices whose elements are in the field. Much progress has also been made in solving linear equations over commutative rings. In this paper, understanding matrices over commutative rings is essential for Section 3.1 of the main text, where we use the rank-nullity theorem for matrices over commutative rings. This appendix reviews relevant aspects of matrices over commutative rings, such as the generalized rank-nullity theorem.

An important notion for matrices over rings is that of regular matrices. Let $R$ be a commutative ring equipped with multiplication $\cdot$ and addition $+$. An $n \times r$ matrix $\mathcal{F}$ over $R$ is said to be regular if there exists an $r \times n$ matrix $\mathcal{G}$ over $R$ such that

$$\mathcal{F} \cdot \mathcal{G} \cdot \mathcal{F} = \mathcal{F}. \tag{A.3}$$

The matrix $\mathcal{G}$ is called the generalized inverse of $\mathcal{F}$, and a given $\mathcal{F}$ can have multiple generalized inverses. A necessary and sufficient condition for $\mathcal{F}$ to be regular is provided by the decomposition theorem [117, 118]. Before we state this theorem, we need to introduce several relevant concepts.

For $a_1, \ldots, a_k \in R$, the ideal generated by $a_1, \ldots, a_k$ is defined as

$$[a_1, \ldots, a_k] = \left\{ \sum_{i=1}^{k} r_i \cdot a_i \ \middle|\ \forall\, r_i \in R \right\}. \tag{A.4}$$

When the ideal $[a]$ is generated by a single element $a$, i.e., $k = 1$, we call it a principal ideal. For example, suppose $R = \mathbb{Z}_N$. For any $a \in \mathbb{Z}_N$ coprime to $N$, that is $\gcd(a, N) = 1$, the principal ideal generated by $a$ is $[a] = \mathbb{Z}_N$. Otherwise, $[a]$ is strictly smaller than $\mathbb{Z}_N$. Furthermore, if $a$ is a prime divisor of $N$, then $[a]$ is a maximal ideal in $\mathbb{Z}_N$, meaning there exists no other proper ideal of $\mathbb{Z}_N$ containing $[a]$.

An element $e \in R$ is idempotent if it obeys $e \cdot e = e$. In the following, we denote by $E(R)$ the set of all idempotent elements in $R$. Let us contextualize this definition in the example $R = \mathbb{Z}_N$. Assuming $N$ has the prime factorization

$$N = p_1^{r_1} \ldots p_k^{r_k}, \tag{A.5}$$

$\mathbb{Z}_N$ always has $2^k$ idempotent elements. Since $\mathbb{Z}_N \simeq \mathbb{Z}_{p_1^{r_1}} \times \cdots \times \mathbb{Z}_{p_k^{r_k}}$, idempotents of $\mathbb{Z}_N$ can be inferred from idempotents of $\mathbb{Z}_{p_i^{r_i}}$. If $e_i \in \mathbb{Z}_{p_i^{r_i}}$ is idempotent, then

$$e_i^2 = e_i \ \mathrm{mod}\ p_i^{r_i} \quad \Longrightarrow \quad e_i(e_i - 1) = 0 \ \mathrm{mod}\ p_i^{r_i}, \tag{A.6}$$

which implies that $p_i^{r_i}$ is a divisor of $e_i$ and $e_i - 1$ and, since $e_i$ and $e_i - 1$ are coprime, $e_i = 0 \ \mathrm{mod}\ p_i^{r_i}$ or $e_i = 1 \ \mathrm{mod}\ p_i^{r_i}$. Because $e \equiv (e_1, \cdots, e_k) \in \mathbb{Z}_N$ is an idempotent of $\mathbb{Z}_N$ if and only if each $e_i$ is an idempotent of $\mathbb{Z}_{p_i^{r_i}}$, the ring $\mathbb{Z}_N$ has $2^k$ idempotents. When $N$ is a prime number $p$, the only $\mathbb{Z}_p$ idempotent elements are $E(\mathbb{Z}_p) = \{0, 1\}$. More generally, $E(F)$ for any field $F$ is the set containing only the additive and multiplicative identity of $F$ (i.e., $E(F) = \{0, 1\}$). When $N$ is not prime, while $0$ and $1$ are still idempotents, there are other nontrivial idempotent elements as well. For instance, when $N = 10$, the idempotent elements are $E(\mathbb{Z}_{10}) = \{0, 1, 5, 6\}$.

The last notion we need to introduce before stating the decomposition theorem is that of Rao-regular matrices. To do so, we first recall that an $\ell \times \ell$ minor of an $n \times r$ matrix $\mathcal{F}$ is the determinant $\det(\mathcal{F}_\ell)$ of an $\ell \times \ell$ sub-matrix $\mathcal{F}_\ell$ of $\mathcal{F}$. We denote by $D_\ell(\mathcal{F})$ the ideal in $R$ generated by the $\ell \times \ell$ minors of $\mathcal{F}$, and by $\rho(\mathcal{F})$ the determinantal rank of $\mathcal{F}$, which is the largest $\ell$ such that $D_\ell(\mathcal{F}) \neq 0$. Then, a matrix $\mathcal{F}$ over $R$ with determinantal rank $\rho(\mathcal{F}) = t$ is Rao-regular if there exist an idempotent $e \in R$ such that

$$D_1(\mathcal{F}) = D_t(\mathcal{F}) = [e]. \tag{A.7}$$

Such idempotents are called Rao-idempotents.

Having introduced all the necessary concepts, we can now state the decomposition theorem (see Ref. 117 for the proof).

**Theorem 1** (Decomposition theorem). *An $n \times r$ matrix $\mathcal{F}$ over the commutative ring $R$ with determinantal rank $t := \rho(\mathcal{F})$ is regular if and only if there exist idempotents $e_0, \ldots, e_t \in E(R)$ such that*

*i) $e_0 + e_1 + \cdots + e_t = 1$ and $e_i \cdot e_j = 0$ for $0 \leq i, j \leq t$ and $i \neq j$,*

*ii) for $i = 0, 1, \ldots, t$, each matrix $\mathcal{F}_i := e_i \mathcal{F}$ is either the zero matrix or a Rao-regular matrix.*

Given a regular matrix over a commutative ring, there is a generalization of the rank-nullity theorem. To state it, however, we must first review how the notions of rank and nullity for matrices over fields generalize for matrices over commutative rings. In what follows, we introduce the minimal terminology required to state the generalized rank-nullity theorem and refer the reader to Ref. 117 for further details.

The generalization of the rank function is fairly straightforward. Given an $n \times r$ matrix $\mathcal{F}$ over the commutative ring $R$ with idempotents $e \in E(R)$, it is defined as the integer-valued function over $E(R)$ such that

$$\mathfrak{R}_{\mathcal{F}}(e) := \rho(e\mathcal{F}). \tag{A.8}$$

When $R = F$ is a field, $E(F) = \{0, 1\}$ and $\mathfrak{R}_{\mathcal{F}}(1)$ recovers the usual rank for matrices over a field.

The generalization of nullity is a bit more involved. Consider the (left) module $R^{\times r}$ over the matrix ring $\mathrm{M}_r(R)$. This describes $r$ dimensional vectors with elements in $R$ acted on by $r \times r$ matrices over $R$. We now assume that the $n \times r$ matrix $\mathcal{F}$ is regular and denote by $\mathcal{G}$ a generalized inverse of it. The kernel $\ker(\mathcal{F})$ of $\mathcal{F}$ is then the sub-module of $R^{\times r}$ generated by the columns of the matrix $(I_r - \mathcal{G} \cdot \mathcal{F})$, where $I_r$ is the $r \times r$ identity matrix. The nullity of $\mathcal{F}$ is then given by an appropriate generalization of its kernel's dimension. To define such a dimension-function, let us consider the $p$-th exterior product $\wedge^p(M)$ of a module $M$ whose elements are the $p$-th fold tensor product elements $m_1 \otimes \ldots \otimes m_p$ with the equivalence relation $m_1 \otimes \ldots \otimes m_p = 0$ if $m_i = m_j$ when $i \neq j$. Letting $\tilde{\rho}(M)$ denote the largest positive integer $p$ such that $\wedge^p(M) \neq (0)$ for the finitely generated sub-module $M$ of $R^{\times r}$, the dimension-function is defined as

$$\mathfrak{D}_M(e) = \tilde{\rho}(eM), \tag{A.9}$$

where $e \in E(R)$. Although this definition seems rather intricate, $\mathfrak{D}_M$ with $M = \ker(\mathcal{F})$ matches precisely our usual notion of nullity.

Given the above definitions, we can now state the generalized rank-nullity theorem for matrices over commutative rings. Denoting by $A$ a regular matrix over $R$ and $e \in E(R)$, we define the binary function

$$\chi_A(e) = \begin{cases} 1, & \text{if } e A \text{ is the zero matrix or Rao-regular with Rao-idempotent } e, \\ 0, & \text{if else.} \end{cases} \tag{A.10}$$

The purpose of this function is to check if $A$ is Rao-regular and if $e$ is a respective Rao-idempotent. The generalized rank-nullity theorem, proven in Ref. 117, is then:

**Theorem 2** (Rank-nullity theorem). *Let $\mathcal{F}$ be an $n \times r$ regular matrix over a commutative ring $R$. Then*

$$\chi_{\mathcal{F}} \, \chi_{I_r - \mathcal{G} \cdot \mathcal{F}} \left( \mathfrak{R}_{\mathcal{F}} + \mathfrak{D}_{\ker(\mathcal{F})} - r \right) = 0 \tag{A.11}$$

*where $\ker(\mathcal{F})$ is the null space of $\mathcal{F}$ and $\mathcal{G}$ is a generalized inverse of $\mathcal{F}$.*

The functions $\chi_{\mathcal{F}}$ and $\chi_{I_r - \mathcal{G} \cdot \mathcal{F}}$ in (A.11) ensure that the matrices $\mathcal{F}$ and $(I_r - \mathcal{G} \cdot \mathcal{F})$, respectively, are Rao-regular. If their respective Rao-idempotent $e = 1$ we can implicitly evaluate the functions in Eq. (A.11) which simplifies to

$$\mathfrak{R}_{\mathcal{F}} + \mathfrak{D}_{\ker(\mathcal{F})} = r, \tag{A.12}$$

where $\mathfrak{R}_{\mathcal{F}} = \rho(\mathcal{F})$. When all rows of the $n \times r$ matrix $\mathcal{F}$ are independent over $R$, the determinantal rank $\rho(\mathcal{F}) = n$ and (A.12) becomes

$$\mathfrak{D}_{\ker(\mathcal{F})} = r - n \tag{A.13}$$

which can be a useful result when showing the uniqueness of generators of $\mathrm{Ann}_{M_L(\mathbb{Z}_N)}(S)$.

**An Example**

As an example, let us consider the commutative ring $R = \mathbb{Z}_N$, with general $N$, and the $3 \times 4$ matrix

$$\mathcal{F}^{(4)} := \begin{pmatrix} f_1^{(1)} & f_2^{(1)} & f_3^{(1)} & f_4^{(1)} \\ f_1^{(2)} & f_2^{(2)} & f_3^{(2)} & f_4^{(1)} \\ f_1^{(3)} & f_2^{(3)} & f_3^{(3)} & f_4^{(3)} \end{pmatrix} \bmod N = \begin{pmatrix} 1 & 1 & 1 & 1 \\ 1 & 2 & 3 & 4 \\ 0 & 2 & 6 & 12 \end{pmatrix} \bmod N. \tag{A.14}$$

This matrix is relevant to the $\mathbb{Z}_N$ quadrupole symmetry from Section 3.2.3, where $f_j^{(1)} = 1$, $f_j^{(1)} = j$, and $f_j^{(3)} = j^2 - j$. It is easy to check by brute force that any generalized inverse of $\mathcal{F}^{(4)}$, if exists, must take the form

$$\mathcal{G}^{(4)} = \begin{pmatrix} a & b & c \\ 6 - 3a & -3b - 3 & 2^{-1} - 3c \\ 3a - 8 & 3b + 5 & 3c - 1 \\ 3 - a & -b - 2 & 2^{-1} - c \end{pmatrix}, \tag{A.15}$$

where $a, b, c \in \mathbb{Z}_N$ are parameters chosen such that all the matrix elements $\mathcal{G}_{ij}^{(4)} \in \mathbb{Z}_N$ and $2^{-1}$ is the multiplicative inverse of 2 module $N$. However, $2 \in \mathbb{Z}_N$ only has a multiplicative inverse when $N$ is odd. Furthermore, for even $N$, one can never eliminate the factors of $2^{-1}$ through convenient choices of parameters $a, b$, and $c$. Explicitly, there is no parameter $c$ such that

$$\mathcal{G}_{13}^{(4)} = c, \qquad \mathcal{G}_{23}^{(4)} = 2^{-1} - 3c, \qquad \mathcal{G}_{33}^{(4)} = 3c - 1, \qquad \text{and} \qquad \mathcal{G}_{43}^{(4)} = 2^{-1} - c \tag{A.16}$$

are all integers mod $N$ for $N$ even. Therefore, $\mathcal{G}^{(4)}$ exists only if $\gcd(2, N) = 1$, so $\mathcal{F}^{(4)}$ is regular only when $N$ is odd.

Let us also illustrate the decomposition theorem through the matrix $\mathcal{F}^{(4)}$ in Eq. (A.14). When $N$ is even, it is clear that $[1] \neq [2, 6]$ and $\mathcal{F}^{(4)}$ is not Rao-regular. We can further prove that $\mathcal{F}^{(4)}$ is not regular as it does not obey the Decomposition Theorem. For concreteness, let us consider $N = 12$, whose idempotent set is $E(\mathbb{Z}_{12}) = \{0, 1, 4, 9\}$. The decomposition involving the nontrivial idempotent elements that obey condition i) is $e_0 = e_1 = 0$, $e_2 = 4$, and $e_3 = 9$, such that

$$\mathcal{F}^{(4)} = \mathcal{F}_0^{(4)} + \mathcal{F}_1^{(4)} + \mathcal{F}_2^{(4)} + \mathcal{F}_3^{(4)}, \tag{A.17}$$

with $\mathcal{F}_i^{(4)} = e_i \mathcal{F}^{(4)}$. For $i = 0$ and 1, the matrices $\mathcal{F}_i^{(4)}$ are all zero. The nontrivial matrices are

$$\mathcal{F}_2^{(4)} = \begin{pmatrix} 4 & 4 & 4 & 4 \\ 4 & 8 & 0 & 4 \\ 0 & 8 & 0 & 0 \end{pmatrix} \quad \text{and} \quad \mathcal{F}_3^{(4)} = \begin{pmatrix} 9 & 9 & 9 & 9 \\ 9 & 6 & 3 & 0 \\ 0 & 6 & 6 & 0 \end{pmatrix}. \tag{A.18}$$

The first evidence that this decomposition fails the Decomposition Theorem is that both matrices have determinantal rank $\rho(\mathcal{F}_2^{(4)}) = \rho(\mathcal{F}_3^{(4)}) = 3$, which should be forbidden. Second, a direct violation of the theorem is that both $\mathcal{F}_2^{(4)}$ and $\mathcal{F}_3^{(4)}$ are not Rao-regular since $D_1(\mathcal{F}_2^{(4)}) = [4]$ and $D_3(\mathcal{F}_2^{(4)}) = [8] \neq [4]$. Similarly, $D_1(\mathcal{F}_3^{(4)}) = [9]$ and $D_3(\mathcal{F}_3^{(4)}) = [6] \neq [9]$.

# B  Polynomial symmetries and translation invariance

In this appendix, we prove a statement made in Section 2.1.3 of the main text. We consider a system of $\mathbb{Z}_p$ qudits, where $p$ is a prime number, on sites $j$ of an infinite chain and acted on by the clock and shift operators $\mathcal{Z}_j$ and $\mathcal{X}_j$. We posit the existence of a translation invariant Hamiltonian that commutes with the modulated symmetry operator

$$U_{p(j)} = \prod_j \mathcal{X}_j^{p(j)} \tag{B.1}$$

where $p(j) = \sum_{n=0}^{m} c_n j^n$ is an order $m < p - 1$ polynomial with $c_n \in \mathbb{Z}$. Here, $m$ is restricted by Fermat's little theorem $j^{p-1} = 1 \bmod p$. Since we assume $U_{p(j)}$ is a symmetry of a translation invariant Hamiltonian, $U_{p(j+x)}$ for any integer $x$ must also commute with the Hamiltonian.

While the modulated functions $\{p(j + x)\}$ are closed under translations, they are seemingly dependent on the choice of coefficients $c_n$ defining $p(j)$. However, as we now argue, there is a choice of generators independent of $c_n$. Firstly, since $U_{p(j)}$ and $U_{p(j+x)}$ are symmetry operators, $U_{p(j+x)} U_{p(j)}^\dagger$ is as well. Then, using the binomial expansion, $p(j + x)$ can be written as

$$p(j + x) = p(j) + \sum_{k=0}^{m-1} \left[ \sum_{n=k+1}^{m} c_n \binom{n}{k} x^{n-k} \right] j^k, \tag{B.2}$$

from which we see that while the modulated lattice functions of $U_{p(j)}$ and $U_{p(j+x)}$ are $m$-th order polynomials, the modulated lattice function of $U_{p(j+x)} U_{p(j)}^\dagger$ is an $(m-1)$-th order polynomial for all $x$. Conjugating this symmetry operator by the translation operator $T^y$, the operator $U_{p(j+x+y)} U_{p(j+y)}^\dagger$ must also commute with the Hamiltonian, and therefore the $(m-2)$-th order polynomial modulated

symmetry $U_{p(j+x+y)}U_{p(j+y)}^{\dagger}U_{p(j+x)}^{\dagger}U_{p(j)}$ does too. This procedure can be repeated $m$ times until we construct a complicated set of modulated operators that commute with the Hamiltonian whose modulated functions are

$$\left\{\sum_{n=0}^{m}c_nj^n, \quad \sum_{n=0}^{m-1}c_n^{(a,m-1)}j^n, \quad \sum_{n=0}^{m-2}c_n^{(b,m-2)}j^n, \quad \cdots, \quad c_0^{(c,1)}+c_1^{(c,1)}j, \quad c_0^{(d,0)}\right\}, \tag{B.3}$$

where $a,b,\cdots,c,d$ in the superscripts labels different polynomials obtained from this procedure.

We can simplify (B.3) using the $\mathbb{Z}_p$ structure. The above tells us that any translation invariant Hamiltonian commuting with (B.1) also commutes with

$$\prod_j \mathcal{X}_j^{c_0^{(d,0)}} \tag{B.4}$$

where $c_0^{(d,0)}$ is an (unimportant) constant constructed from the coefficients $c_n$ of the original polynomial $p(j)$. Since $p$ is prime, there exists an element $[c_0^{(d,0)}]^{-1}\in\mathbb{Z}_p$ such that $[c_0^{(d,0)}]^{-1}c_0^{(d,0)}=1 \bmod p$ and

$$\left(\prod_j \mathcal{X}_j^{c_0^{(d,0)}}\right)^{[c_0^{(d,0)}]^{-1}} = \prod_j \mathcal{X}_j. \tag{B.5}$$

Therefore, $\prod_j \mathcal{X}_j$ must also commute with the Hamiltonian. Multiplying this operator with the modulated operators whose modulated functions were (B.3), we can find a new set of commuting operators whose modulated functions are

$$\left\{\sum_{n=1}^{m}c_nj^n, \quad \sum_{n=1}^{m-1}c_n^{(a,m-1)}j^n, \quad \sum_{n=1}^{m-2}c_n^{(b,m-2)}j^n, \quad \cdots, \quad c_1^{(c,1)}j, \quad 1\right\}. \tag{B.6}$$

This can be repeated for the modulated operators with polynomials $c_1^{(c,1)}j$ to replace them with a single modulated operator with polynomial $j$ and then drop all $j$ terms in higher order polynomials to reduce (B.6) to $\left\{\sum_{n=2}^{m}c_nj^n, \quad \sum_{n=2}^{m-1}c_n^{(a,m-1)}j^n, \quad \sum_{n=2}^{m-2}c_n^{(b,m-2)}j^n, \quad \cdots, \quad j, \quad 1\right\}$. Repeating this for each order of polynomial reduces the set of polynomials to $\{j^n:n\in\{0,1,\cdots,m-1,m\}\}$. Therefore, if an $m$th order polynomial modulated operator commutes with a translation invariant Hamiltonian, the operators

$$U_{j^n}=\prod_j \mathcal{X}_j^{j^n}, \qquad n\in\{0,1,\cdots,m-1,m\}, \tag{B.7}$$

must also commute, which are generators of a $\mathbb{Z}_p^{\times m+1}$ modulated symmetry.

## C   Position-dependent Gauss's laws

In this appendix, we discuss an alternative way of gauging the modulated symmetries from Section 2. Recall such symmetries acted on translation invariant systems of $\mathbb{Z}_p$ qudits, where $p$ is a prime integer, and were generated by

$$U_q=\prod_j(\mathcal{X}_j)^{f_j^{(q)}}, \tag{C.1}$$

with $q = 1, 2, \cdots, n$. Since this generates finite symmetry in a translation invariant system, each lattice function $f_j^{(q)}$ satisfies

$$f_j^{(q)} = f_{j+x^{(q)}}^{(q)} \mod p, \tag{C.2}$$

with $x^{(q)}$ denoting the smallest such positive integer. Therefore, the symmetry's spatial modulation is periodic with period

$$x := \operatorname{lcm}(x^{(1)}, x^{(2)}, \cdots, x^{(n)}). \tag{C.3}$$

Indeed, the symmetry operators $U_q$ can be written as non-modulated operators with respect to the enlarged length $x$ unit cell:

$$U_q = \prod_{j \in x\,\mathbb{Z}} \left( \prod_{k=1}^{x} (\mathcal{X}_{j+k})^{f_{j+k}^{(q)}} \right). \tag{C.4}$$

Here, to gauge the symmetry we introduce $n$-types of $\mathbb{Z}_p$ qudits onto link $\langle j, j+1 \rangle$ that are respectively acted on by $X_{j,j+1}^{(q)}$. The Gauss's laws relating these new qudits to the originals are

$$G_j^{(q)} = \mathcal{X}_j \, (X_{j-1,j}^{(q)})^{f_{j-1}^{(q)}} (X_{j,j+1}^{(q)})^{-f_{j+1}^{(q)}} = 1, \tag{C.5}$$

which trivialize the entire $\mathbb{Z}_p^{\times n}$ symmetry since

$$U_q = \prod_j (G_j^{(q)})^{f_j^{(q)}}. \tag{C.6}$$

These Gauss's laws introduce the gauge redundancy

$$\mathcal{Z}_j \sim (\omega_p)^{\sum_q \lambda_j^{(q)}} \mathcal{Z}_j \qquad Z_{j-1,j}^{(q)} \sim (\omega_p)^{\lambda_j^{(q)} f_{j-1}^{(q)} - \lambda_{j-1}^{(q)} f_j^{(q)}} Z_{j-1,j}^{(q)}, \tag{C.7}$$

which is generated by $\prod_{j,q} (G_j^{(q)})^{\lambda_j^{(q)}}$.

Before gauging, the bond algebra is generated by

$$\mathfrak{B} := \left\langle \mathcal{X}_j, \ \prod_{\ell=j}^{j+n} \mathcal{Z}_\ell^{\Delta_{j,\ell}} \right\rangle, \tag{C.8}$$

where $\Delta_{j,\ell}$ obeys $\sum_{\ell=j}^{j+n} \Delta_{j,\ell} f_\ell^{(q)} = 0$ for all $q = 1, \cdots, n$. After gauging, the bond algebra can be made gauge invariant by minimal coupling, which leads to

$$\mathfrak{B}^\vee := \left\langle \mathcal{X}_j, \ \prod_{\ell=j}^{j+n} \mathcal{Z}_\ell^{\Delta_{j,\ell}} \prod_{\ell=j}^{j+n-1} \left( \prod_{q=1}^{n} \left( Z_{\ell,\ell+1}^{(q)} \right)^{\nabla_{j,\ell}^{(q)}} \right) \right\rangle. \tag{C.9}$$

where $\nabla_{j,\ell}^{(q)}$ obeys

$$\begin{aligned}
\nabla_{j,j-1}^{(q)} &= \nabla_{j,j+n}^{(q)} = 0 \mod p, \\
\Delta_{j,\ell} + f_{\ell-1}^{(q)} \nabla_{j,\ell-1}^{(q)} - f_{\ell+1}^{(q)} \nabla_{j,\ell}^{(q)} &= 0 \mod p, \quad \text{for } \ell = j, \cdots, j+n.
\end{aligned} \tag{C.10}$$

The solution to these equations is

$$\nabla_{j,\ell}^{(q)} = (f_\ell^{(q)} f_{\ell+1}^{(q)})^{-1} \sum_{k=j}^{\ell} f_k^{(q)} \Delta_{j,k} \mod p, \quad \text{for } \ell = j-1, \cdots, j+n, \tag{C.11}$$

where $(f_\ell^{(q)} f_{\ell+1}^{(q)})^{-1}$ denotes the $\mathbb{Z}_p$ inverse of $f_\ell^{(q)} f_{\ell+1}^{(q)}$.

While the Gauss operator (C.5) trivializes the modulated symmetry, it depends explicitly on the lattice site $j$ through the modulated functions $f_j^{(q)}$. Therefore, gauging with it projects the lattice translation symmetry down to the subgroup generated by $T^x$. Importantly, since the modulated symmetry operators (C.1) all commute with $T^x$, gauging with (C.5) does not preserve the symmetry's spatial modulation and is equivalent to gauging the non-modulated operators (C.4) with respect to the enlarged length $x$ unit cell. This is why we did not use this Gauss operator to gauge the full modulated $\mathbb{Z}_p^{\times n}$ symmetry in Section 2. However, this Gauss operator may still be useful for gauging sub-symmetries of a modulated symmetry that are not closed under lattice translations.

Let us end this appendix by proving that the Gauss operator (C.5) gauges the symmetry with respect to the length $x$ unit cell. We do so using the unitary transformation $U = \prod_{q,j:\ j \notin x\mathbb{Z}} U_j^{(q)}$, where $U_j^{(q)}$ satisfies

$$U_j^{(q)}\, G_j^{(q)}\, (U_j^{(q)})^\dagger = (X_{j,j+1}^{(q)})^{f_{j+1}}. \tag{C.12}$$

After rotating the Hilbert space using $U$, since $p$ is a prime integer, the Gauss's laws $G_j^{(q)}$ for $j \notin x\mathbb{Z}$ set $X_{j,j+}^{(q)} = 1$ for $j \notin x\mathbb{Z}$. However, the $n$-types of $\mathbb{Z}_p$ qudits on links $\langle j_x, j_x + 1 \rangle$ with $j_x \in x\mathbb{Z}$ survive the transformation, and are constrained by the remaining Gauss's laws

$$\widetilde{G}_{j_x}^{(q)} := \prod_{k=1}^{x} (G_{j_x+k}^{(q)})^{f_{j_x+k}^{(q)}} = (X_{j_x,j_x+1}^{(q)} X_{j_x+x,j_x+x+1}^{(q)\dagger})^{f_{j_x}^{(q)} f_{j_x+1}^{(q)}} \prod_{k=1}^{x} (\mathcal{X}_{j_x+k})^{f_{j_x+k}^{(q)}}. \tag{C.13}$$

Therefore, gauging using the Gauss operators (C.5) is equivalent to trivializing (C.4) using the length-$x$ unit cell with $n$-types of $\mathbb{Z}_p$ qudits on $j_x \in x\mathbb{Z}$ that are acted on by $(X_{j_x,j_x+1}^{(q)})^{f_{j_x}^{(q)} f_{j_x+1}^{(q)}}$.

# D    Details on Kramers-Wannier self-duality symmetry

In this appendix, we provide the details on non-invertible KW duality operators on infinite and finite chains that are defined in Section 4.3. Following the discussion in Section 4, we assume that $\Delta_{j,\ell}$ can be parameterized as (recall Eq. (4.27))

$$\Delta_{j,j} = 1, \quad \Delta_{j,j+1} = g_1, \quad \Delta_{j,j+2} = g_2, \quad \cdots, \quad \Delta_{j,j+n-1} = g_{n-1}, \quad \Delta_{j,j+n} = g_n, \tag{D.1}$$

where $g_n$ are coefficients valued in either $\mathbb{Z}_p$ with prime $p$ or $\mathbb{Z}_N$ with non-prime $N$, and $n$ is the number of independent modulated symmetries.

## D.1    KW duality on infinite chain

As we discussed in Section 4.3, on an infinite chain, the $\widetilde{\mathsf{D}}_{\mathrm{KW}}$ operator can be constructed as the sequential circuit out of product of infinitely many unitary operators

$$\widetilde{\mathsf{D}}_{\mathrm{KW}} := \cdots \mathfrak{H}_{j+1}\, \mathrm{CZ}_{j+1}\, \mathfrak{H}_j\, \mathrm{CZ}_j\, \mathfrak{H}_{j-1}\, \mathrm{CZ}_{j-1} \cdots . \tag{D.2a}$$

Here, $\mathfrak{H}_j$ is the Hadamard operator

$$\mathfrak{H}_j := \frac{1}{\sqrt{p}} \sum_{\alpha,\beta=0}^{p-1} \omega_p^{-\alpha\beta} \mathcal{X}_j^{\alpha-\beta} P_{\mathcal{Z}_j}^{(\beta)}, \qquad P_{\mathcal{Z}_j}^{(\alpha)} := \frac{1}{p} \sum_{\beta=0}^{p-1} \omega_p^{-\alpha\beta} \mathcal{Z}_j^\beta, \tag{D.2b}$$

with $P_{\mathcal{Z}_j}^{(\alpha)}$ being the projector onto the $\mathcal{Z}_j = \omega_p^\alpha$ subspace, and (ii) $\mathrm{CZ}_j$ is a modified controlled Z operator with $\Delta$ dependence and is defined as

$$\mathrm{CZ}_j := \sum_{\alpha=0}^{p-1} \mathcal{Z}_j^\alpha \mathfrak{P}_j^{(\alpha)}, \qquad \mathfrak{P}_j^{(\alpha)} := \frac{1}{p} \sum_{\beta=0}^{p-1} \omega_p^{-\alpha\beta} \left( \mathcal{Z}_{j-n}^{-g_n} \cdots \mathcal{Z}_{j-2}^{-g_2} \mathcal{Z}_{j-1}^{-g_1} \right)^\beta. \tag{D.2c}$$

In the above, $\mathfrak{P}_j^{(\alpha)}$ is a projector onto the $\mathcal{Z}_{j-n}^{-g_n} \cdots \mathcal{Z}_{j-2}^{-g_2} \mathcal{Z}_{j-1}^{-g_1} = \omega_p^\alpha$ subspace. It is straightforward to verify that these operators implement the transformations

$$\mathfrak{H}_j \, \mathcal{X}_j \, \mathfrak{H}_j^\dagger = \mathcal{Z}_j^\dagger, \qquad \mathfrak{H}_j \, \mathcal{Z}_j \, \mathfrak{H}_j^\dagger = \mathcal{X}_j, \tag{D.3a}$$

and

$$\begin{aligned}
\mathrm{CZ}_j \, \mathcal{Z}_j \, \mathrm{CZ}_j^\dagger &= \mathcal{Z}_j \\
\mathrm{CZ}_j \, \mathcal{X}_j \, \mathrm{CZ}_j^\dagger &= \mathcal{Z}_{j-n}^{-g_n} \mathcal{Z}_{j-(n-1)}^{-g_{n-1}} \cdots \mathcal{Z}_{j-2}^{-g_2} \mathcal{Z}_{j-1}^{-g_1} \mathcal{X}_j, \\
\mathrm{CZ}_j \, \mathcal{X}_\ell \, \mathrm{CZ}_j^\dagger &= \mathcal{X}_\ell \, \mathcal{Z}_j^{-g_{j-\ell}}, \qquad \ell = j-1, \, j-2, \cdots, \, j-n.
\end{aligned} \tag{D.3b}$$

Observing that local operators $\mathcal{X}_j$ and $\mathcal{Z}_{j-n}^{g_n} \mathcal{Z}_{j-n+1}^{g_{n-1}} \cdots \mathcal{Z}_{j-1}^{g_1} \mathcal{Z}_i$ are only affected by a finite number of unitaries, we deduce the transformation rules

$$\widetilde{\mathrm{D}}_{\mathrm{KW}} \, \mathcal{Z}_j \, \mathcal{Z}_{j+1}^{g_1} \mathcal{Z}_{j+2}^{g_2} \cdots \mathcal{Z}_{j+n}^{g_n} = \mathcal{X}_j \, \widetilde{\mathrm{D}}_{\mathrm{KW}}, \quad \widetilde{\mathrm{D}}_{\mathrm{KW}} \, \mathcal{X}_j = \mathcal{Z}_{j-n}^{-g_n} \mathcal{Z}_{j-n+1}^{-g_{n-1}} \cdots \mathcal{Z}_{j-1}^{-g_1} \mathcal{Z}_j^{-1} \widetilde{\mathrm{D}}_{\mathrm{KW}}. \tag{D.4}$$

## D.2 Non-invertible reflection for exponential symmetry

We consider the $\mathbb{Z}_p$ exponential symmetry ($p \geq 3$) defined in Eq. (4.36). The KW transformation is implemented by the non-invertible operator $\tilde{D}_{\mathrm{KW}}$ defined as

$$\tilde{D}_{\mathrm{KW}} = \sqrt{p} \, P_{a^{-1}} \, W \, \mathfrak{H}_L \, \mathrm{CZ}_L \, \mathfrak{H}_{L-1} \, \mathrm{CZ}_{L-1} \cdots \mathfrak{H}_3 \, \mathrm{CZ}_3 \, \mathfrak{H}_2 \, \mathrm{CZ}_2. \tag{D.5}$$

Here, $\mathfrak{H}_j$ is the Hadamard operator defined in Eq. (D.2b), while the modified controlled Z operator $\mathrm{CZ}_j$ is

$$\mathrm{CZ}_j = \sum_{\alpha=0}^{p-1} \mathcal{Z}_j^\alpha \mathfrak{P}_j^{(\alpha)}, \qquad \mathfrak{P}_j^{(\alpha)} := \frac{1}{p} \sum_{\beta=0}^{p-1} \omega_p^{-\alpha\beta} \left( \mathcal{Z}_{j-1}^{a^{-1}} \right)^\beta. \tag{D.6}$$

The unitary operator $W$ acts on the first and last sites of the lattice and defined as

$$W := \frac{1}{p} \sum_{\alpha,\beta=0}^{p-1} \omega_p^{-\alpha\beta} \mathcal{Z}_1^\alpha \left( \mathcal{Z}_1^{-1} \mathcal{Z}_L^{a^{-1}} \right)^\beta, \tag{D.7a}$$

with its only non-trivial actions being

$$W\,\mathcal{X}_1\,W^\dagger = \mathcal{Z}_1^{-1}\,\mathcal{X}_1\,\mathcal{Z}_1^{-1}\,\mathcal{Z}_L^{a^{-1}}, \qquad W\,\mathcal{X}_L\,W^\dagger = \mathcal{Z}_1^{a^{-1}}\,\mathcal{X}_L. \tag{D.7b}$$

The unitary operator $W$ consists of controlled Z-type operators that is modified for the boundary terms. The operator (D.2) on infinite chain then can be understood as the $L \to \infty$ limit of the operator (D.5). Finally, $P_{a^{-1}}$ is a projector onto the subspace which is symmetric under $U_{a^{-1}} \equiv M\,U_a\,M^\dagger$.

To verify that the operator (D.5) indeed implements the duality transformation, we act by the unitary operators on the symmetric local operators sequentially. Let us momentarily focus on how the symmetric operators

$$
\begin{aligned}
\mathcal{Z}_2\,\mathcal{Z}_3^{-a^{-1}}, && \mathcal{X}_2, \\
\mathcal{Z}_3\,\mathcal{Z}_4^{-a^{-1}}, && \mathcal{X}_3,
\end{aligned}
\tag{D.8}
$$

are transformed. Under the unitary $\mathrm{CZ}_2$ we have

$$
\begin{aligned}
\mathcal{Z}_2\,\mathcal{Z}_3^{-a^{-1}} \mapsto \mathcal{Z}_2\,\mathcal{Z}_3^{-a^{-1}}, && \mathcal{X}_2 \mapsto \mathcal{Z}_1^{a^{-1}}\mathcal{X}_2, \\
\mathcal{Z}_3\,\mathcal{Z}_4^{-a^{-1}} \mapsto \mathcal{Z}_3\,\mathcal{Z}_4^{-a^{-1}}, && \mathcal{X}_3 \mapsto \mathcal{X}_3,
\end{aligned}
\tag{D.9}
$$

Next, acting with the unitary $\mathfrak{H}_2$ gives

$$
\begin{aligned}
\mathcal{Z}_2\,\mathcal{Z}_3^{-a^{-1}} \mapsto \mathcal{X}_2\,\mathcal{Z}_3^{-a^{-1}}, && \mathcal{Z}_1^{a^{-1}}\,\mathcal{X}_2 \mapsto \mathcal{Z}_1^{a^{-1}}\,\mathcal{Z}_2^{-1}, \\
\mathcal{Z}_3\,\mathcal{Z}_4^{-a^{-1}} \mapsto \mathcal{Z}_3\,\mathcal{Z}_4^{-a^{-1}}, && \mathcal{X}_3 \mapsto \mathcal{X}_3,
\end{aligned}
\tag{D.10}
$$

Now, acting with the pair of unitaries $\mathrm{CZ}_3$ and $\mathfrak{H}_3$ gives

$$
\begin{aligned}
\mathcal{X}_2\,\mathcal{Z}_3^{-a^{-1}} \mapsto \mathcal{X}_2, && \mathcal{Z}_1^{a^{-1}}\,\mathcal{Z}_2^{-1} \mapsto \mathcal{Z}_1^{a^{-1}}\,\mathcal{Z}_2^{-1}, \\
\mathcal{Z}_3\,\mathcal{Z}_4^{-a^{-1}} \mapsto \mathcal{X}_3\,\mathcal{Z}_4^{-a^{-1}}, && \mathcal{X}_3 \mapsto \mathcal{Z}_2^{a^{-1}}\,\mathcal{Z}_3^{-1}.
\end{aligned}
\tag{D.11}
$$

We note that with these three steps, the first line matches the duality transformation (D.4). Acting with the remaining pairs of $\mathfrak{H}$ and $\mathrm{CZ}$ operators up to the unitary $W$ implements

$$
\begin{aligned}
\mathcal{Z}_2\,\mathcal{Z}_3^{-a^{-1}} \mapsto \mathcal{X}_2, && \mathcal{X}_2 \mapsto \mathcal{Z}_1^{a^{-1}}\,\mathcal{Z}_2^{-1}, \\
\mathcal{Z}_3\,\mathcal{Z}_4^{-a^{-1}} \mapsto \mathcal{X}_3, && \mathcal{X}_3 \mapsto \mathcal{Z}_2^{a^{-1}}\,\mathcal{Z}_3^{-1}, \\
\vdots && \vdots \\
\mathcal{Z}_{L-1}\,\mathcal{Z}_L^{-a^{-1}} \mapsto \mathcal{X}_{L-1}, && \mathcal{X}_L \mapsto \mathcal{Z}_{L-1}^{a^{-1}}\,\mathcal{Z}_L^{-1},
\end{aligned}
\tag{D.12a}
$$

which is the duality transformation (D.4) for all symmetric operators in the bulk except the remaining three

$$
\begin{aligned}
\mathcal{Z}_1\,\mathcal{Z}_2^{-a^{-1}} &\mapsto \mathcal{Z}_1\,\mathcal{X}_2^{-a^{-1}}\,\mathcal{X}_3^{-a^{-2}}\,\mathcal{X}_4^{-a^{-3}}\cdots\mathcal{X}_L^{-a^{1-L}}, \\
\mathcal{Z}_L\,\mathcal{Z}_1^{-a^{-1}} &\mapsto \mathcal{X}_L\,\mathcal{Z}_1^{-a^{-1}}, \\
\mathcal{X}_1 &\mapsto \mathcal{X}_1\,\mathcal{X}_2^{a^{-1}}\,\mathcal{X}_3^{a^{-2}}\,\mathcal{X}_4^{a^{-3}}\cdots\mathcal{X}_L^{a^{1-L}}.
\end{aligned}
\tag{D.12b}
$$

The unitary $W$ has a non-trivial action on these remaining three operators. It transforms them into

$$\mathcal{Z}_1\,\mathcal{Z}_2^{-a^{-1}} \mapsto \mathcal{Z}_1\,\mathcal{X}_2^{-a^{-1}}\,\mathcal{X}_3^{-a^{-2}}\,\mathcal{X}_4^{-a^{-3}}\cdots\mathcal{Z}_1^{-a^{-L}}\,\mathcal{X}_L^{-a^{1-L}},$$
$$\mathcal{Z}_L\,\mathcal{Z}_1^{-a^{-1}} \mapsto \mathcal{X}_L, \tag{D.13}$$
$$\mathcal{X}_1 \mapsto \mathcal{Z}_1^{-1}\,\mathcal{X}_1\,\mathcal{Z}_1^{-1}\,\mathcal{Z}_L^{a^{-1}}\,\mathcal{X}_2^{a^{-1}}\,\mathcal{X}_3^{a^{-2}}\,\mathcal{X}_4^{a^{-3}}\cdots\mathcal{Z}_1^{a^{-L}}\,\mathcal{X}_L^{a^{1-L}}.$$

Notice that the middle term now transforms according to Eq. (D.4). Using the fact that $a^L = 1$, which follows from imposing periodic boundary conditions, we can simplify the first and the last terms to

$$\mathcal{Z}_1\,\mathcal{Z}_2^{-a^{-1}} \mapsto \mathcal{X}_2^{-a^{-1}}\,\mathcal{X}_3^{-a^{-2}}\,\mathcal{X}_4^{-a^{-3}}\cdots\mathcal{X}_L^{-a^{1-L}},$$
$$\mathcal{X}_1 \mapsto \mathcal{Z}_L^{a^{-1}}\,\mathcal{Z}_1^{-1}\,\mathcal{X}_1\,\mathcal{X}_2^{a^{-1}}\,\mathcal{X}_3^{a^{-2}}\,\mathcal{X}_4^{a^{-3}}\cdots\mathcal{X}_L^{a^{1-L}}. \tag{D.14}$$

Finally, noting that the first and second string operators are nothing but $\mathcal{X}_1\,U_{a^{-1}}^{-1}$ and $U_{a^{-1}}$, the action of the projector $P_{a^{-1}}$ delivers

$$\mathcal{Z}_1\,\mathcal{Z}_2^{-a^{-1}} \mapsto \mathcal{X}_1,$$
$$\mathcal{X}_1 \mapsto \mathcal{Z}_L^{a^{-1}}\,\mathcal{Z}_1^{-1}, \tag{D.15}$$

which completes the duality transformation (D.4).

## D.3   Double exponential symmetry

We consider the modulated $\mathbb{Z}_p \times \mathbb{Z}_p$ symmetry ($p \geq 3$) generated by two $\mathbb{Z}_p$ exponential symmetries with $f_j^{(1)} = a^j$ and $f_j^{(2)} = a^{-j}$ for $a \in \mathbb{Z}_p$. We find $n = 2$ and set $g_1 = -a - a^{-1}$ and $g_2 = 1$ in Eq. (D.4). As claimed in Section 4.3, the corresponding duality $\widetilde{\mathsf{D}}_{\mathrm{KW}}$ operator is given by

$$\widetilde{\mathsf{D}}_{\mathrm{KW}} = p\,P_1\,P_2\,W\,\mathfrak{H}_L\,\mathrm{CZ}_L\,\mathfrak{H}_{L-1}\,\mathrm{CZ}_{L-1}\cdots\mathfrak{H}_3\,\mathrm{CZ}_3. \tag{D.16}$$

Here, $\mathfrak{H}_j$ is the Hadamard operator defined in Eq. (D.2b), while the modified controlled Z operator $\mathrm{CZ}_j$ is

$$\mathrm{CZ}_j = \sum_{\alpha=0}^{p-1}\mathcal{Z}_j^\alpha\,\mathfrak{P}_j^{(\alpha)}, \qquad \mathfrak{P}_j^{(\alpha)} := \frac{1}{p}\sum_{\beta=0}^{p-1}\omega_p^{-\alpha\beta}\left(\mathcal{Z}_{j-2}^{-1}\,\mathcal{Z}_{j-1}^{a+a^{-1}}\right)^\beta \tag{D.17}$$

The unitary operator $W$ only acts on the first and last $n = 2$ sites of the lattice and is defined as

$$W := W_{L-1}\,W_L\,W_1\,W_2,$$

$$W_{L-1} := \frac{1}{p}\sum_{\alpha,\beta=0}^{p-1}\omega_p^{-\alpha\beta}\,\mathcal{Z}_1^{-\alpha}\,\mathcal{Z}_{L-1}^\beta,$$

$$W_L := \frac{1}{p}\sum_{\alpha,\beta=0}^{p-1}\omega_p^{-\alpha\beta}\,\mathcal{Z}_2^{-\alpha}\,\mathcal{Z}_1^{(a+a^{-1})\alpha}\,\mathcal{Z}_L^\beta, \tag{D.18a}$$

$$W_1 := \frac{1}{p}\sum_{\alpha,\beta=0}^{p-1}\omega_p^{-\alpha\beta}\,\mathcal{Z}_1^{-\alpha}\,\mathcal{Z}_2^{-(a+a^{-1})\beta}\,\mathcal{Z}_1^\beta,$$

$$W_2 := \frac{1}{p}\sum_{\alpha,\beta=0}^{p-1}\omega_p^{-\alpha\beta}\,\mathcal{Z}_2^{-\alpha}\,\mathcal{Z}_2^\beta,$$

with its only nontrivial actions being

$$W\,\mathcal{X}_{L-1}\,W^\dagger = \mathcal{Z}_1^{-1}\,\mathcal{X}_{L-1} \qquad\qquad W\,\mathcal{X}_L\,W^\dagger = \mathcal{Z}_1^{a+a^{-1}}\,\mathcal{Z}_2^{-1}\,\mathcal{X}_L,$$

$$W\,\mathcal{X}_2\,W^\dagger = \mathcal{Z}_1^{a+a^{-1}}\,\mathcal{Z}_2^{-1}\,\mathcal{X}_2\,\mathcal{Z}_2^{-1}\,\mathcal{Z}_L^{-1}, \quad W\,\mathcal{X}_1\,W^\dagger = \mathcal{Z}_1^{-1}\,\mathcal{X}_1\,\mathcal{Z}_1^{-1}\,\mathcal{Z}_2^{a+a^{-1}}\,\mathcal{Z}_L^{a+a^{-1}}\,\mathcal{Z}_{L-1}^{-1}. \quad \text{(D.18b)}$$

The unitary operator $W$ consists of controlled Z-type operators that is modified for the boundary terms. For the double exponential symmetry, the duality operator (D.2) on infinite chain then can be understood as the $L \to \infty$ limit of the operator (D.16). Finally, the projectors $P_1$ and $P_2$ projects onto the subspace which is symmetric under both exponential symmetries.

As in the previous section, to verify that the operator (D.16) indeed implements the duality transformation, which can be seen through the sequential action of the unitary operators. Acting with all the unitary operators up to $W$, implements

$$
\begin{aligned}
\mathcal{Z}_3\,\mathcal{Z}_4^{-a-a^{-1}}\,\mathcal{Z}_5 &\mapsto \mathcal{X}_3, & \mathcal{X}_3 &\mapsto \mathcal{Z}_1^{-1}\,\mathcal{Z}_2^{a+a^{-1}}\,\mathcal{Z}_3^{-1} \\
\mathcal{Z}_4\,\mathcal{Z}_5^{-a-a^{-1}}\,\mathcal{Z}_6 &\mapsto \mathcal{X}_4, & \mathcal{X}_4 &\mapsto \mathcal{Z}_2^{-1}\,\mathcal{Z}_3^{a+a^{-1}}\,\mathcal{Z}_4^{-1} \\
&\;\vdots & &\;\vdots \\
\mathcal{Z}_{L-3}\,\mathcal{Z}_{L-2}^{-a-a^{-1}}\,\mathcal{Z}_{L-1} &\mapsto \mathcal{X}_{L-3}, & \mathcal{X}_{L-1} &\mapsto \mathcal{Z}_{L-3}^{-1}\,\mathcal{Z}_{L-2}^{a+a^{-1}}\,\mathcal{Z}_{L-1}^{-1}, \\
\mathcal{Z}_{L-2}\,\mathcal{Z}_{L-1}^{-a-a^{-1}}\,\mathcal{Z}_L &\mapsto \mathcal{X}_{L-2}, & \mathcal{X}_L &\mapsto \mathcal{Z}_{L-2}^{-1}\,\mathcal{Z}_{L-1}^{a+a^{-1}}\,\mathcal{Z}_L^{-1},
\end{aligned}
\tag{D.19a}
$$

which is the duality transformation (D.4) on the local symmetric operators except the remaining six

$$
\begin{aligned}
\mathcal{Z}_L\,\mathcal{Z}_1^{-a-a^{-1}}\,\mathcal{Z}_2 &\mapsto \mathcal{X}_L\,\mathcal{Z}_1^{-a-a^{-1}}\,\mathcal{Z}_2, \\
\mathcal{Z}_{L-1}\,\mathcal{Z}_L^{-a-a^{-1}}\,\mathcal{Z}_1 &\mapsto \mathcal{X}_{L-1}\,\mathcal{Z}_1, \\
\mathcal{Z}_2\,\mathcal{Z}_3^{-a-a^{-1}}\,\mathcal{Z}_4 &\mapsto \mathcal{Z}_2\,\mathcal{X}_3^{-w_1}\,\mathcal{X}_4^{-w_2}\,\mathcal{X}_5^{-w_3}\cdots\mathcal{X}_{L-1}^{-w_{L-3}}\,\mathcal{X}_L^{-w_{L-2}}, \\
\mathcal{Z}_1\,\mathcal{Z}_2^{-a-a^{-1}}\,\mathcal{Z}_3 &\mapsto \mathcal{Z}_1\,\mathcal{Z}_2^{-a-a^{-1}}\,\mathcal{X}_3^{w_0}\,\mathcal{X}_4^{w_1}\,\mathcal{X}_5^{w_2}\cdots\mathcal{X}_{L-1}^{w_{L-4}}\,\mathcal{X}_L^{w_{L-3}}, \\
\mathcal{X}_1 &\mapsto \mathcal{X}_1\,\mathcal{X}_3^{-w_0}\,\mathcal{X}_4^{-w_1}\,\mathcal{X}_5^{-w_2}\cdots\mathcal{X}_{L-1}^{-w_{L-4}}\,\mathcal{X}_L^{-w_{L-3}}, \\
\mathcal{X}_2 &\mapsto \mathcal{X}_2\,\mathcal{X}_3^{w_1}\,\mathcal{X}_4^{w_2}\,\mathcal{X}_5^{w_3}\cdots\mathcal{X}_{L-1}^{w_{L-3}}\,\mathcal{X}_L^{w_{L-2}},
\end{aligned}
\tag{D.19b}
$$

where

$$w_j := \sum_{\alpha=0}^{j} a^{j-2\alpha}. \tag{D.19c}$$

While the first two terms remain local, the last four are mapped to non-local string operators. The final unitary operator $W$ maps these six operators to

$$
\begin{aligned}
\mathcal{Z}_L\,\mathcal{Z}_1^{-a-a^{-1}}\,\mathcal{Z}_2 &\mapsto \mathcal{X}_L, \\
\mathcal{Z}_{L-1}\,\mathcal{Z}_L^{-a-a^{-1}}\,\mathcal{Z}_1 &\mapsto \mathcal{X}_{L-1}, \\
\mathcal{Z}_2\,\mathcal{Z}_3^{-a-a^{-1}}\,\mathcal{Z}_4 &\mapsto \mathcal{Z}_2\,\mathcal{X}_3^{-w_1}\cdots\mathcal{X}_{L-1}^{-w_{L-3}}\,\mathcal{X}_L^{-w_{L-2}}\,\mathcal{Z}_1^{w_{L-3}-(a+a^{-1})\,w_{L-2}}\,\mathcal{Z}_2^{w_{L-2}}, \\
\mathcal{Z}_1\,\mathcal{Z}_2^{-a-a^{-1}}\,\mathcal{Z}_3 &\mapsto \mathcal{Z}_1\,\mathcal{Z}_2^{-a-a^{-1}}\,\mathcal{X}_3^{w_0}\cdots\mathcal{X}_{L-1}^{w_{L-4}}\,\mathcal{X}_L^{w_{L-3}}\,\mathcal{Z}_1^{-w_{L-4}+(a+a^{-1})\,w_{L-3}}\,\mathcal{Z}_2^{-w_{L-3}}, \\
\mathcal{X}_1 &\mapsto \mathcal{Z}_1^{-1}\,\mathcal{X}_1\,\mathcal{Z}_1^{w_{L-4}-(a+a^{-1})\,w_{L-3}-1}\,\mathcal{Z}_2^{w_{L-3}+a+a^{-1}}\,\mathcal{Z}_{L-1}^{-1}\,\mathcal{Z}_L^{a+a^{-1}}\,\mathcal{X}_3^{-w_0}\cdots\mathcal{X}_L^{-w_{L-3}}, \\
\mathcal{X}_2 &\mapsto \mathcal{Z}_2^{-1}\,\mathcal{X}_2\,\mathcal{Z}_2^{-1}\,\mathcal{Z}_L^{-1}\,\mathcal{X}_3^{w_1}\cdots\mathcal{X}_L^{w_{L-2}}\,\mathcal{Z}_1^{a+a^{-1}-w_{L-3}+(a+a^{-1})\,w_{L-2}}\,\mathcal{Z}_2^{-w_{L-2}}.
\end{aligned}
$$
$$\tag{D.20}$$

We recall that when imposing periodic boundary conditions we demand $L = 0 \mod p - 1$, and therefore $L = n(p-1)$ for some positive integer $n$. Using this fact, we obtain

$$w_{L-3} + a + a^{-1} = a^{-3}\left(\sum_{\alpha=0}^{L-3} a^{-2\alpha} + a^4 + a^2\right) = a^{-3}\sum_{\alpha=0}^{n(p-1)-1} a^{-2\alpha}$$

$$= n\, a^{-3}\sum_{\alpha=0}^{p-2} a^{-2\alpha} = 0 \mod p, \qquad (D.21)$$

where in reaching the last line, we have used the facts that (i) $a^{p-1} = 1 \mod p$ and, thus, the terms in the summation are periodic with period $p - 1$, and (ii) the summation of even powers of $a$ vanishes modulo $p$. Similarly, one can show that

$$w_{L-2} = -1 \mod p, \qquad w_{L-4} = -a^2 - a^{-2} - 1 \mod p. \qquad (D.22)$$

Therefore, one finds that all the $\mathcal{Z}$ operators with exponents $w_i$ in Eq. (D.20) simplify to

$$
\begin{aligned}
&\mathcal{Z}_L\,\mathcal{Z}_1^{-a-a^{-1}}\,\mathcal{Z}_2 \mapsto \mathcal{X}_L,\\
&\mathcal{Z}_{L-1}\,\mathcal{Z}_L^{-a-a^{-1}}\,\mathcal{Z}_1 \mapsto \mathcal{X}_{L-1},\\
&\mathcal{Z}_2\,\mathcal{Z}_3^{-a-a^{-1}}\,\mathcal{Z}_4 \mapsto \mathcal{X}_3^{-w_1}\cdots\mathcal{X}_{L-1}^{-w_{L-3}}\,\mathcal{X}_L^{-w_{L-2}},\\
&\mathcal{Z}_1\,\mathcal{Z}_2^{-a-a^{-1}}\,\mathcal{Z}_3 \mapsto \mathcal{X}_3^{w_0}\cdots\mathcal{X}_{L-1}^{w_{L-4}}\,\mathcal{X}_L^{w_{L-3}},\\
&\mathcal{X}_1 \mapsto \mathcal{Z}_{L-1}^{-1}\,\mathcal{Z}_L^{a+a^{-1}}\,\mathcal{Z}_1^{-1}\,\mathcal{X}_1\,\mathcal{X}_3^{-w_0}\cdots\mathcal{X}_L^{-w_{L-3}},\\
&\mathcal{X}_2 \mapsto \mathcal{Z}_2^{-1}\,\mathcal{Z}_1^{a+a^{-1}}\,\mathcal{Z}_L^{-1}\,\mathcal{X}_2\,\mathcal{X}_3^{w_1}\cdots\mathcal{X}_L^{w_{L-2}}.
\end{aligned}
\qquad (D.23)
$$

Through similar manipulations, one verifies that the last two string operators in Eq. (D.20) can be expressed in terms of products of exponential symmetry operators and hence become identity when hit with the projectors $P_1$ and $P_2$. The first two strings are then simply mapped to $\mathcal{X}_2$ and $\mathcal{X}_1$, respectively. To summarize, the operators on the left-hand side of Eq. (D.23) are mapped to

$$
\begin{aligned}
&\mathcal{Z}_L\,\mathcal{Z}_1^{-a-a^{-1}}\,\mathcal{Z}_2 \mapsto \mathcal{X}_L,\\
&\mathcal{Z}_{L-1}\,\mathcal{Z}_L^{-a-a^{-1}}\,\mathcal{Z}_1 \mapsto \mathcal{X}_{L-1},\\
&\mathcal{Z}_2\,\mathcal{Z}_3^{-a-a^{-1}}\,\mathcal{Z}_4 \mapsto \mathcal{X}_2,\\
&\mathcal{Z}_1\,\mathcal{Z}_2^{-a-a^{-1}}\,\mathcal{Z}_3 \mapsto \mathcal{X}_1,\\
&\mathcal{X}_1 \mapsto \mathcal{Z}_{L-1}^{-1}\,\mathcal{Z}_L^{a+a^{-1}}\,\mathcal{Z}_1^{-1},\\
&\mathcal{X}_2 \mapsto \mathcal{Z}_L^{-1}\,\mathcal{Z}_1^{a+a^{-1}}\,\mathcal{Z}_2^{-1}.
\end{aligned}
\qquad (D.24)
$$

which completes the KW transformation (D.4).

## D.4 $\mathbb{Z}_N$ dipole symmetry

For a $\mathbb{Z}_N$ dipole symmetry, we find $n = 2$ and set $g_1 = -2$ and $g_2 = 1$ in Eq. (D.4). As claimed in Section 4, the corresponding KW duality operator is given by

$$\widetilde{\mathsf{D}}_{\mathrm{KW}} = N\,P_U\,P_D\,W\,\mathfrak{H}_L\,\mathrm{CZ}_L\,\mathfrak{H}_{L-1}\,\mathrm{CZ}_{L-1}\cdots\mathfrak{H}_3\,\mathrm{CZ}_3. \qquad (D.25)$$

Here, $\mathfrak{H}_j$ is the Hadamard operator defined in Eq. (D.2b), while the modified controlled Z operator $\mathrm{CZ}_j$ is

$$\mathrm{CZ}_j = \sum_{\alpha=0}^{p-1} \mathcal{Z}_j^\alpha \, \mathfrak{P}_j^{(\alpha)}, \qquad \mathfrak{P}_j^{(\alpha)} := \frac{1}{p} \sum_{\beta=0}^{p-1} \omega_p^{-\alpha\beta} \left( \mathcal{Z}_{j-2}^{-1} \, \mathcal{Z}_{j-1}^2 \right)^\beta \tag{D.26}$$

The unitary operator $W$ only acts on the first and last two sites of the lattice and is defined as

$$
\begin{aligned}
W &:= W_{L-1} \, W_L \, W_1 \, W_2, \\
W_{L-1} &:= \frac{1}{p} \sum_{\alpha,\beta=0}^{p-1} \omega_p^{-\alpha\beta} \, \mathcal{Z}_1^{-\alpha} \, \mathcal{Z}_{L-1}^\beta, \\
W_L &:= \frac{1}{p} \sum_{\alpha,\beta=0}^{p-1} \omega_p^{-\alpha\beta} \, \mathcal{Z}_2^{-\alpha} \, \mathcal{Z}_1^{2\alpha} \, \mathcal{Z}_L^\beta, \\
W_1 &:= \frac{1}{p} \sum_{\alpha,\beta=0}^{p-1} \omega_p^{-\alpha\beta} \, \mathcal{Z}_1^{-\alpha} \, \mathcal{Z}_2^{-2\beta} \, \mathcal{Z}_1^\beta, \\
W_2 &:= \frac{1}{p} \sum_{\alpha,\beta=0}^{p-1} \omega_p^{-\alpha\beta} \, \mathcal{Z}_2^{-\alpha} \, \mathcal{Z}_2^\beta,
\end{aligned}
\tag{D.27a}
$$

with its only nontrivial actions being

$$
\begin{aligned}
W \, \mathcal{X}_{L-1} \, W^\dagger &= \mathcal{Z}_1^{-1} \, \mathcal{X}_{L-1}, & W \, \mathcal{X}_L \, W^\dagger &= \mathcal{Z}_2^{-1} \, \mathcal{Z}_1^2 \, \mathcal{X}_L, \\
W \, \mathcal{X}_2 \, W^\dagger &= \mathcal{Z}_1^2 \, \mathcal{Z}_2^{-1} \, \mathcal{X}_2 \, \mathcal{Z}_2^{-1} \, \mathcal{Z}_L^{-1}, & W \, \mathcal{X}_1 \, W^\dagger &= \mathcal{Z}_1^{-1} \, \mathcal{X}_1 \, \mathcal{Z}_1^{-1} \, \mathcal{Z}_2^2 \, \mathcal{Z}_L^2 \, \mathcal{Z}_{L-1}^{-1}.
\end{aligned}
\tag{D.27b}
$$

The unitary operator $W$ consists of control-Z-type operators that is modified for the boundary terms. For dipole symmetry, the KW duality operator (D.2) on infinite chain then can be understood as the $L \to \infty$ limit of the operator (D.25). Finally, the projectors $P_U$ and $P_D$ project onto the symmetric subspace, i.e., $U = 1$ and $D = 1$.

As in the previous section, to verify that the operator (D.25) indeed implements the duality transformation, we act by the unitary operators on the symmetric local operators sequentially. Acting with all the unitary operators up to $W$, implements

$$
\begin{aligned}
\mathcal{Z}_3 \mathcal{Z}_4^{-2} \mathcal{Z}_5 &\mapsto \mathcal{X}_3, & \mathcal{X}_3 &\mapsto \mathcal{Z}_1^{-1} \, \mathcal{Z}_2^2 \, \mathcal{Z}_3^{-1}, \\
\mathcal{Z}_4 \mathcal{Z}_5^{-2} \mathcal{Z}_6 &\mapsto \mathcal{X}_4, & \mathcal{X}_4 &\mapsto \mathcal{Z}_2^{-1} \, \mathcal{Z}_3^2 \, \mathcal{Z}_4^{-1}, \\
&\;\;\vdots & &\;\;\vdots \\
\mathcal{Z}_{L-3} \mathcal{Z}_{L-2}^{-2} \mathcal{Z}_{L-1} &\mapsto \mathcal{X}_{L-3}, & \mathcal{X}_{L-1} &\mapsto \mathcal{Z}_{L-3}^{-1} \, \mathcal{Z}_{L-2}^2 \, \mathcal{Z}_{L-1}^{-1}, \\
\mathcal{Z}_{L-2} \mathcal{Z}_{L-1}^{-2} \mathcal{Z}_L &\mapsto \mathcal{X}_{L-2}, & \mathcal{X}_L &\mapsto \mathcal{Z}_{L-2}^{-1} \, \mathcal{Z}_{L-1}^2 \, \mathcal{Z}_L^{-1},
\end{aligned}
\tag{D.28a}
$$

which is the KW duality transformation (D.4) on the local symmetric operators except the remaining

six

$$\begin{aligned}
\mathcal{Z}_L\,\mathcal{Z}_1^{-2}\,\mathcal{Z}_2 &\mapsto \mathcal{X}_L\,\mathcal{Z}_1^{-2}\,\mathcal{Z}_2, \\
\mathcal{Z}_{L-1}\,\mathcal{Z}_L^{-2}\,\mathcal{Z}_1 &\mapsto \mathcal{X}_{L-1}\,\mathcal{Z}_1, \\
\mathcal{Z}_2\,\mathcal{Z}_3^{-2}\,\mathcal{Z}_4 &\mapsto \mathcal{Z}_2\,\mathcal{X}_3^{-2}\,\mathcal{X}_4^{-3}\,\mathcal{X}_5^{-4}\cdots\mathcal{X}_{L-1}^{2-L}\,\mathcal{X}_L^{1-L}, \\
\mathcal{Z}_1\,\mathcal{Z}_2^{-2}\,\mathcal{Z}_3 &\mapsto \mathcal{Z}_1\,\mathcal{Z}_2^{-2}\,\mathcal{X}_3\,\mathcal{X}_4^2\,\mathcal{X}_5^3\cdots\mathcal{X}_{L-1}^{L-3}\,\mathcal{X}_L^{L-2}, \\
\mathcal{X}_1 &\mapsto \mathcal{X}_1\,\mathcal{X}_3^{-1}\,\mathcal{X}_4^{-2}\,\mathcal{X}_5^{-3}\cdots\mathcal{X}_{L-1}^{3-L}\,\mathcal{X}_L^{2-L}, \\
\mathcal{X}_2 &\mapsto \mathcal{X}_2\,\mathcal{X}_3^2\mathcal{X}_4^3\,\mathcal{X}_5^4\cdots\mathcal{X}_{L-1}^{L-2}\,\mathcal{X}_L^{L-1},
\end{aligned}$$ (D.28b)

While the first two terms remain local, the last four are mapped to non-local string operators. The final unitary operator $W$ maps these six operators to

$$\begin{aligned}
\mathcal{Z}_L\,\mathcal{Z}_1^{-2}\,\mathcal{Z}_2 &\mapsto \mathcal{X}_L, \\
\mathcal{Z}_{L-1}\,\mathcal{Z}_L^{-2}\,\mathcal{Z}_1 &\mapsto \mathcal{X}_{L-1}, \\
\mathcal{Z}_2\,\mathcal{Z}_3^{-2}\,\mathcal{Z}_4 &\mapsto \mathcal{Z}_1^{-L}\,\mathcal{Z}_2^L\,\mathcal{X}_3^{-2}\,\mathcal{X}_4^{-3}\,\mathcal{X}_5^{-4}\cdots\mathcal{X}_{L-1}^{2-L}\,\mathcal{X}_L^{1-L}, \\
\mathcal{Z}_1\,\mathcal{Z}_2^{-2}\,\mathcal{Z}_3 &\mapsto \mathcal{Z}_1^L\,\mathcal{Z}_2^{-L}\,\mathcal{X}_3\,\mathcal{X}_4^2\,\mathcal{X}_5^3\cdots\mathcal{X}_{L-1}^{L-3}\,\mathcal{X}_L^{L-2}, \\
\mathcal{X}_1 &\mapsto \mathcal{Z}_1^{-1}\,\mathcal{X}_1\,\mathcal{Z}_1^{-1}\,\mathcal{Z}_2^2\,\mathcal{Z}_L^2\,\mathcal{Z}_{L-1}^{-1}\,\mathcal{X}_3^{-1}\,\mathcal{X}_4^{-2}\,\mathcal{X}_5^{-3}\cdots\mathcal{Z}_1^{1-L}\,\mathcal{Z}_2^{L-2}\,\mathcal{X}_{L-1}^{3-L}\,\mathcal{X}_L^{2-L}, \\
\mathcal{X}_2 &\mapsto \mathcal{Z}_1^2\,\mathcal{Z}_2^{-1}\,\mathcal{X}_2\,\mathcal{Z}_2^{-1}\,\mathcal{Z}_L^{-1}\mathcal{X}_3^2\mathcal{X}_4^3\,\mathcal{X}_5^4\cdots\mathcal{Z}_1^L\,\mathcal{Z}_2^{1-L}\,\mathcal{X}_{L-1}^{L-2}\,\mathcal{X}_L^{L-1}.
\end{aligned}$$ (D.29)

Note that each string resembles the dipole symmetry operator. In particular, using the fact that consistency with periodic boundary conditions requires $L = 0 \bmod N$, we can cancel all terms the raised to the power $L$ and obtain the six terms

$$\begin{aligned}
\mathcal{Z}_L\,\mathcal{Z}_1^{-2}\,\mathcal{Z}_2 &\mapsto \mathcal{X}_L, \\
\mathcal{Z}_{L-1}\,\mathcal{Z}_L^{-2}\,\mathcal{Z}_1 &\mapsto \mathcal{X}_{L-1}, \\
\mathcal{Z}_2\,\mathcal{Z}_3^{-2}\,\mathcal{Z}_4 &\mapsto \mathcal{X}_3^{-2}\,\mathcal{X}_4^{-3}\,\mathcal{X}_5^{-4}\cdots\mathcal{X}_{L-1}^2\,\mathcal{X}_L^1, \\
\mathcal{Z}_1\,\mathcal{Z}_2^{-2}\,\mathcal{Z}_3 &\mapsto \mathcal{X}_3\,\mathcal{X}_4^2\,\mathcal{X}_5^3\cdots\mathcal{X}_{L-1}^{-3}\,\mathcal{X}_L^{-2}, \\
\mathcal{X}_1 &\mapsto \mathcal{Z}_{L-1}^{-1}\,\mathcal{Z}_L^2\,\mathcal{Z}_1^{-1}\,\mathcal{X}_1\,\mathcal{X}_3^{-1}\,\mathcal{X}_4^{-2}\,\mathcal{X}_5^{-3}\cdots\mathcal{X}_{L-1}^3\,\mathcal{X}_L^2, \\
\mathcal{X}_2 &\mapsto \mathcal{Z}_L^{-1}\,\mathcal{Z}_1^2\,\mathcal{Z}_2^{-1}\,\mathcal{X}_2\,\mathcal{X}_3^2\mathcal{X}_4^3\,\mathcal{X}_5^4\cdots\mathcal{X}_{L-1}^{-2}\,\mathcal{X}_L^{-1}.
\end{aligned}$$ (D.30)

Using the fact that projectors $P_U$ and $P_D$ sets $U = 1$ and $D = 1$, one verifies that the last two strings are trivialized while the first two strings are just equal to $\mathcal{X}_2$ and $\mathcal{X}_1$, respectively. To summarize, the operators on the left-hand side of Eq. (D.28b) are mapped to

$$\begin{aligned}
\mathcal{Z}_L\,\mathcal{Z}_1^{-2}\,\mathcal{Z}_2 &\mapsto \mathcal{X}_L, \\
\mathcal{Z}_{L-1}\,\mathcal{Z}_L^{-2}\,\mathcal{Z}_1 &\mapsto \mathcal{X}_{L-1}, \\
\mathcal{Z}_2\,\mathcal{Z}_3^{-2}\,\mathcal{Z}_4 &\mapsto \mathcal{X}_2, \\
\mathcal{Z}_1\,\mathcal{Z}_2^{-2}\,\mathcal{Z}_3 &\mapsto \mathcal{X}_1, \\
\mathcal{X}_1 &\mapsto \mathcal{Z}_{L-1}^{-1}\,\mathcal{Z}_L^2\,\mathcal{Z}_1^{-1}, \\
\mathcal{X}_2 &\mapsto \mathcal{Z}_L^{-1}\mathcal{Z}_1^2\,\mathcal{Z}_2^{-1},
\end{aligned}$$ (D.31)

which completes the KW transformation (D.4).

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
