# Peer review of "Gauging modulated symmetries: Kramers-Wannier dualities and non-invertible reflections"

_SciPost Physics_

## Round 2 · Referee Report · Anonymous (Referee 1) · 2024-8-22

Report

In this paper the authors give a treatment of the gauging of spatially modulated symmetries in translation-invariant spin chains. The authors apply this to a class of generalized Ising models with such modulated symmetries, and identify non-invertible Kramers-Wannier type dualities (sometimes involving a spatial reflection). A number of natural follow up questions are suggested.

The material on gauging spatial symmetries consists of a mix of examples and general results. The examples are certainly helpful, but it is not always clear what the most important material is. The authors give a summary in the introduction, nevertheless it would be helpful to streamline these two sections, perhaps by moving some of the examples to an appendix.

The treatment of the Kramers-Wannier dualities is interesting. It would be useful to comment on whether the generalized Ising models have previously appeared in the literature, and if not, more could be said about their physics. For example, the duality becomes a symmetry at the point $J=h$ - is this a critical point of the model?

Requested changes

Some additional comments on the generalized Ising models.

Consider streamlining the main text to improve readability of the paper.

I found the following typos that should be corrected: 1) Footnote 5 "refereed to" 2) (2.47) right hand equation $j\rightarrow j+1$. [Also consider writing $p=2$ in (2.44) and replacing $Z^\dagger$ with $Z$ in (2.48).]

Recommendation

Publish (meets expectations and criteria for this Journal)

---

## Round 2 · Referee Report · Anonymous (Referee 2) · 2024-9-20

Report

The authors discuss the construction of generalized reflection symmetries in 1+1d lattice Hamiltonian systems. They study generalized Ising models with internal symmetries that do not necessarily commute with lattice translations. Furthermore, they gauge these internal symmetries and construct non-invertible reflection symmetries when the system is self-dual under gauging. I recommend the draft for publication and have the following questions:

The authors implement gauging using Gauss's law in equation (1.15). They should comment on how this equation is derived and whether the expression is unique. In particular, there might be multiple (untwisted/twisted) ways to gauge the symmetry, corresponding to different forms of Gauss's law.

What are the possible gapped or gapless phases that remain invariant under non-invertible reflection symmetries? Specifically, what is the low-energy behavior of these systems with non-invertible reflection symmetries at J=h?

Requested changes

  1. Provide more detail on Guass' law and its uniqueness.

  2. Comment on the low-energy phase of the generalized Ising models at the self-dual point.

Recommendation

Publish (meets expectations and criteria for this Journal)

---

## Editorial Decision

resubmitted